EMBO
Molecular Medicine

# Corneal biomechanical cues mediated by PAI-2: the origin of PM2.5-induced corneal disease

Shengjie Hao[1,10], Guangsong Xie [ID][2,10], Dan Li[3,10], Kexin Su[4,10], Feiyin Sheng[1], Lu Chen[1], Yuzhou Gu[1], Hongying Jin[1], Yili Xu[1], Rongrong Chen[1], Zhenwei Qin[1], Dandan Xu[5], Peiwei Xu [ID][5], Lei Zhou[6], Na Kong[7], Hao Ding[8,9], Zhijian Chen [ID][5✉], Shuai Liu [ID][1,4✉], Baohua Ji [ID][2✉], Ke Yao [ID][1✉] & Qiuli Fu [ID][1✉]

## Abstract

**The biomechanical signature is directly correlated with the progression of disease in multiple soft tissues. However, their variations and roles, particularly during the initiation period of the disease, remain unclear. Here, we report that PM2.5 exposure induces corneal biomechanical cues alterations prior to corneal injury, as evidenced by increased corneal hysteresis in humans, thickened corneal thickness in rats, and enhanced tensile stress and cortical stiffness in HCECs. Specifically, intracellular PAI-2 is identified as a crucial mediator of the biomechanical responses in HCECs. It modulates PM2.5-induced autophagy and inflammation through a PAI-2/myosin II/F-actin/YAP-positive feedback loop, which ultimately drives HCEC injury. Furthermore, extracellular secretory PAI-2 levels in tears reflect PM2.5-related corneal damage in real time, making it a specific biomarker for the early diagnosis when combined with biomechanical cues. Early intervention for PM2.5-induced ocular damage could be achieved by developing an LNP-siPAI-2 ocular local delivery system targeting intracellular PAI-2. Overall, we propose that biomechanical cues in conjunction with specific biomarkers may serve as targets for the early diagnosis and intervention of soft tissue diseases.**

**Keywords** Corneal Biomechanics; Subclinical Stage; SerpinB2; PM2.5; Corneal Diseases
**Subject Categories** Biomarkers; Cell Adhesion, Polarity & Cytoskeleton; Molecular Biology of Disease

## Introduction

Current research has demonstrated that biomechanical cues of soft tissues directly vary with disease progression. For instance, the rheological behavior of tissues has been identified as a sensitive marker for early-stage myocardial infarction diagnosis (Chang et al, 2023a). In the liver, as the disease progresses (cirrhosis—hepatocellular carcinoma—recurrent hepatocellular carcinoma) and regresses, the biomechanical profiles of the tissue show different patterns (Chang et al, 2023b; Tian et al, 2015). However, in the initial stages of disease, when clinical symptoms have not yet manifested, i.e., the subclinical stage (Marrie et al, 2022), the variations and the roles of the biomechanical cues remain unclear.

The cornea, as the soft tissue located in the outermost region of the eye (Kong et al, 2023), is a suitable model to study the relationship between its biomechanical cues and the diseases, which have a variety of biomechanical properties, including nonlinearity, anisotropy, and viscoelasticity (Lombardo et al, 2012). As corneas are directly exposed to air, fine particulate matter less than 2.5 μm in diameter (PM2.5) has emerged as a significant contributing factor to for multiple corneal diseases (e.g., keratitis, dry eye, and corneal limbal stem cell deficiency) (Mo et al, 2019; Lee et al, 2018; Hao et al, 2023), which are often accompanied by changes in biomechanical properties at the time of clinical symptoms (i.e., clinical stage/disease stage) (Efraim et al, 2020; Marcos-Fernández et al, 2018; Robertson et al, 2021). The epithelial layer, situated on the outermost surface, is the first to interact with atmospheric PM2.5, the cytoskeleton within the epithelial cells, such as F-actin and intermediate filaments (Thomasy et al, 2024) plays a pivotal role in modulating corneal biomechanics. A rat study demonstrated that long-term exposure to PM2.5 resulted in damage to intercellular junctions (Hao et al, 2023), and a study in vitro showed that PM2.5 induced dysfunction of cytoskeletal F-actin in corneal epithelial cells (Cui et al, 2018), implying that the biomechanical properties of corneal epithelium could be affected by PM2.5. Nevertheless, the alterations and roles of biomechanical cues in PM2.5-induced corneal disease during the subclinical stage have yet to be addressed.

It is imperative to elucidate the manner in which biomechanical cues contribute to the PM2.5-induced injury. A significant advancement in the comprehension of this matter has been the identification of the mechanotransduction pathways, such as the

[1]Zhejiang University, Eye Center of Second Affiliated Hospital, School of Medicine; Zhejiang Provincial Key Laboratory of Ophthalmology; Zhejiang Provincial Clinical Research Center for Eye Diseases; Zhejiang Provincial Engineering Institute on Eye Diseases, Hangzhou, Zhejiang Province, China. [2]Biomechanics and Mechanomedicine Laboratory, Department of Engineering Mechanics, Zhejiang University, Hangzhou, Zhejiang Province, China. [3]Hangzhou Heima Eye Clinic, Hangzhou, Zhejiang Province, China. [4]College of Pharmaceutical Sciences, Zhejiang University, Hangzhou, Zhejiang Province, China. [5]Department of Environmental Health, Zhejiang Provincial Center for Disease Control and Prevention, Hangzhou, Zhejiang Province, China. [6]School of Optometry, Department of Applied Biology and Chemical Technology, Hong Kong Polytechnic University, Hong Kong, China. [7]Liangzhu Laboratory, Zhejiang University School of Medicine, Hangzhou, Zhejiang Province, China. [8]Key Laboratory of Environmental Pollution Control Technology of Zhejiang Province, Hangzhou, Zhejiang Province, China. [9]Environmental Science Research & Design Institute of Zhejiang Province, Hangzhou, Zhejiang Province, China. [10]These authors contributed equally: Shengjie Hao, Guangsong Xie, Dan Li, Kexin Su. ✉E-mail: zhjchen@cdc.zj.cn; shuailiu@zju.edu.cn; bhji@zju.edu.cn; xlren@zju.edu.cn; 2313009@zju.edu.cn

FAK/RhoA, Wnt, and TGF-β pathways (Thomasy et al, 2024). In the FAK/RhoA pathway, myosin II activation and F-actin polymerization can alter the localization of YAP/TAZ by modulating phosphorylation modifications, which are essential for regulating cellular functions (Panciera et al, 2017). YAP and TAZ are currently the most widely studied mechanical effector molecules. They shuttle between the cytoplasm and the nucleus depending on their phosphorylation modification and serve as transcription co-activators to interact with intranuclear transcription factor TEAD1 and regulate transcription (Panciera et al, 2017). Previous reports have indicated that PM2.5 can disturb integrin, cytoskeleton organization, and FAK/RhoA signaling pathway (Cui et al, 2018), suggesting that cell damage may be associated with mechanotransduction. However, the exact role of the potential mechanotransduction pathway in PM2.5-induced corneal injury needs further exploration。

Our previous work has identified plasminogen activator inhibitor-2 (PAI-2) as a major contributor to PM2.5-induced corneal injury (Lyu et al, 2020). PAI-2, also known as SerpinB2, is a member of the serine protease inhibitor protein family. It is noteworthy that there are several studies that indicated the correlation between PAI-2 and tissue mechanics, suggesting that PAI-2 may be the primary regulator of corneal biomechanics. For instance, PAI-2 has been found to co-localize with actin in focal adhesions (Schroder et al, 2019). Furthermore, Kindlin-2, a member of an actin cytoskeleton-organizing and integrin activator protein, has been shown to regulate PAI-2 in breast cancer (Sossey-Alaoui et al, 2019). In addition, PAI-2 can be secreted extra-cellularly, and its levels in body fluids can vary in response to a multitude of physiologic and pathologic conditions, such as pregnancy and infection (Boncela et al, 2013). Thus, the level of PAI-2 in tears may serve as a real-time indicator of corneal health, yet few studies have been conducted in this regard.

This study employed a multidisciplinary approach to investigate the corneal biomechanical cues in cellular, animal, and human models during the subclinical stage of PM2.5-induced corneal disease (Fig. 1A). Our findings elucidated that the corneal biomechanical cues change precede the onset of clinical symptoms and drive disease onset; therefore, they may characterize the subclinical stage. We revealed the mechanism by which intracellular PAI-2 regulates the cellular biomechanical response and triggers damage in corneal epithelial cells, explored early diagnostic markers in tears, and developed an early intervention ocular local delivery system based on lipid nanoparticle (LNP)-encapsulated small interfering RNA technology.

# Results

## Corneal biomechanical cues changes precede clinical symptoms under PM2.5 exposure in humans and rats

Epidemiologic studies have shown that chronic exposure to PM2.5 increases the risk of multiple corneal diseases, such as keratitis, dry eye, and pterygium. To investigate the corneal biomechanical cues in the subclinical stage, we collected corneal biomechanical data from subjects without clinical symptoms under short-term PM2.5 exposure. Biomechanical data were obtained from the eye clinic by Corvis ST (Fig. 1A). Subjects were divided into two groups

according to the daily average atmospheric PM2.5 concentration on the day of the examination: high-exposure group (PM2.5 ≥ 75 μg/cm³) and low-exposure group (PM2.5 < 35 μg/cm³) (Fig. 1C). Demographic characteristics of subjects and detailed individual information were showed in Tables EV1 and EV2. A total of 18 parameters were collected and analyzed, of which stiffness parameter of the first applanation (SP-A1), deformation amplitude (DA), DA ratio, peak distance (PD), corneal velocity during the first applanation moment (A1V), corneal velocity during the second applanation moment (A2V), and maximum keratometry (Kmax) were significantly different between the two groups (Figs. 1B and EV1A). The results showed that DA, DA ratio, A1V, PD, and Kmax were significantly lower in the high-exposure group compared to the low-exposure group, and SP-A1 and A2V were notably elevated in the high-exposure group (Fig. 1B). Our findings suggested that changes in corneal biomechanical cues precede clinical symptoms under PM2.5 exposure in humans.

We also constructed a rat model of PM2.5 exposure and examined corneal biomechanical properties and injury after short-term (2-day) and long-term (3-week) PM2.5 exposure, respectively (Fig. 1A). The results showed that there was no obvious corneal epithelial cell defect after short-term exposure (Fig. 1D), indicating that it was still in the subclinical stage, but the corneal biomechanical properties were significantly altered, which was manifested by thickening cornea as shown by the anterior segment-OCT (AS-OCT) (Fig. 1E), H&E staining demonstrated an augmented thickness in both the corneal epithelial and the stromal layers (Fig. EV1B), and corneal hydration was significantly elevated (Fig. 1F). In contrast, long-term PM2.5 exposure brought the disease into the clinical stage, characterized by significant corneal epithelial defects (Fig. 1D) and reduced corneal epithelial and stromal thicknesses (Figs. 1E and EV1B). These findings confirmed that the corneal biomechanical properties altered earlier than PM2.5-induced corneal clinical symptoms.

## Cellular mechanical cues changes precede cell damage and trigger it under PM2.5 exposure

The corneal epithelium covers the outermost layer of the ocular surface and is more susceptible to irritation by environmental exposure, such as PM2.5. Therefore, we detected the biomechanical cues of human corneal epithelial cells (HCECs) under PM2.5 exposure (Fig. 1A). Our previous study showed that PM2.5 damage to HCECs was not apparent in the first 6 h but worsened over time (Fu et al, 2017), implying there is a subclinical phase of PM2.5-induced cellular damage. In this study, cellular mechanical responses were significantly activated within the first 6 h following PM2.5 exposure, as evidenced by enhanced activity of non-myosin II—a key driver of cellular mechanical behavior—specifically manifested by a marked increase in its phosphorylation levels (Fig. 2A). To further confirm this, cellular tensile stress was evaluated by traction force microscopy (TFM), and results showed that cellular tensile stress was enhanced upon 3-h exposure (Fig. 2B). Cellular cortical stiffness was measured through an atomic force microscope (AFM), and the results showed an increase in Young's modulus upon 3-h exposure, indicating enhanced cell cortical stiffness (Fig. 2C). These findings suggested a significant biomechanical response in cells during the subclinical stage of PM2.5-induced cellular damage (Fig. 2F).

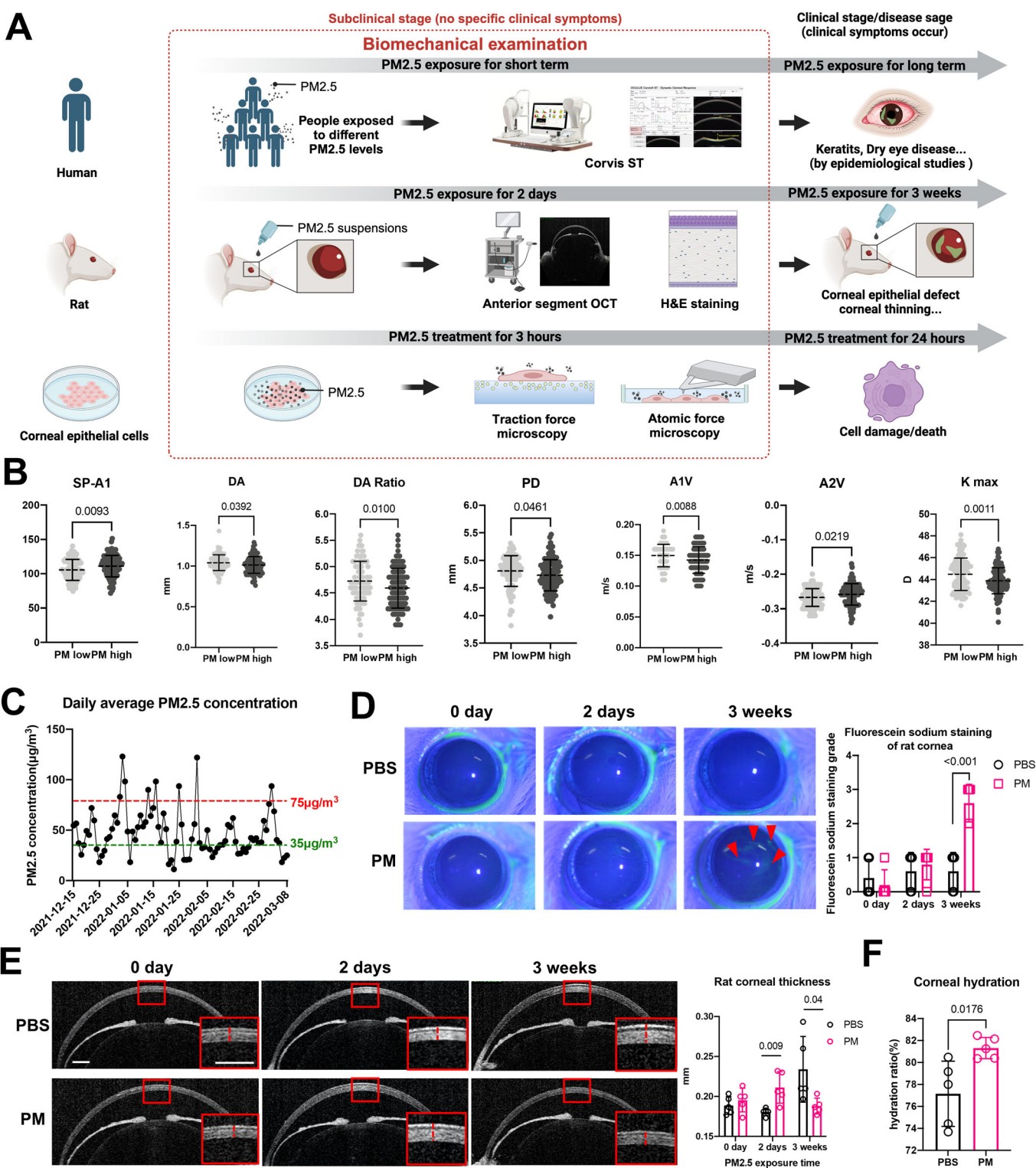

To understand the role of the biomechanical response in PM2.5-induced cellular damage, blebbistatin (non-muscle myosin II inhibitor, blebb) and Y-27632 (rho-associated protein kinase inhibitor, RI) were used to separately inhibit myosin II or its upstream protein rho-associated protein kinase (ROCK). The cellular tensile stress of HCECs

was notably suppressed in response to blebb (Fig. 2B). Interestingly, damage induced by prolonged (24 h) PM2.5 exposure was attenuated if the biomechanical response of the subclinical stages was limited (Fig. 2D), illustrating that cellular biomechanical response in the subclinical stage triggers PM2.5-induced cell damage. In addition, the

◄ **Figure 1. Short-term PM2.5 exposure changes corneal biomechanical cues in humans and rat models.**

(A) The workflow of biomechanical examination in humans, rats, and cells. Under short-term PM2.5 exposure, human corneal biomechanical properties were examined by Corvis ST, rat corneal biomechanical properties were examined by anterior segment OCT and H&E staining, and cellular biomechanical properties were examined by traction force microscopy and atomic force microscopy. (B) Corneal biomechanical parameters from Corvis ST in PM2.5 high-exposure group (PM2.5 ≥ 75 μg/cm³, $n = 113$) and PM2.5 low-exposure group (PM2.5 < 35 μg/cm³, $n = 105$). (C) Daily average PM2.5 concentration in Hangzhou City, Zhejiang Province, China. Air quality monitoring data were supported by the Zhejiang Provincial Center for Disease Control and Prevention. (D) The corneal epithelium defect was examined by slit lamp examination using fluorescein sodium staining under cobalt-blue light. The fluorescein sodium staining grades were evaluated. $n = 5$. The red arrows pointed to the corneal epithelium defect. (E) Rat corneal thickness was measured by anterior segment OCT in two groups (scale bar, 250 μm). $n = 5$. The red dotted line refers to the corneal thickness. (F) Rat corneal hydration rate was elevated after short-term PM2.5 exposure. $n = 5$. Data in (B, D–F) are graphed as mean ± standard deviation with individual values shown as dots, circles or squares. Statistical analysis was conducted using the unpaired $t$ test in (B, D–F). The P values are labeled in the figure. Source data are available online for this figure.

cellular mechanical response was also detected in rat models, as evidenced by the increased circularity of the rat corneal epithelial cells under short-term PM2.5 exposure (Fig. 2E).

In conclusion, short-term PM2.5 exposure did not cause significant damage to HCECs, rat cornea, and human corneal tissues, except for significant biomechanical cues changes, which is a precursor to PM2.5-induced corneal diseases (Fig. 2F).

## A transcriptome analysis suggests a potential correlation between PAI-2 and cellular mechanical cues

We previously found that PAI-2 mediates long-term PM2.5 exposure-induced corneal damage in vivo and in vitro (Lyu et al, 2020), while its role during the subclinical stage remains unclear. In this study, we found that the expression of PAI-2 was elevated upon short-term PM2.5 exposure in rat corneal superficial epithelial cells and HCECs (Figs. 3A–C and EV2A), illustrating a possible relationship between PAI-2 and corneal biomechanical cues. Furthermore, we constructed PAI-2 knockout human corneal epithelial cells (KO) and the negative control (NC). The efficiency of the PAI-2 knockout was evaluated on the mRNA and protein levels (Fig. EV2B,C). After PAI-2 knockout, the cellular viability of HCECs was significantly improved under 24-h PM2.5 exposure (Fig. 3D).

Then, transcriptome analysis was performed in three groups: NC, NC with 3-h PM2.5 exposure, and KO with 3-h PM2.5 exposure. The heatmap is shown in Fig. 3E. As illustrated in the Venn diagram, we found that 2077 genes were significantly altered in NC after PM2.5 exposure, and 1219 of these genes were affected after PAI-2 knockdown (Fig. 3F). Therefore, we extracted these 1219 genes for further gene function enrichment analysis. In GO (Gene Ontology) analysis, we observed several alterations related to cellular mechanics, such as actin cytoskeleton and myosin complex in cellular components (CC), and Rho protein signal transduction in biological processes (BP) (Fig. 3G). KEGG (Kyoto Encyclopedia of Genes and Genomes) analysis also revealed a significant change in the Hippo signaling pathway (Fig. 3H), which has been reported to participate in biomechanical signal transduction in multiple studies (Zeybek et al, 2021). These analyses suggested cellular mechanical cues might play an important role in PAI-2-induced corneal injury upon PM2.5 exposure (Fig. 3I).

## PAI-2 mediates cellular biomechanical response through myosin II/F-actin in response to PM2.5 exposure

To confirm the role of PAI-2 in corneal cell mechanical response, we further established PAI-2-overexpressing HCECs (OE) and the

negative control (OENC) (Fig. EV2D,E), and we examined phosphorylated myosin II in PAI-2 knockout and overexpressing HCECs. After 1–6 h of PM2.5 exposure, the level of myosin II phosphorylation in KO was significantly lower than that in NC (Figs. 4A and EV3A), while the OE group showed higher myosin II phosphorylation levels than the OENC group (Figs. 4B and EV3B), indicating PAI-2 regulated myosin II activity. Results of TFM confirmed that the cellular tensile stress was enhanced in NC upon exposure, but they were limited in KO (Fig. 4C). Results of AFM showed that cellular cortical stiffness in NC was increased and was higher than KO as well (Fig. 4D). All of these findings implied that PAI-2 mediated cellular mechanical response at the subclinical stage of PM2.5 exposure.

Since myosin II regulates F-actin polymerization (Panciera et al, 2017), we further explored the changes in F-actin under PM2.5 exposure. It was found that short-term (2-day) PM2.5 caused a significant enhancement of F-actin polymerization in vivo (Fig. 4E). In vitro, short-term (3 h) PM2.5 exposure induced similar phenomena, while F-actin polymerization was suppressed following PAI-2 knockout (Fig. 4F,G) and enhanced upon PAI-2 overexpression (Fig. EV4A). Upon inhibition of myosin II or its upstream by the addition of blebb or RI, F-actin polymerization was also blocked (Fig. 4F,G), suggesting that PAI-2 could regulate F-actin polymerization by influencing the myosin II activation.

It is worth mentioning that we noticed the presence of a ring-like actin structure, the perinuclear actin ring (PAR), around the nucleus in the resting state of the cell (Fig. 4F). This structure disappeared with PM2.5-induced F-actin polymerization, and it can be efficiently maintained by inhibiting F-actin polymerization using myosin II inhibitors (RI or blebb) (Fig. 4F,H), Thus, there is a PAR/F-actin transition in cells during the mechanotransduction process. Additionally, more PARs were retained in PM2.5-exposed KO compared to NC, and promotion of F-actin polymerization using the F-actin stabilizer jasplakinolide (jasp) significantly inhibited PAR synthesis in KO (Fig. 4F,G). Upon overexpressing PAI-2, we also noticed less PAR in PM2.5-exposed OE compared to OENC under PM2.5 exposure (Fig. EV4A). All the above suggest that the PAR/F-actin transition is regulated by PAI-2.

Moreover, we investigated the role of PAR under PM2.5 exposure. We found that inhibiting PAR synthesis with jasp exacerbated PM2.5-induced cellular damage in KO (Fig. EV4C). Similarly, cytochalasin D (cytoD, an F-actin depolymerizer) can disrupt actin structures, therefore also leading to reduced PAR levels (Fig. 4F,H) and consequently increased cellular damage (Fig. EV4C). All of these implied that PAR may play an important role in protecting cells.

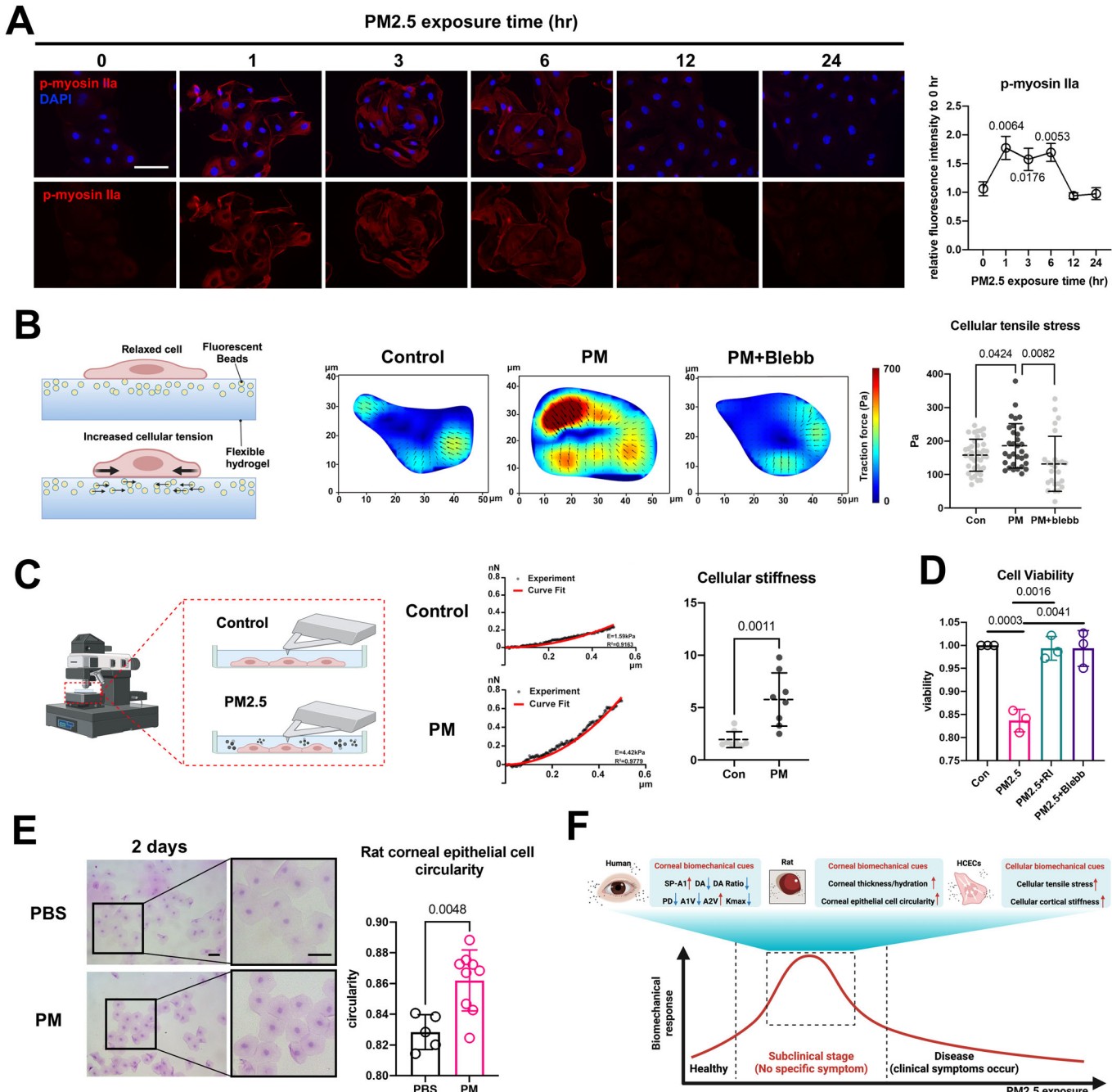

**Figure 2. Short-term PM2.5 exposure activates cellular mechanical response and triggers cell damage.**

(A) Immunofluorescence staining for phosphorylated non-muscle myosin IIa of HCECs exposed for different times to PM2.5 (scale bar, 75 μm) ($n = 3$ biological replicates). (B) Cellular tensile stress of HCECs exposed to PM2.5 (with or without blebb) for 3 h was measured by traction force microscopy ($n = 40$ for control, $n = 32$ for PM, $n = 25$ for PM+blebb). (C) Cellular stiffness of HCECs exposed to PM2.5 for 3 h measured by atomic force microscopy ($n = 8$). (D) Cellular viability of HECEs in the presence of RI or blebb ($n = 3$ biological replicates). (E) Impression cytology was used to evaluate rat corneal epithelial cell circularity in two groups (scale bar, 25 μm). $n = 5$ (PBS), 9 (PM). (F) PM2.5 exposure leads to the activated biomechanical response at the subclinical stage in humans, rats, and HCECs. This curve provides a conceptual illustration of the correlation between biomechanical response and PM2.5-induced corneal disease progression; it does not reflect precise numerical changes. Data in (A–E) are graphed as mean ± standard deviation with individual values shown as dots or circles. Statistical analysis was conducted using the unpaired $t$ test in (A–E). The $P$ values are labeled in the figure. Source data are available online for this figure.

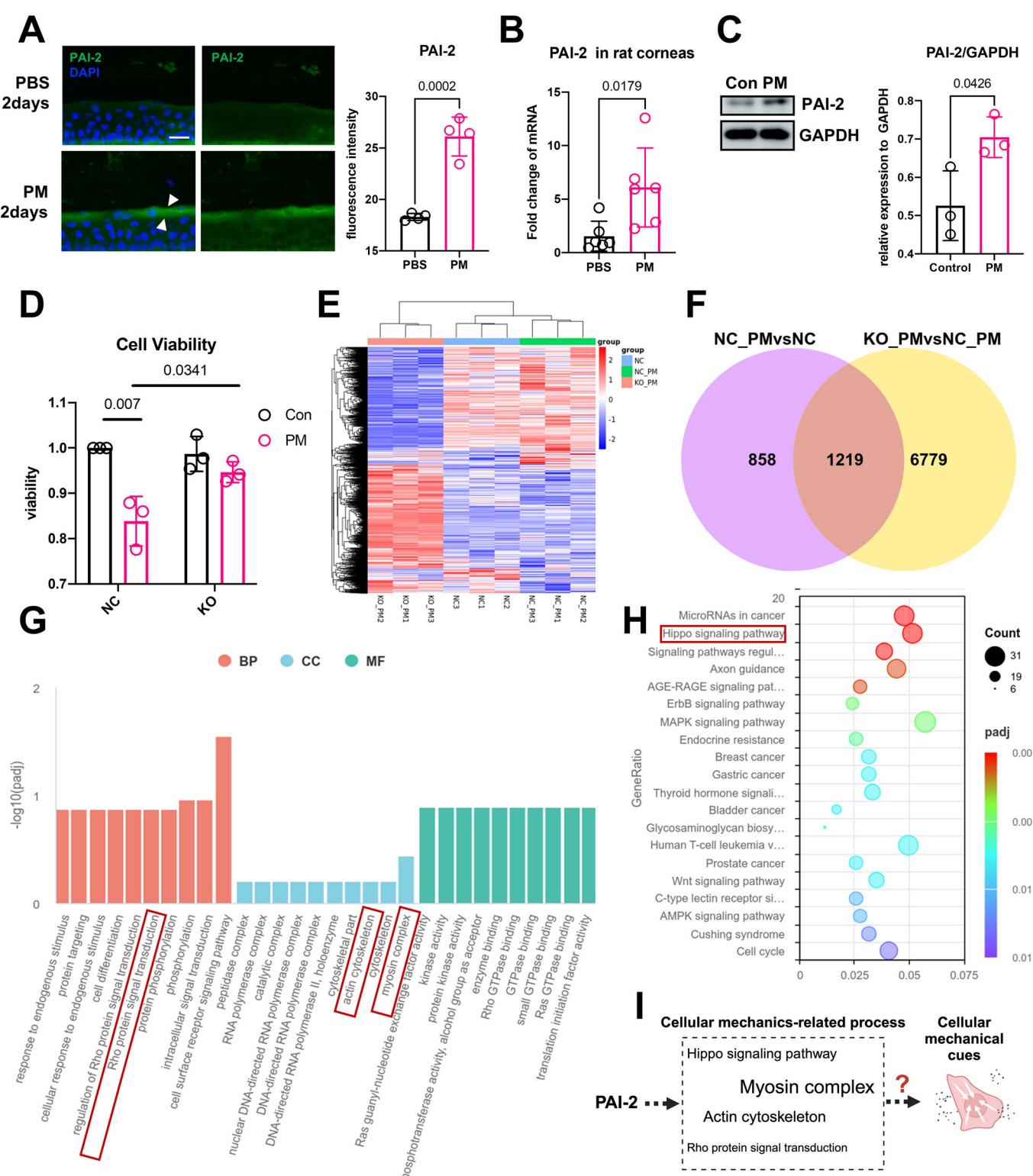

## PAI-2/myosin II/F-actin/YAP-positive feedback loop regulates autophagy and IL-1a in PM2.5-exposed corneal epithelial cells

YAP/TAZ are co-transcriptional factors shuttling between the nucleus and cytoplasm, and F-actin polymerization reduces its

phosphorylation level, which further facilitates YAP/TAZ nuclear translocation and bound with TEAD1 to initiate transcription of downstream genes (Panciera et al, 2017). For the clinical stage with corneal damage, this study found that long-term PM2.5 exposure accelerated YAP nuclear translocation in rat cornea (Fig. 5A,C) and inhibited the phosphorylation level of YAP (Fig. 5E), which could be

◄

**Figure 3. A transcriptome analysis suggests a potential correlation between PAI-2 and cellular mechanical cues.**

(A) PAI-2 staining in rat cornea after short-term (2-day) PM2.5 exposure (scale bar, 25 μm). The while arrow refers to the rat corneal superficial epithelial cells. n = 4. (B) The mRNA level of PAI-2 in rat corneas after short-term (2-day) PM2.5 exposure. n = 4. (C) The protein level of PAI-2 in HCECs after short-term (3-h) PM2.5 exposure (n = 3 biological replicates). (D) The viability of NC and KO after 24-h PM2.5 treatment (n = 3 biological replicates). (E) The heatmap showed the different expression profiles in three groups. (F) The Venn diagram demonstrated the common DEGs between the two sets of comparisons. (G) GO analysis revealed biological process, cellular component, and molecular function enriched in DEGs. (H) KEGG pathway analysis of DEGs. (I) PAI-2 is associated with several cellular mechanics-related processes, which may regulate PM2.5-induced cellular mechanical response. Data in (A–D) are graphed as mean ± standard deviation with individual values shown as circles. Statistical analysis was conducted using the unpaired t test in (A–D). The P values are labeled in the figure. Source data are available online for this figure.

related to the F-actin polymerization during the subclinical stage. In vitro, we observed the same results in NC under long-term PM2.5 exposure, while PAI-2 knockout resulted in limited YAP nuclear translocation with higher phosphorylation of YAP (Fig. 5B,D,F), and PAI-2 overexpression led to accelerated YAP nuclear translocation (Fig. EV4B), suggesting PAI-2 regulated YAP nuclear translocation. The use of myosin II inhibitor (blebb, RI) or F-actin depolymerizer (cytoD) all led to the higher phosphorylation of YAP with limited YAP nuclear translocation in cells (Fig. EV5A–C), proving that PAI-2 regulated YAP nuclear translocation according to myosin II /F-actin.

Our previous study has pointed out that PAI-2 mediates accelerated autophagy in PM2.5-exposed corneal epithelial cells (Lyu et al, 2020). This study further found that besides the autophagy-related proteins (BECN1 and ATG5), inflammatory factor IL-1a can be initiated and regulated by PAI-2 as well in vivo and in vitro during the clinical stage (Figs. 5F and EV5D). Interestingly, limited YAP translocation caused by blebb and RI resulted in the alleviation of autophagy-related proteins (BECN1 and ATG5) and IL-1a (Fig. EV5B), which indicated that PM2.5 induced autophagy and IL-1a via PAI-2/myosin II/F-actin/YAP mechanotransduction pathway. We further used verteporfin (VTP) to block YAP-TEAD1 interaction, and the protein levels of BECN1, ATG5, and IL-1a were also notably decreased, indicating that PM2.5 induced autophagy and IL-1a via YAP-TEAD1 interaction (Fig. 5G).

Intriguingly, the expression of PAI-2 protein was also remarkably inhibited in the presence of blebb, RI, or VTP (Figs. 5G and EV5B), suggesting that PAI-2 and YAP could reciprocally regulate each other, and there was a PAI-2/myosin II/F-actin/YAP-positive feedback loop (Fig. 5H). In addition, VTP likewise reduced the expression of BECN1, ATG5 at the mRNA level (Fig. EV5E), suggesting that YAP-TEAD1 may be involved in the regulation of their transcription; however, the mRNAs of PAI-2 and IL-1a were instead elevated under the action of VTP (Fig. EV5E), implying that there may be a post-transcriptional regulation of them by YAP-TEAD1.

In summary, our findings indicate that PAI-2 plays a pivotal role in mediating the cellular mechanical response to PM2.5 exposure during the subclinical phase of the disease. PAI-2 activation led to the acceleration of the PAR/F-actin switch and F-actin polymerization, resulting in elevated cellular tensile stress and cortical stiffness. The PAI-2/myosin II/F-actin/YAP-positive feedback loop enhanced autophagy and inflammation, ultimately contributing to the progression of the disease into the clinical stage (Fig. 5H).

## Extracellular PAI-2 has the potential as a specific diagnostic indicator of PM2.5-induced corneal injury

Based on the above findings, there is a great need to find sensitive biomarkers reflecting PM2.5-induced corneal damage progression

to enable specific intervention. Therefore, we collected rat tear fluid 2 days after PM2.5 exposure for mass spectrometry (Fig. 6A). Heatmap and volcano plot are presented in Fig. EV6A,B. A total of 51 proteins were significantly differentially expressed, including 21 up-regulated proteins and 30 downregulated proteins (Fig. EV6B). Subcellular localization, GO, KEGG pathways, and PPI network analysis were performed based on 51 differentially expressed proteins (DEPs) (Fig. EV6C–F). In GO and KEGG analysis, DEPs were mainly enriched in actin polymerization-associated processes, such as "actin cytoskeleton organization" and "actin filament-based process" in BP, "structural constituent of cytoskeleton" in MF, and "regulation of actin cytoskeleton" in KEGG (Fig. EV6D,E). In PPI, we further found that DEPs located at the extracellular region (red spots) were tightly associated with actin cytoskeleton organization-related DEPs (purple spots) (Fig. EV6F), so actin organization-related proteins may serve as the specific PM2.5-related damage markers.

We have proved that PAI-2 can regulate actin organization, as it also serves as a secreted protein, we wonder if extracellular PAI-2 has the potential to serve as a specific PM2.5-related damage marker. We evaluated PAI-2 content in rat tears during PM2.5 exposure (Fig. 6A). Results showed that the level of PAI-2 in tears decreased with exposure time (Fig. 6B). The consistent results were also obtained in supernatant of PM2.5-exposed HCECs (Fig. 6C), which meant that PM2.5 affected PAI-2 exocytosis, and the reduction of extracellular PAI-2 level in tears may reflect the PM2.5-induced corneal damage.

We constructed a human panel study to explore the relationship between PM2.5 exposure and PAI-2 levels in tear fluid. A total of 28 participants were recruited in Jinhua City, Zhejiang Province, China. Panel study population characteristics and detailed individual information are shown in Tables EV3 and EV4. Each participant completed 24 h of continuous individual exposure monitoring followed by a survey questionnaire, ophthalmic examinations, and tear collection for PAI-2 detection (Fig. 6A). Summary Statistics of 24 h mean individual PM2.5 and its constituents are shown in Table EV5.

Spearman correlation analysis was performed to analyze the correlation between the level of PM2.5 exposure with the ocular disease risk, ocular surface damage, and PAI-2 content in tears in humans. According to the results of the questionnaire, we found that the Ocular Surface Disease Index (OSDI) of the population increased with increasing levels of exposure to PM2.5, suggesting that PM2.5 contributes to the increased risk of developing ocular surface diseases (Fig. 6D). Ophthalmologic examination showed that the eyesight was not significantly affected by PM2.5, whereas the tear film break-up time and tear secretion level decreased with PM2.5 exposure, indicating that PM2.5 impaired tear film stability

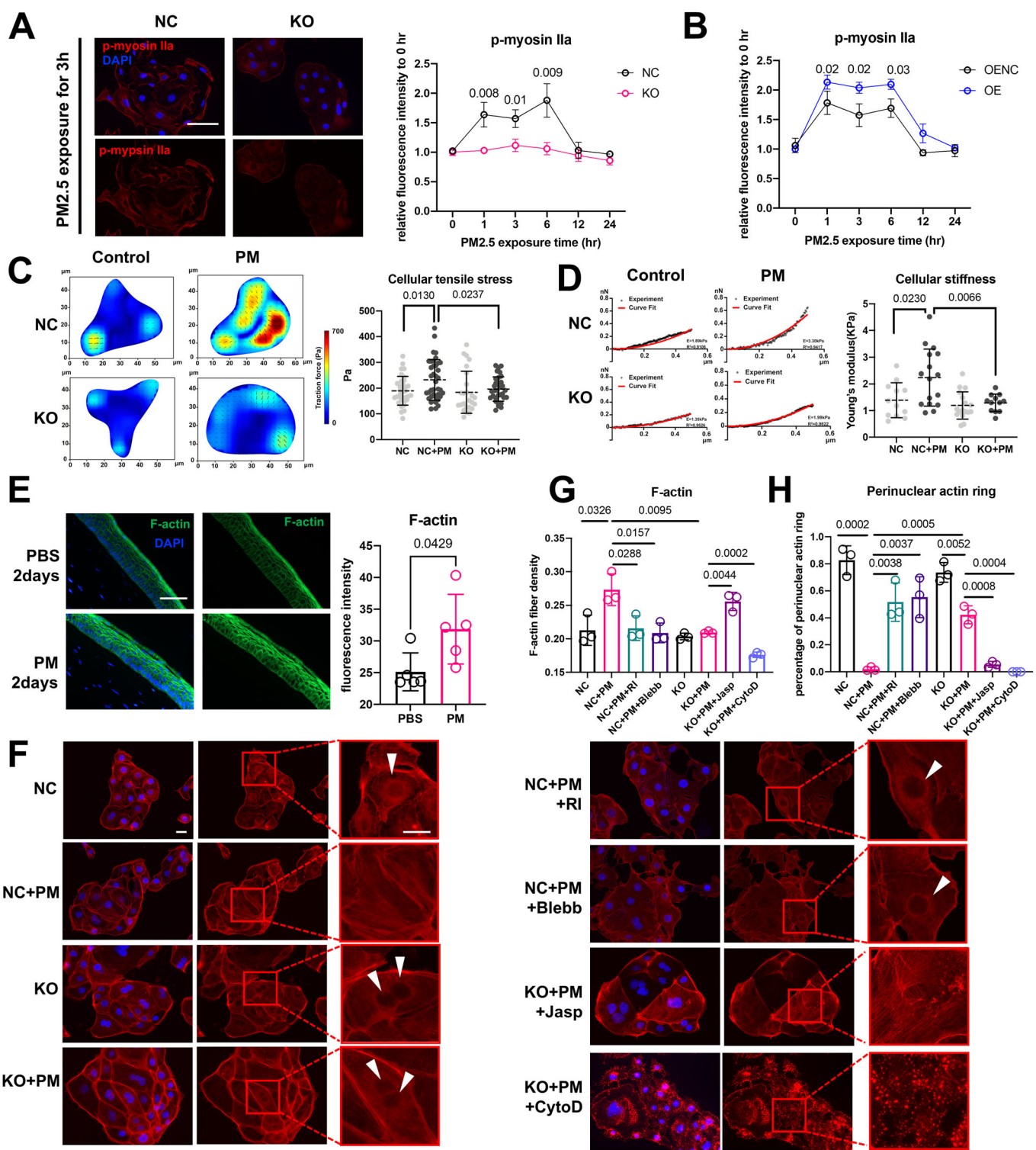

(Fig. 6E–G). The level of corneal epithelial defects increased with PM2.5 exposure, suggesting that PM2.5 induced epithelial cell damage (Fig. 6H). We found that PAI-2 levels in tears were negatively correlated with PM2.5 exposure in humans (Fig. 6I), which is consistent with results on the animal and cellular models,

suggesting that PAI-2 may be able to reflect the PM2.5-induced corneal damage.

In addition, we analyzed the independent effects of each component of PM2.5 on tear PAI-2 content using the Bayesian kernel machine regression (BKMR) model. Posterior inclusion

probabilities (PIPs) were calculated to characterize the importance of each exposure variable in the mixture. Exposure variables with PIPs >0.5 in the BKMR model were considered important components of the mixture. All variables we tested had PIPs above 0.5, of which Se, Hg, Cd, Al, Cr, Ti, OC (organic carbon), and As gave higher PIPs (>0.9) values than other variables for PAI-2, indicating higher importance (Fig. 6J).

**Figure 4.  PAI-2 mediates cellular biomechanical cues response through myosin II/F-actin remodeling during the subclinical stage under PM2.5 exposure.**

(**A**) Phosphorylated non-muscle myosin IIa staining in NC and KO with PM2.5 exposure (scale bar, 75 μm) ($n = 3$ biological replicates). This panel shows only the NC and KO images of phosphorylated myosin IIa after 3 h of PM2.5 exposure. The full-time course images from 0 to 24 h of PM2.5 exposure are shown in Fig. EV3A. (**B**) Phosphorylated non-muscle myosin IIa staining in OENC and OE with PM2.5 exposure (scale bar, 75 μm) ($n = 3$ biological replicates). (**C**) Cellular tensile stress of NC and KO exposed to PM2.5 for 3 h measured by TFM ($n = 32$ for NC, $n = 38$ for NC + PM, $n = 25$ for KO, $n = 34$ for KO + PM). (**D**) Cellular stiffness of NC and KO exposed to PM2.5 for 3 h measured by AFM ($n = 12$ for NC, $n = 16$ for NC + PM, $n = 17$ for KO, $n = 12$ for KO + PM). (**E**) F-actin staining of rat corneas in two groups (scale bar, 75 μm). $n = 5$. (**F**) Staining of F-actin in NC and KO exposed to PM2.5 for 3 h and staining of F-actin in the presence of 10 μM RI, 10 μM Blebb, 2 nM Jasp or 1 nM CytoD (scale bar, 25 μm). The white arrow pointed to PAR. (**G**) Statistics of F-actin fiber density in (**F**) ($n = 3$ biological replicates). (**H**) Statistics of PARs in (**F**) ($n = 3$ biological replicates). Data in (**A–E, G, H**) are graphed as mean ± standard deviation with individual values shown as dots or circles. Statistical analysis was conducted using the unpaired $t$ test in (**A–E, G, H**). The $P$ values are labeled in the figure. Source data are available online for this figure.

Based on the above findings, we further examined the univariate exposure-response relationships between the eight prioritized mixture components and PAI-2 levels. The exposure-response curves exhibited approximately linear associations for all components except Se (which showed a non-linear trend). Notably, PAI-2 levels were inversely associated with Hg, Cd, Al, and organic carbon (OC), suggesting that these constituents may be key contributors to reduced PAI-2 in tears. However, we interpret these univariate findings with caution given the limited sample size, further validation in larger cohorts is warranted (Fig. 6K).

All of these results revealed that the PM2.5 exposure level was associated with the risk of corneal disease and corneal damage, and the extracellular PAI-2 content of tears varied with the exposure level, suggesting that PAI-2 levels of tears could be a potential specific indicator for PM2.5 exposure-induced corneal disease.

### Targeting intracellular PAI-2 using an LNP-siPAI-2 ocular local delivery system relieves PM2.5-induced ocular damage

To explore whether PAI-2 can serve as a key early intervention target for PM2.5-induced corneal injury, we developed an LNP-siPAI-2 ocular local delivery system based on commercial DLin-MC3-DMA LNPs. Its efficacy was tested in an animal model of long-term PM2.5 exposure (Fig. 7E). Initially, we delivered luciferase mRNA via DLin-MC3-DMA LNPs, thereby demonstrating that DLin-MC3-DMA LNPs is capable of delivering its cargo to rat corneal epithelial cells via eye drops (Fig. EV7A). Following the clarification of the knockdown efficiency of siPAI-2 (Fig. EV7B), LNP-siPAI-2 was delivered topically to the ocular surface of rats concurrently with 3-week PM2.5 exposure. The rat ocular surface injury was examined after 3 weeks, and it was found that LNP-siPAI-2 significantly attenuated the PM2.5-induced corneal epithelial defects in the rats (Fig. 7A). H&E staining also demonstrated that the thickness of the rat corneal epithelium was effectively restored (Fig. 7B). Immunofluorescence staining demonstrated that LNP-siPAI-2 suppressed the PM2.5-induced elevated expression of PAI-2 in corneal epithelial cells (Fig. 7C). The autophagy-related protein LC3B and IL-1a were also inhibited by LNP-siPAI-2 (Fig. 7C,D). These findings suggest that this treatment system can

effectively mitigate PM2.5-induced corneal damage and has the potential for early intervention (Fig. 7E).

In addition, we found that long-term exposure to PM2.5 also led to intraocular damage, as shown by abnormal morphology of rat lens epithelial cells and reduced retinal thickness, and the LNP-siPAI-2 system also attenuated these pathologic alterations (Figs. 7B and EV7D), which may be related to the reduction of inflammation within the eye (Fig. EV7C). These results indicate a potential correlation between corneal epithelial injury and PM2.5-induced damage in other ocular regions. Early intervention targeting the corneal epithelium may prove effective in preventing PM2.5-induced intraocular disorders.

## Discussion

As is recorded in the Inner Canon of Huangdi, a classic of traditional Chinese medicine, "Treating disease by preventing illness before it began, just as a good government or emperor was able to take the necessary steps to avert war", which emphasized the importance of prevention at the subclinical stage. Our study, modeling PM2.5-induced corneal disease, for the first time found that changes in corneal biomechanical properties precede the onset of clinical symptoms in humans, animals, and cells, suggesting a key role for biomechanical cues in driving disease progression during the subclinical phase. Intracellular PAI-2, previously identified as a key molecule contributing to corneal toxicity, was further elucidated to regulate cellular mechanical response in corneal epithelial cells, and trigger exacerbated autophagy and inflammation through a PAI-2/myosin II/F-actin/YAP-positive feedback loop under PM2.5 exposure, ultimately leading to cellular injury. In light of these findings, we proposed that extracellular PAI-2 in tear fluid can be combined with corneal biomechanical properties as early diagnostic indicators and LNP-siPAI-2 ocular local delivery system as an early intervention strategy, which enable the prevention of PM2.5-induced corneal and whole eye injuries/diseases at the subclinical stage.

In this study, we confirmed the corneal biomechanical cues change in the subclinical stage of PM2.5-induced corneal diseases at three levels: tissue, cellular, and cytoskeletal. First, at the corneal tissue level, short-term PM2.5 exposure resulted in increased corneal hysteresis in humans. In other studies, smoking was found to be associated with enhanced corneal hysteresis (Stuart et al, 2024; Hafezi, 2009), further confirming the effect of environmental exposure on corneal biomechanics. Second, a notable mechanical response was observed in HCECs in vitro at the early phase of

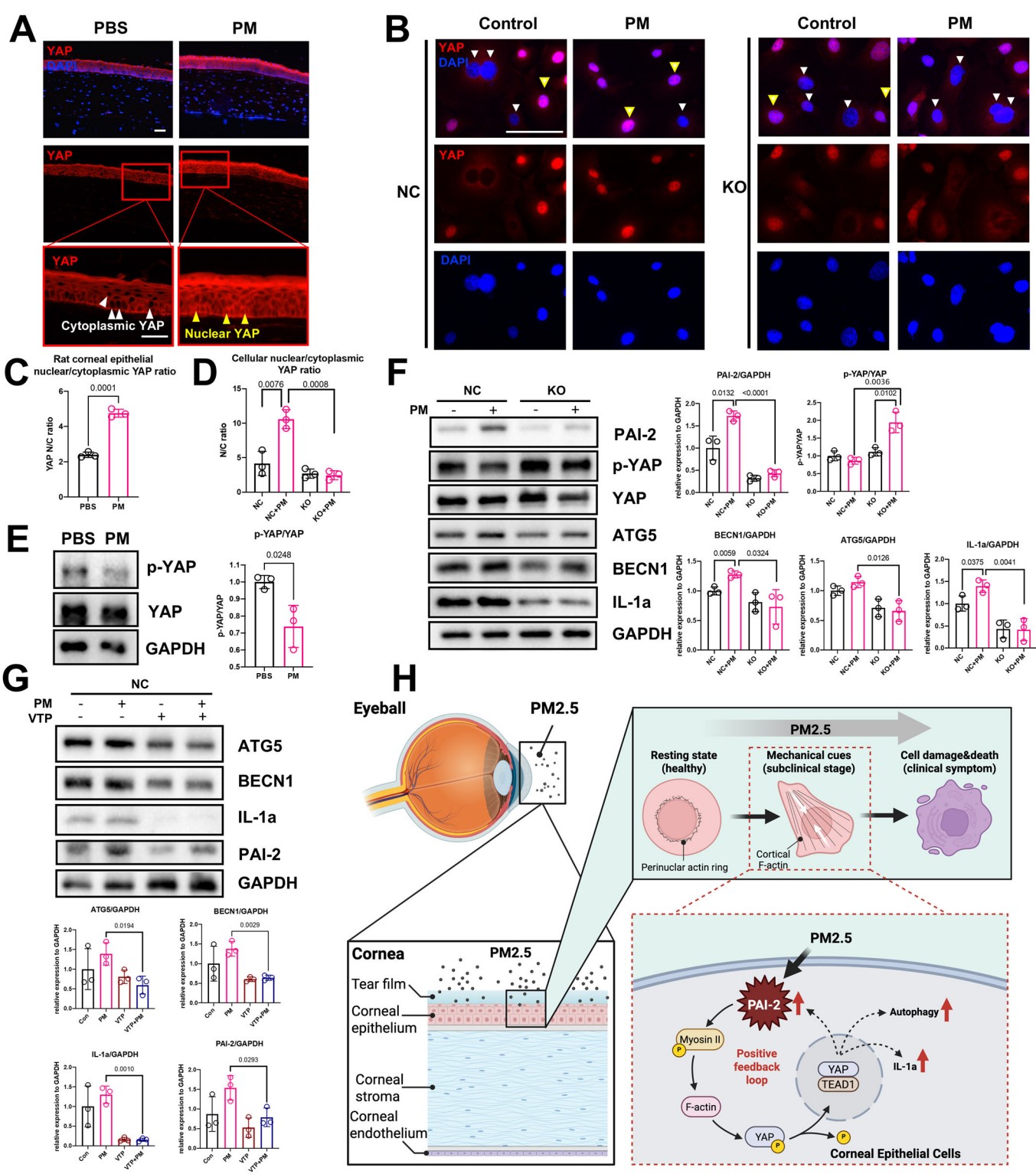

**Figure 5.  PAI-2/myosin/F-actin/YAP-positive feedback loop regulates autophagy and IL-1a in PM2.5-exposed corneal epithelial cells.**

(A, C) YAP staining and nuclear/cytoplasmic YAP ratio of rat corneas with 3-week PM2.5 exposure (scale bar, 50 μm). The white arrow indicates cytoplasmic YAP, the yellow arrow indicates nuclear YAP. $n = 3$. (B, D) YAP staining and nuclear/cytoplasmic YAP ratio of NC and KO with 24-h PM2.5 exposure (scale bar, 75 μm) ($n = 3$ biological replicates). The white arrow indicates cytoplasmic YAP, the yellow arrow indicates nuclear YAP. (E) Protein expression ratio of p-YAP and YAP of rat corneas with 3-week PM2.5 exposure. $n = 3$. (F) Protein level of p-YAP/YAP, ATG5, BECN1, IL-1a, and PAI-2 in NC and KO after 24-h PM2.5 exposure ($n = 3$ biological replicates). (G) Protein level of ATG5, BECN1, IL-1a, and PAI-2 in NC after 24-h PM2.5 exposure accompanied with 1 nM VTP ($n = 3$ biological replicates). (H) Mechanism diagram of PAI-2/myosin/F-actin/YAP-positive feedback loop. Data in (C–G) are graphed as mean ± standard deviation with individual values shown as circles. Statistical analysis was conducted using the unpaired $t$ test in (C–G). The $P$ values are labeled in the figure. Source data are available online for this figure.

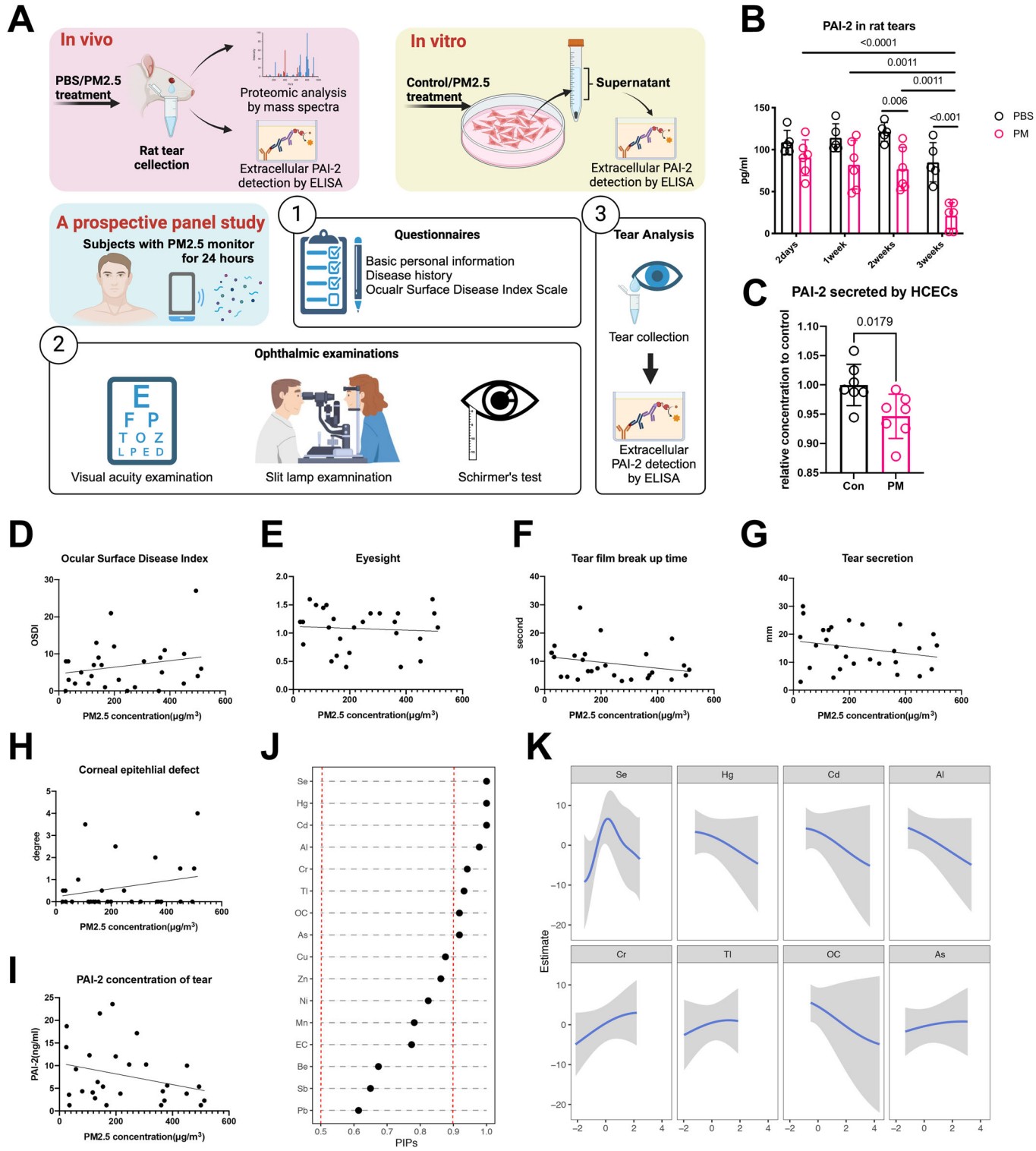

PM2.5 exposure, as evidenced by increased cellular tensile stress and cortex stiffness. Similarly, Ronzier et al demonstrated that macrophages exhibit an early contraction response to LPS/IFNγ, indicating the existence of a mechanical response process to environmental stimuli in cells (Ronzier et al, 2022). Third, our study demonstrated the activation of myosin II and F-actin polymerization in cells subjected to short-term PM2.5 exposure. As a key molecule that affects cell mechanics, the actomyosin system (myosin II and F-actin) plays a role in maintaining cell morphology, cell motility, and regulating the dynamics of the

◄ **Figure 6. Extracellular PAI-2 has potential as a specific diagnostic indicator of PM2.5-induced corneal injury.**

(A) The workflow of extracellular PAI-2 detection in rat tears, HCEC conditioned medium, and human tears. For the details of the panel study in humans, participants completed 24 h of continuous personal air monitoring followed by a questionnaire survey, tear sample collection, and ophthalmic examination. Tear samples were then sent to a laboratory to detect PAI-2 content. $n = 28$. (B) The PAI-2 concentration in rat tears upon PM2.5 exposure for different times was detected via ELISA. $n = 5$ (PBS group), 6 (PM2.5 group). (C) The PAI-2 level in HCEC conditioned medium upon 24-h PM2.5 exposure was detected via ELISA ($n = 7$ biological replicates). (D–I) Spearman analysis was performed to separately define the correlation between PM2.5 concentration and ocular surface disease index/eyesight/tear film break-up time/tear secretion/corneal epithelial defect/PAI-2 content in tears. $n = 28$. Individual values are shown as scattered dots. (J) PIPs of PM2.5 constituents on the PAI-2 level calculated by the BKMR model. (K) Univariate exposure−response curves for PM2.5 constituents with PIPs above 0.9, while all other PM2.5 elemental constituents were at their median concentrations (shown with credible interval). Data in (B, C) are graphed as mean ± standard deviation with individual values shown as circles. Statistical analysis was conducted using the unpaired $t$ test in (B, C). The $P$ values are labeled in the figure. Source data are available online for this figure.

cytoskeleton within the cell (Rassier and Månsson, 2024). Thus, we have demonstrated at various levels that activation of biomechanical response suggested the disease was progressing to a subclinical stage.

Moreover, how do the biomechanical cues drive corneal disease progression? Our work suggested that mechanical signaling induced autophagy and inflammation through activation of the myosin II/F-actin/YAP mechanotransduction pathway. The activation of myosin II/F-actin/YAP had been mentioned in multiple diseases, such as central nervous system injury and in-stent-restenosis (He et al, 2020; Stern et al, 2021). Nevertheless, the prevailing mechanistic studies of PM2.5-related injury consistently highlighted the suppression of key components within mechanotransduction pathways, including stress fibers and integrins (Song et al, 2022; Yu et al, 2019; Cui et al, 2018). This discrepancy may be attributed to the fact that the mechanism of PM2.5-induced damage varied at different stages. In addition, while most studies currently point to inflammation, autophagy, and oxidative damage as the cause of PM2.5-induced damage (Song et al, 2021), our study points to mechanotransduction pathways that regulate these biological processes upstream. The interconnection between cellular mechanics and autophagy has been extensively investigated. F-actin polymerization plays a role in the synthesis of autophagosomes on the endoplasmic reticulum membrane, while actomyosin-based transport is necessary for the delivery of the selected cargoes and debris from other organelles to the growing phagophore (Kast and Dominguez, 2017). Furthermore, myosin-mediated mechanics have been linked to the activation of inflammation (Atcha et al, 2021), thereby reinforcing our conclusions. Taken together, activation of the myosin II/F-actin/YAP mechanotransduction pathway is crucial in driving the progression of PM2.5-induced corneal disease.

Furthermore, we identified intracellular PAI-2 as a modulator of corneal cellular mechanical cues in the subclinical stage. Previous studies have associated PAI-2 with a variety of cellular functions intracellularly, including tumor metastasis, immune recruitment, and embryo implantation (Vaher et al, 2020). However, no studies have ever suggested a clear relationship with cellular biomechanical cues. Our model showed that knockout of PAI-2 led to suppressed cellular tensile stress and cortical stiffness, and inactivation of myosin II/F-actin/YAP signaling pathway under PM2.5 exposure, indicating that PAI-2 regulates the biomechanical response and drive disease progression through the mechanotransduction pathway. The positive feedback between YAP and PAI-2 is noteworthy. YAP is a co-transcription factor that functions in the nucleus in conjunction with TEAD1 (Panciera et al, 2017). However, the mRNA level of PAI-2 was not suppressed after YAP-TEAD1

inhibition. Ronzier et al reported that blocking the initial cellular contraction by blebb (which can also inhibit YAP) could lower iNOS protein through post-transcriptional regulation in macrophages (Ronzier et al, 2022). We postulated that the YAP-TEAD1 interaction might post-transcriptionally regulate PAI-2 as well. PAI-2 was associated with autophagy in the previous studies (Lyu et al, 2020; Chuang et al, 2013). Chuang et al demonstrated that Toll-like receptor 2/4 induced the expression of PAI-2, which subsequently stabilized BECN1 to accelerate autophagy (Chuang et al, 2013). Our study provided a novel perspective that PAI-2 regulated autophagy by the myosin/F-actin/YAP pathway. The PAI-2/myosin II/F-actin/YAP-positive feedback loop led to the accumulation of autophagy and inflammation, which forms a vicious circle, ultimately resulting in corneal epithelial damage.

In addition, we observed an intriguing PAR structure, which was also regulated by PAI-2 and F-actin polymerization. PAR has been demonstrated to be involved in nuclear movement and positioning, as well as regulating nuclear shape and mechanotransduction (Khatau et al, 2009; Calero-Cuenca et al, 2018; Maninova et al, 2017). Our study further suggested that PAR may protect cells from the toxicity of PM2.5. STEF colocalizes with nesprin-2 and myosin IIB and serves as a stabilizer of the PAR (Woroniuk et al, 2018). And INF2 was found to be important for PAR assembly under mechanical stimulations (Shao et al, 2015). This study identified PAI-2 as a novel regulator of PAR, which facilitates the PAR/cortical F-actin switch.

For early diagnosis of PM2.5-related corneal diseases, we would like to explore biomarkers from tears. Since tear collection is noninvasive and there are a large number of components in tears secreted by corneal epithelial cells, the tears could be examined to reflect the physiologic status of the cornea in real time. The previous studies were mainly focused on the detection of inflammatory contributors, such as interleukins (Jing et al, 2022) and FA ω-6 (Gutierrez et al, 2019), in tears. Girshevitz et al observed elevated levels of certain trace elements in the tears of subjects exposed to the specific environment (Girshevitz et al, 2022). However, these markers still lack specificity. As for PAI-2, only one research demonstrated a temporary increase in human tears following laser refractive surgery (Csutak et al, 2008). Our results indicated that PM2.5 induced a reduction in extracellular PAI-2 secretion that became more pronounced with increasing injury in vitro, in vivo, and in a human panel study. This study provides the first evidence that extracellular PAI-2 in tears may serve as a specific biomarker for the progression of PM2.5-induced corneal damage, thereby facilitating early diagnosis.

It is worth discussing why intracellular PAI-2 increases while extracellular PAI-2 decreases with PM2.5 exposure. Research by

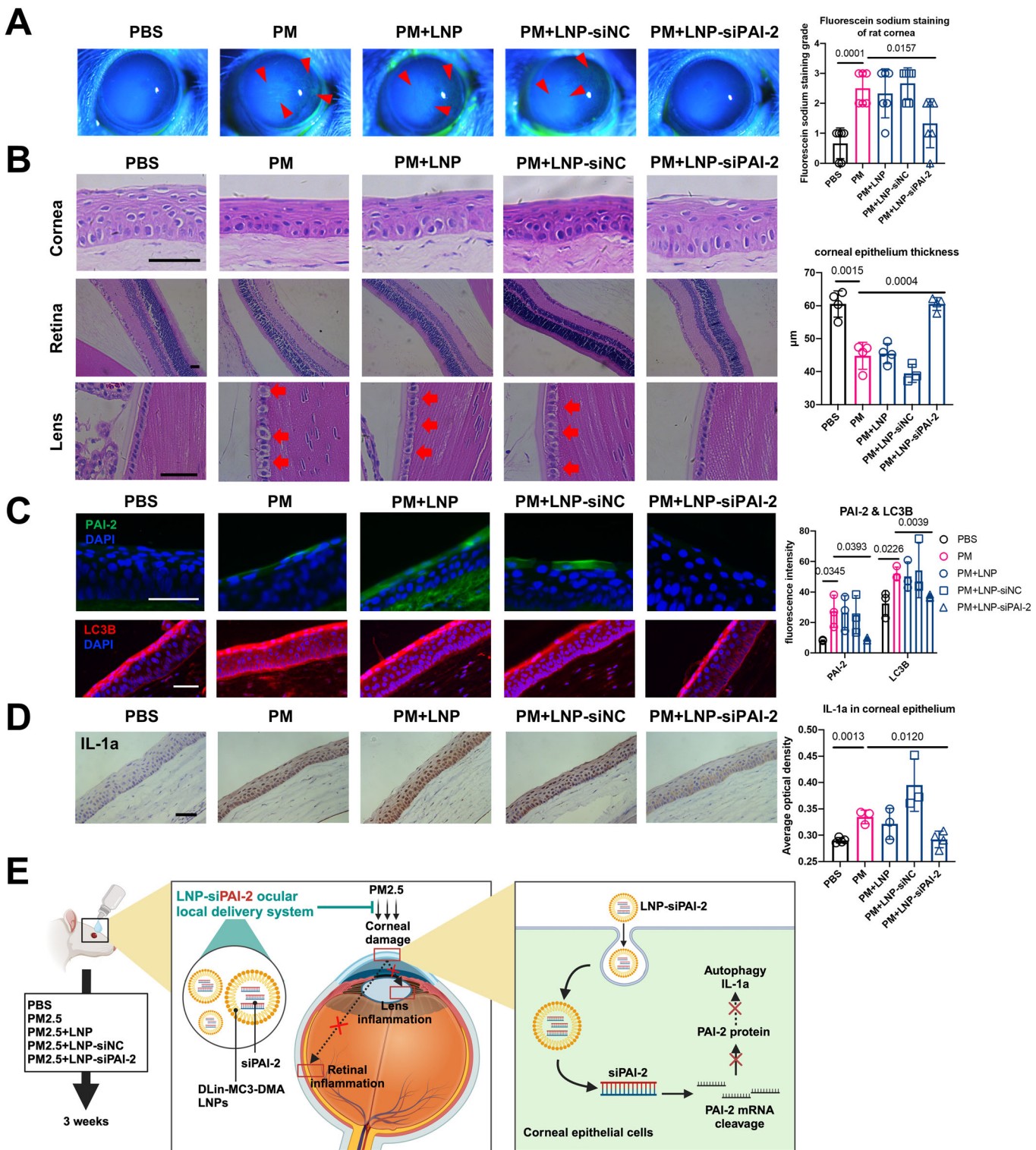

Mikus et al found that in cells with high expression of PAI-2 protein, the protein is prone to spontaneous polymerization during secretion through the loop-sheet polymerization mechanism, resulting in a decrease in its secretion efficiency (Mikus and Ny, 1996). In addition, the study of Wilczynska et al indicated that the redox state of the cell can regulate the polymerization of PAI-2

(Wilczynska et al, 2003). Therefore, the decrease in exocrine PAI-2 levels with PM2.5 exposure in this study may result from the polymerization of PAI-2 protein caused by high intracellular PAI-2 expression.

There is also a lack of effective pharmaceutical agent for the treatment of corneal lesions induced by PM2.5. This study

**Figure 7.  LNP-siPAI-2 ocular local delivery system relieves PM2.5-induced ocular damage.**

(A) The corneal epithelium defect was examined by slit lamp examination using fluorescein sodium staining under cobalt-blue light. The red arrows pointed to the corneal epithelium defect. The fluorescein sodium staining grades were evaluated. $n = 6$. (B) H&E staining showed the morphological changes in the rat cornea, lens, and retina with different treatments (scale bar, 50 μm). The corneal epithelial thickness was measured based on H&E staining. $n = 4$ (PBS), 4 (PM), 4 (PM + LNP), 3 (PM + LNP-siNC), 4 (PM + LNP-siPAI-2). (C) Immunofluorescence staining of rat cornea for PAI-2 and LC3B (scale bar, 50 μm). $n = 3$. The fluorescence intensities of PAI-2 and LC3B were calculated based on immunofluorescence staining. (D) Immunohistochemistry showed the expression level of IL-1a of rat corneas in different groups (scale bar, 50 μm). $n = 4$ (PBS), 3 (PM), 3 (PM + LNP), 3 (PM + LNP-siNC), 4 (PM + LNP-siPAI-2). The average optical density of IL-1a in rat corneal epithelium was measured based on immunohistochemistry. (E) The rat model workflow and therapeutic mechanism diagram of the LNP-siPAI-2 system. LNP-siPAI-2 delivers siPAI-2 to rat corneal epithelial cells, thereby targeting and reducing PAI-2 expression levels and inhibiting autophagy and IL-1a to attenuate corneal epithelial cell damage. At the same time, the reduction in corneal damage reduces the inflammatory response of the intraocular lens and retina, thus realizing the protection of the whole eye by the LNP-siPAI-2 system. Data in (A–D) are graphed as mean ± standard deviation with individual values shown as circles, squares, or triangles. Statistical analysis was conducted using the unpaired $t$ test in (A–D). The $P$ values are labeled in the figure. Source data are available online for this figure.

developed the LNP-siPAI-2 system based on the finding that PAI-2 is an effective therapeutic target. Frequent blinking and tear barriers pose challenges to corneal drug delivery. Although adeno-associated virus has been extensively researched for gene therapy in ocular diseases due to its strong tropism and long-term expression (He et al, 2023), LNPs offer significant advantages. Their superior safety profile, minimal immunogenicity, and ability to deliver larger genetic payloads make LNPs a highly promising alternative for treating a broader range of corneal diseases (Han et al, 2023). Mohanna et al achieved the delivery of LNP-packaged Cas9/gRNA ribonucleoprotein complexes into the mouse cornea by intrastromal injection (Mirjalili Mohanna et al, 2022). In this study, we selected DLin-MC3-DMA LNPs as the carrier to achieve effective delivery of siPAI-2 via noninvasive topical eye drops, significantly expanded the potential applications of LNPs in ocular therapy. DLin-MC3-DMA LNPs had been demonstrated to be an effective carrier for RNA therapy in a variety of organs, with no apparent adverse effects (Zimmermann et al, 2022; Miao et al, 2020). However, there is currently no evidence of its use in ocular delivery. In this study, we confirmed that DLin-MC3-DMA LNP is a type of LNP carrier that could be stably loaded onto the cornea, suggesting a promising future for DLin-MC3-DMA LNPs in ocular gene therapy. Furthermore, given the crucial role of PAI-2 in PM2.5-induced tissue damage in multiple organs, such as the eye, lung, bronchus, and endometrium (Longhin et al, 2018; Park et al, 2021; Lyu et al, 2020; Liu et al, 2015), the LNP-siPAI-2 system developed in this study had the potential to serve as a novel therapeutic strategy for a wide range of pathologies and injuries caused by air pollution.

It is noteworthy that the LNP-siPAI-2 system demonstrated efficacy in protecting the corneal epithelium while concurrently reducing PM2.5-induced damage to the lens and retina. The study by Gu et al posited that inflammation may be a contributing factor to PM2.5-induced intraocular damage (Gu et al, 2022). Our study indicated that the treatment system was effective in reducing ocular inflammation, which may be a mechanism for reducing intraocular damage. This finding suggests that the system may be a valuable tool for the early prevention of PM2.5-induced intraocular damage (e.g., cataract, glaucoma, age-related macular degeneration) (Chua et al, 2019; Li et al, 2023; Liang et al, 2022).

Our study has several limitations. Firstly, the rat model was constructed by dropping PM2.5 eye drops, which cannot fully simulate the real PM2.5 exposure environment. Our model remains valid for studying the direct damage of PM2.5 on the cornea, while there is still a translational gap between topical PM2.5 dosing and ambient exposure. Moreover, the relatively small panel study sizes and limited timeframe do not fully reflect the relationship between PM2.5 and PAI-2 in tears. Further confirmation of our findings is required through the implementation of large multicenter cohort studies.

In conclusion, our study, using a model of PM2.5-induced corneal disease, demonstrated in humans, animals, and cells that changes in corneal biomechanical cues precede the onset of clinical symptoms and are a key factor driving disease progression in the subclinical stage. Intracellular PAI-2 is a key regulator of the cellular mechanical response in corneal epithelial cells, and the PAI-2/myosin II/F-actin/YAP-positive feedback loop enhances autophagy and inflammation in response to PM2.5, thereby triggering the onset of the corneal disease. In light of these findings, extracellular PAI-2 in tear fluid, when considered alongside corneal biomechanical indexes, has the potential to serve as specific indicators for the early diagnosis of PM2.5-associated corneal diseases. Furthermore, the targeting of intracellular PAI-2 by the LNP-siPAI-2 ocular delivery system can be employed for the early intervention of such diseases.

## Methods

**Reagents and tools table**

| Reagent/resource | Reference or source | Identifier or catalog number |
| --- | --- | --- |
| **Experimental models** | | |
| Human corneal epithelial cell lines | ATCC | PCS-700-010 |
| Sprague-Dawley rats (Rattus norvegicus) | Slac Laboratory Animal Co., Ltd. (Shanghai, China) | N/A |
| Human study | This study | N/A |
| **Recombinant DNA** | | |
| pLV-hU6-shSERPINB2-CBh-GFP-IRES-Puro | Genechem | N/A |
| pLV-hU6-shCtrl-CBh-GFP-IRES-Puro | Genechem | N/A |
| pCMV-vector-SV40-Neo | Genechem | N/A |
| pCMV-SERPINB2(human)-SV40-Neo | Genechem | N/A |
| **Antibodies** | | |
| Rabbit anti-SerpinB2 | Abmart | TA5174S |

| Reagent/resource | Reference or source | Identifier or catalog number |
|---|---|---|
| Mouse anti-IL-1a | Abcam | ab239517 |
| Rabbit anti-ATG5 | CST | 12994 |
| Rabbit anti-BECN1 | CST | 3495 |
| Rabbit anti-YAP1 | Proteintech | 13584-1-AP |
| Rabbit anti-YAP1 (phospho S127) | Abcam | ab76252 |
| Mouse anti-GAPDH | Proteintech | 60004-1-Ig |
| Rabbit anti-myosin IIa (phospho Ser1943) | CST | 14611 T |
| Mouse anti-YAP1 | Santa Cruz | sc-101199 |
| Rabbit anti-IL-1a | Proteintech | 16765-1-AP |
| Mouse anti-SerpinB2 | R&D | MAB8550 |
| Rabbit anti-LC3B | Sigma-Aldrich | L7543 |
| Alexa Fluor® 488 Anti-Firefly Luciferase | Abcam | Ab214950 |
| HRP-conjugated anti-rabbit IgG | Multi Sciences | GAR007 |
| HRP-conjugated anti-mouse IgG | Multi Sciences | GAM007 |
| Goat anti-mouse IgG (H + L), Alexa Fluor 488 | Invitrogen | A-11029 |
| Goat anti-rabbit IgG (H + L), Alexa Fluor 555 | Invitrogen | A-21428 |
| **Oligonucleotides and other sequence-based reagents** | | |
| *IL-1a*_forward primer | NM_000575.5 | TGGTAGTAGCAACCAACGGGA |
| *IL-1a*_reverse primer | NM_000575.5 | ACTTTGATTGAGGGCGTCATTC |
| *Becn1*_forward primer | NM_001313998.2 | CCATGCAGGTGAGCTTCGT |
| *Becn1*_reverse primer | NM_001313998.2 | GAATCTGCGAGAGACACCATC |
| *Atg5*_forward primer | NM_004849.4 | AGAAGCTGTTTCGTCCTGTGG |
| *Atg5*_reverse primer | NM_004849.4 | AGGTGTTTCCAACATTGGCTC |
| *Pai-2*_forward primer | NM_001143818.2 | CAGCACCGAAGACCAGATGG |
| *Pai-2*_reverse primer | NM_001143818.2 | CCTGCAAAATCGCATCAGGATAA |
| *Gapdh*_forward primer | NM_002046.7 | GAAGGTGAAGGTCGGAGTC |
| *Gapdh*_reverse primer | NM_002046.7 | GAAGATGGTGATGGGATTTC |
| *Pai-2*_forward primer | NM_021696.2 | CTCAAGCCCATTCTGCAACG |
| *Pai-2*_reverse primer | NM_021696.2 | CCACCATGACCAGTTCTCCC |
| *Gapdh*_forward primer | NM_017008.4 | TGGAGTCTACTGGCGTCTT |
| *Gapdh*_reverse primer | NM_017008.4 | TGTCATATTTCTCGTGGTTCA |
| siRNA | This study | Methods |
| **Chemicals, enzymes, and other reagents** | | |
| Actin-Tracker Red-Rhodamine | Beyotime | C2207S |
| Rat PAI-2 ELISA KIT | mlbio | ml544754V |
| Human PAI-2 ELISA KIT | mlbio | ml544757V |
| Blebbistatin | MedChemExpress | HY-13813 |
| Y-27632 | R&D Systems | 1254 |
| Cytochalasin D | MedChemExpress | HY-N6682 |
| Verteporfin | MedChemExpress | HY-B0146 |
| Jasplakinolide | Sigma-Aldrich | 420127 |
| N, N-methylenebis (acrylamide) | Sigma-Aldrich | 146072 |
| 3-(trimenthoxysilyl) propyl methacrylate | Sigma-Aldrich | M6514 |
| Dichlorodimethylsilane | Sigma-Aldrich | 440272 |

| Reagent/resource | Reference or source | Identifier or catalog number |
|---|---|---|
| Sulfo-SANPAH | Pierce Chemical | 22589 |
| Collagen type I | Corning | 354249 |
| HEPES | Sigma-Aldrich | H4034 |
| DLin-MC3-DMA | MedChemExpress | HY-128788 |
| 1, 2-Distearoyl-sn-glycero-3-phosphocholine (DSPC) | Sigma-Aldrich | P1138 |
| 1,2-dimyristoyl-rac-glycero-3-methoxypolyethylene glycol-2000 (DMG-PEG2000) | Avanti Polar Lipids | 880151 P |
| **Software** | | |
| GraphPad Prism 9 | GraphPad Software | N/A |
| R software | The R Foundation | N/A |
| Image J | NIH | N/A |
| COMSOL | COMSOL AB | N/A |
| **Other** | | |
| PM2.5 | Fu et al, 2017 | N/A |
| NEBNext® UltraTM RNA Library Prep Kit for Illumina® | NEB, Beverly, MA, USA | N/A |

## Ethics

The use of clinical data/specimens from humans adhered to the tenets of the Declaration of Helsinki and the Department of Health and Human Services Belmont Report, and were ethically approved by the Ethical Committee of the Second Affiliated Hospital of Zhejiang University School of Medicine (approval number: 2023-0614) and the Ethical Committee of Zhejiang Provincial Center for Disease Control and Prevention (approval number: AF/SL-02/01.0), and the written informed consent was received prior to participation. All experiments involving animals in this study were approved by the Animal Ethics Committee of the Second Affiliated Hospital of Zhejiang University School of Medicine (approval number: 2023-118).

## Corneal biomechanical data collection and analysis

Corneal biomechanical data were collected in the ophthalmology clinic via the Corvis ST (OCULUS, Germany), and were derived from preoperative examination results of individuals scheduled for myopic refractive surgery. Among them, those with high myopia, a history of other ophthalmic diseases and surgery, systemic diseases, and smoking were excluded. Differences in the age composition ratio and male/female ratio between the groups were insignificant.

Data between December 15, 2022, and March 8, 2023 were selected. Based on air quality monitoring data in Hangzhou City, Zhejiang Province, China, those who came to clinics for preoperative examinations on dates when the average daily PM2.5 concentration was higher than 75 μg/cm³ were included in the high PM2.5 exposure group, and those who came to clinics on dates when the average daily PM2.5 was lower than 35 μg/cm³ were included in the low PM2.5 exposure group. The grouping was based

on a reference to the Chinese Ambient Air Quality Standards (GB3095-2012). In addition, all other pollutants (including $NO_2$, $SO_2$, CO, and $O_3$) should not exceed the minimum thresholds in the above standards.

## PM2.5 extraction and preparation

The PM2.5 extraction process was mentioned in our previous study (Fu et al, 2017). Briefly, ambient PM2.5 samples were collected using a quartz filter, then immersed and ultrasonicated in 75% ethanol for 30 min to obtain a PM2.5 suspension. Then, PM2.5 samples were lyophilized, resuspended, and diluted to 5 mg/mL in phosphate-buffered saline (PBS, 1×) for storage at −80 °C and subsequent experiments. PM2.5 samples were collected in the courtyard of the Zhejiang Center for Disease Control and Prevention in Hangzhou, Zhejiang Province, China, at a height of 1.5 m above the ground, corresponding to the height of the breathing zone. Our previous study described the analysis of organic, ionic, and metal components of lyophilized PM2.5 samples (Fu et al, 2017).

## PM2.5-exposed animal models

For rat models, all Sprague-Dawley rats (8 weeks old, 200–300 g in weight) were purchased from Slac Laboratory Animal Co., Ltd. (Shanghai, China). They were raised in a standard environment at a temperature of 20–24 °C and humidity of 50–60%, with a 12-h light–dark cycle and no food or water restrictions. All rats were randomly divided into two groups. Rats with PM2.5 drops (PM2.5 dissolved in PBS, 1 mg/ml) in both eyes were the PM2.5 group, and rats with PBS drops in both eyes were the control group. Rats were given eye drops four times a day for 2 days (short-term models, $n = 5$ per group) or 21 days (long-term models, $n = 10$ per group), followed by corneal clinical detection and specimen collection.

## Specimen collections

All animals were sacrificed by intraperitoneal injection of an overdose of an anesthetic. Rat eyeballs were removed after 2 or 21 days of exposure, and mouse eyeballs were removed after 21 days of exposure. Some eyeballs were fixed in 4% paraformaldehyde for H&E staining, some were embedded in optimum cutting temperature compound (OCT) for frozen sections and immunofluorescence staining, and the rest were stored at -80 degrees or in liquid nitrogen for subsequent experiments.

## Cell culture and treatment

HCEC lines (PCS-700-010, obtained from ATCC) were cultured in DMEM/F12 (Gibco, CA, USA) with 10% fetal bovine serum (FBS) (Atlanta Biologicals, Norcross, GA) and 1% penicillin/streptomycin (Life Technologies), under 5% $CO_2$ at 37 °C, and were passaged by 0.25% trypsin (Gibco) when reaching 90% confluency. Cell lines were tested for mycoplasma contamination.

HCECs were seeded on six-well, 24-well plates or 6-cm dishes overnight before PM2.5 treatment. HCECs were treated with 100 µg/ml PM2.5 for 3 h (short-term exposure) or 24 h (long-term exposure).

## Measurement of Young's modulus of the cell

AFM measurements were conducted using an Asylum MFP-3D-BIO atomic force microscope equipped with pyramidal-tipped OMCL-TR400PSA cantilevers (spring constant: 0.08 N/m). Prior to stiffness measurements, all cantilever sensitivity calibrations were performed in situ under liquid conditions using the instrument's integrated calibration routines; calibration was repeated independently before each measurement group. By moving the AFM tip at a constant speed, a controllable force was applied to the sample in the z-direction. This would cause an indentation on the soft sample, and the depth of the indentation could be recorded as a function of force, which was the depth-force curve. A consistent experimental protocol was employed across all measurements: the probe loading speed was fixed at 500 nm/s, and the sampling rate was set to 1.7 kHz. Due to the varying mechanical properties of cells, the indentation depth did not correspond directly to the applied displacement. Consequently, the loading displacement was set to 1.2 µm, but for subsequent stiffness calculation, only the segment corresponding to an indentation depth of 600 nm was used for fitting.

For cell stiffness calculations, the approach segment (loading curve) was exclusively used, rather than the withdrawal segment (unloading curve). The indentation depth (δ) into the cell was determined as the difference between the probe position (Z) and the probe deflection (d) [i.e., $δ = Z − d$]. The relationship between force (F) and indentation depth (δ) could be represented by the Hertz model,

$$F = \frac{E \tan(\theta)}{\sqrt{2}(1 - \nu^2)} \delta^2$$

where $E$ is the elastic modulus of the cell, $\theta$ is semi-included angle of the indenting cone, $\nu$ is the Poisson's ratio of the cell, which was assumed to be 0.5. To characterize the stiffness of individual cells, three force-indentation curves were acquired per cell. For each experimental group, measurements were performed on 8–20 distinct cells. Finally, the Young's modulus of the sample could be obtained by using the least mean square fit of the Hertz model.

## Measurement of cellular tensile stress

Polyacrylamide substrates were prepared for the measurement of cellular tensile stress according to existing literature protocols with minor modifications. Firstly, N, N-methylenebis-acrylamide (Sigma) were mixed with Fluorescent beads (Sigma) and the curing agent. The mixture was sandwiched between a glass slide and a coverslip to form a flat gel layer. The glass slide was pretreated with 3-(trimenthoxysilyl) propyl methacrylate (Sigma), and the coverslip was pretreated with dichlorodimethylsilane (Sigma). After the polyacrylamide gel solidified, the glass slide was peeled off. The gel then adhered the coverslip, and the surface of the gel substrate was washed with HEPES (Sigma) and coated with sulfo-SANPAH (Pierce Chemical, Rockford, IL) for 30 min under UV irradiation. Then, a drop of 300 µl of collagen type I (Corning, 0.2 mg/ml) solution was spread on the surface of the gel and left overnight at 4 °C. Before seeding cells, excess collagen solution was removed, the gel was washed with PBS, and sterilized under UV light for 15 min. After sterilization, the cells were inoculated and cultured on the gel according to the experimental requirements.

To obtain the tension in the cell layer, the deformation of the substrate needs to be calculated. We first captured images of fluorescent particles on the surface of the substrate before and after the cells were removed by trypsin in the region of interest. Based on these two images of the fluorescent particles, the surface displacement field of the substrate was calculated using the digital image correlation method, we used the Fourier transform traction cytometry (FTTC), a new and computationally efficient method to compute the traction force between the cell and substrate. Here, the traction force we measured was actually the shear stress exerted by the cell on the substrate. Finally, a two-dimensional model was constructed using the commercial software COMSOL to calculate the stresses within the cell. The previously obtained traction forces were applied on the bottom surface of the cell, by which the in-plane tensile stress in the cell was calculated. All experiments described above were performed in at least three independent replicates. In each replicate, measurements and subsequent calculations were carried out on datasets obtained from 8-15 distinct cells per experimental group.

## Prospective design panel study and tear collection

A total of 28 participants aged 21–68 years were recruited from a community in Jinhua City, Zhejiang Province, China, in March 2023. The main inclusion and exclusion criteria are as follows:

Inclusion criteria:(1) Age ≥18 years; (2) Residence in the study area for more than 6 months with no plan to move out; (3) Able to cooperate in completing questionnaires and examinations; (4) Signed informed consent. Exclusion criteria: (1) History of ophthalmic disease or surgery; (2) The subject or his/her immediate family members within three generations have hereditary ophthalmic diseases; (3) History of other systemic diseases related to ophthalmic diseases, including hypertension, diabetes mellitus, hyperthyroidism, and so on. (4) History of smoking or long-term exposure to smoke and dust. Participants completed 24 h of continuous personal air monitoring followed by a questionnaire survey, tear sample collection, and ophthalmic examination.

PM2.5 exposure flow rate was obtained by a portable personal exposure monitor (PEM-200, BUCK, USA). The sampling filters were removed from the monitor to weigh the total PM2.5 mass, and the PM2.5 concentration was obtained by dividing the total mass by the total flow rate recorded by the monitor. Thermal/optical reflectance was used to examine the organic carbon (OC) and elemental carbon (EC). Inductively coupled plasma/mass spectrometry was used to evaluate the other 14 elemental components.

For tear collection, 40 μl of sterile saline was dropped onto the surface of the subject's eye, which was immediately followed by recovery with a sterile siphon. This approach was used to minimize potential differences in PAI-2 concentrations due to uneven tear volume. Tear samples were stored at -80 °C and subsequently assayed for PAI-2 concentrations by ELISA kits.

## In vivo small interfering RNA (siRNA) delivery via DLin-MC3-DMA LNPs

The LNPs were formulated using the ethanol dilution method. The ethanol phase consisted of DLin-MC3-DMA, cholesterol, 1,2-Distearoyl-sn-glycero-3-phosphocholine (DSPC) and 1,2-dimyristoyl-rac-glycero-3-methoxypolyethylene glycol-2000 (DMG-PEG2000). The aqueous phase was composed of siRNA diluted in sodium citrate buffer (10 mM, pH 4.0). DLin-MC3-DMA LNPs were formulated with a DLin-MC3-DMA/DSPC/cholesterol/DMG-PEG2000 molar ratio of 50/10/38.5/1.5. These two solutions were quickly mixed with a pipette at an ethanol to aqueous volume ratio of 1/3. After mixing, the LNPs were incubated at room temperature for 15 min. These LNPs were encapsulated with 5 μg siRNA. The sequence of siPAI-2: sense (5'-3') GAUACAUAAAGGACCU GAA(dT)(dT), antisense (5'-3') UUCAGGUCCUUUAUGUA UC(dT)(dT); The sequence of siNC: sense (5'-3') UUCUCCGAC AGUGUCACGU(dT)(dT), antisense (5'-3') ACGUGACACUGU CGGAGAA(dT)(dT). For in vivo experiments, LNPs required further dialysis against 1× PBS buffer in Pur-A-Lyzer chambers for 2 h. LNPs were used concurrently with PM2.5 in rat eye drops, 5 μl per eye, four times per day, and rat ocular surface health was examined after 3 weeks.

## Ocular surface fluorescein sodium staining

The rat corneal epithelium defect was detected by fluorescein sodium strips (Meizilin) after exposure for 3 weeks in rats. Detailed procedures were mentioned in our previous study (Lyu et al, 2020). In brief, PBS-moistened strips were placed in the lower lid conjunctival sac of rats after anesthesia. The eyes of the rats were then gently opened and closed several times, and the staining of the ocular surface was observed by a slit lamp under cobalt-blue light. To quantify the extent of damage, we rated corneal damage based on fluorescein sodium staining of rat corneas, using the following criteria: no staining = 0; 1–30 punctate stains = 1; >30 punctate stains but no fusion = 2; fusion of punctate stains = 3.

## Measurement of rat corneal thickness

Rat corneal thickness was measured by anterior segment-optical coherence tomography (AS-OCT). After anesthesia, rat eyes were gently opened and placed in front of the acquisition lens, then rat anterior segment pictures were obtained by video mode, and the corneal thickness was measured manually in the system.

## Corneal impression cytology

After exposure for 2 days, impression cytology was performed on the rat cornea to detect the shape of corneal epithelial cells. Corneal epithelial cells were collected with a 45-μm cellulose acetate film previously cut into $0.5 \text{ cm}^2$ and kept in contact with each eye for 10 s, followed by fixation in 4% paraformaldehyde for 10 min. After that, the film was stained with Periodic acid–Schiff (PAS) and hematoxylin. Finally, cells on the film were observed under a Leica DM3000 white light microscope (Leica, Wetzlar, Germany). Cellular area and perimeter were determined using Image J, and the cellular circularity was calculated using the following equation:

$$Cellular\ circularity = \frac{4\pi\ area(\mu m^2)}{perimeter^2(\mu m^2)}$$

## Measurement of rat corneal hydration

After the animals were executed, fresh corneas were detached from the eyeballs and weighed immediately for wet weight, and the corneal tissue was dried at 4 °C and weighed for dry weight. The

hydration rate was calculated using the following formula:

$$Hydration\ rate = \frac{Corneal\ wet\ weight\ (g) - Corneal\ dry\ weight\ (g)}{Corneal\ wet\ weight\ (g)} \times 100\%$$

## Cell viability detection

Cells were seeded at a density of 5000 cells/well on 96-well plates overnight before PM2.5 treatment. Then, the cells were treated with 100 μg/ml PM2.5. After 24 h, cell viability was detected by CCK8 Kit (Vazyme, Nanjing, China).

## RNA knockout

The specific lentiviral shRNA was purchased from Genechem (Shanghai, China). Lentiviral vectors GV493 (hU6 - MCS - CBh - gcGFP - IRES - puromycin) were cloned with the target sequence of PAI-2, 5'- gcagatccagaagggtagta-3', which were referred to as shPAI-2. HCECs were infected with lentivirus expressing shPAI-2 (referred to as KO) or empty vector (referred to as NC). According to the manufacturer's instructions, briefly, HCECs were infected with lentivirus overnight and then incubated in a fresh medium for an additional 24 h. Successfully infected cells were screened in a medium containing 0.5 μg/mL puromycin, and monoclonal colonies were collected to confirm the reduction in PAI-2 expression at the RNA level by qRT-PCR.

## PAI-2 overexpression

To achieve PAI-2 overexpression, HCECs were transfected with a SERPINB2-expressing vector (Genechem, Shanghai, China) using Lipofectamine 3000 reagent (Invitrogen) following the supplier's instructions. In brief, the plasmid DNA was combined with Lipofectamine 3000 in Opti-MEM medium (Gibco) devoid of FBS and antibiotics, and the entire transfection complex mixture was added to HCECs. After 5 h of transfection, replace the medium with fresh DMEM/F12 medium supplemented with 10% FBS.

## RNA-Seq and data analysis

The total RNA of cells with or without 3-h PM2.5 exposure was isolated and quantified. Then the RNA sample was sent for RNA-seq by Novogene Co., LTD (Beijing, China). In brief, NEBNext® UltraTM RNA Library Prep Kit for Illumina® (NEB, Beverly, MA, USA) was applied to prepare sequencing cDNA libraries. Reads containing ploy-N and low-quality reads were removed from raw data to get clean data. Differential expression analysis was performed via the DESeq2 R package. DEGs were defined with an adjusted $P$ value < 0.05 and | log2(Fold Change) | > 0. ClusterProfiler R package was used for GO enrichment analysis and KEGG pathways. Terms with corrected $P$ value < 0.05 were considered to be significant. All raw and processed RNA sequencing data reported in this manuscript have been deposited in NCBI's Gene Expression Omnibus (GEO; accession number GSE290537).

## Quantitative real-time PCR (qRT-PCR)

The total RNA of cells or tissues was extracted by FlaPure Animal Tissue Total RNA Extraction Kit (Genesand, RE705).

Spectrophotometers (NanoDrop 2000c, Thermo Scientific) were used to detect the quality and concentration of RNA samples. PrimeScript™ RT Master Mix Kit (Takara, RR036) was used for reverse transcription. TB Green® Premix Ex Taq™ II (Takara, RR420) was used for the amplification of synthesized cDNA and detection on an ABI Prism 7500 Fast Sequence Detection System (Thermo Fisher Scientific). All samples were denatured at 95 °C for 30 s, followed by 40 cycles at 95 °C for 3 s, and 60 °C for 30 s. The primers used in this study are listed in the Reagents and tools table.

## Western blotting

The protein of cells or tissues was extracted by the Protein Extraction Kit (Sangon Biotech, Shanghai, China) and quantified by a Pierce BCA Protein Assay Kit (Thermo Fisher Scientific, USA). Protein samples were subjected to SDS-PAGE gel electrophoresis, electrotransferred to polyvinylidene difluoride (PVDF) membranes (Merck Millipore, MA, USA), blocked for 1 h at room temperature using skim milk, and incubated in primary antibody overnight at 4 °C, and then in secondary antibodies at room temperature for 1 h. A BIO-RAD ChemiDoc MP chemiluminescence (Bio-Rad, USA) was used for exposure. Antibodies used are as follows: anti-SerpinB2 (Abmart, 1:1000), anti-IL-1a (Abcam, 1:2000), anti-ATG5 (CST, 1:2000), anti-BECN1 (CST, 1:1000), anti-YAP1 (Proteintech, 1:5000), anti-YAP (phospho S127) (Abcam, 1:5000), anti-GAPDH (Proteintech, 1:10,000).

## Immunofluorescence

For immunofluorescence staining, eyeballs in OCT were sectioned into 8 μm using a Leica CM1950 (Leica, Wetzlar, Germany). Sections were immersed in PBS for 10 min and in 4% paraformaldehyde for 15 min, followed by permeabilization in 0.5% TritonX-100 (Sigma-Aldrich, MO, USA) for 15 min and blockage in 10% goat serum for 1 h at room temperature. The sections were incubated with primary antibodies at 4 °C overnight. The next day, they were washed with PBS and subjected to secondary antibodies at room temperature for 1 h, after being washed by PBS, sections were labeled with 4,6-diamido-2-phenylindole dihydrochloride (DAPI) (Sigma-Aldrich, MO, USA; final concentration of 1 μg/ mL). Imaging and observation were achieved by a Leica DM6B fluorescence microscope (Leica, Wetzlar, Germany). Actin-Tracker Red-Rhodamine (Beyotime, C2207S, 1:500) was used for staining of F-actin. Antibodies used are as follows: anti-myosin IIa (phospho Ser1943) (CST, 1:200), anti-YAP1 (Santa Cruz, 1:100), anti-SerpinB2 (R&D, 1:100), anti-LC3B (Sigma-Aldrich, 1:400), Alexa Fluor® 488 Anti-Firefly Luciferase (Abcam, 1:200).

## Proteomic analysis of rat tear fluid

Rat tear fluid was collected using ultrafiltration to remove impurities and concentrated for protein concentration determination using the BSA method. The obtained proteins were digested into peptides by trypsin, and peptides from different sample sources were labeled with corresponding tandem mass tags. The labeled peptides of different origins were mixed in equal proportions, and then the peptides were divided into several fractions by High Performance Liquid Chromatography. Peptides in each fraction were detected by liquid chromatography-tandem

mass spectrometry (LC-MS/MS) separately, and all the raw data were subjected to the subsequent analytical process.

Analytical processes are as follows: (1) Secondary mass spectrometry data were retrieved using MaxQuant (v1.6.15.0). The database was derived from the Uniprot database of Rattus_norvegicus_10116_proteome (release: 2022-09-30, sequence: 29928), common contaminant libraries were added to the database, and contaminant proteins were deleted when the data were analyzed. The false discovery rate for protein identification and peptide spectrum matches was set to 1%. (2) The DEPs were determined based on adjusted $P$ value < 0.05 and | log2(fold change) | > 0.5. (3) Subcellular localization classification of the DEPs was performed by WolF Psort software. GO and KEGG analysis was conducted by R software using the "clusterProfiler" package. Protein-protein interaction (PPI) analysis was achieved by STRING database (https://cn.string-db.org/). The mass spectrometry proteomics data have been deposited to the ProteomeXchange Consortium via the PRIDE partner repository with the dataset identifier PXD059302.

## Measurement of PAI-2 level

Tears of rats from the PM2.5 or PBS group were collected at 0, 1, 2, and 3 weeks. In all, 10 μl PBS was dropped on the rat's ocular surface and recovered, then the level of rat PAI-2 was measured by Rat PAI-2 ELISA KIT (mlbio, #ml544754V).

The supernatant of HCECs medium was collected after PM2.5 exposure. Human tears were collected from a panel study. The level of human PAI-2 was measured by Human PAI-2 ELISA KIT (mlbio, #ml544757V).

After rat or human tears were collected, they were immediately sent to the laboratory for testing or stored at −80 °C. Before testing, centrifuge the tear sample for 20 min at $1000 \times g$ and take the supernatant for testing. The test strictly followed the steps outlined in the ELISA kit to ensure reproducibility. According to the manual, for sensitivity, the minimum detectable dose of rat PAI-2 is typically less than 1.0 pg/ml, the minimum detectable dose of human PAI-2 is typically less than 0.1 ng/ml. This assay recognizes recombinant and natural rat/human PAI-2. No significant cross-reactivity or interference was observed. The detailed assay procedures are as follows:

a. Prepare all reagents before starting the assay procedure. It is recommended that all Standards and Samples be added in duplicate to the Microtiter plate.

b. Add 50 μl of Standard or Sample to the appropriate wells. Blank well does not add anything.

c. Add 100 μl of Enzymeconjugate to standard wells and sample wells except the blank well, cover with an adhesive strip and incubate for 60 min at 37 °C.

d. Wash the Microtiter Plate four times.

e. Add Substrate A 50 μl and Substrate B 50 μl to each well. Gently mix and incubate for 15 min at 37 °C. Protect from light.

f. Add 50 μl Stop Solution to each well. The color in the wells should change from blue to yellow. If the color in the wells is green or the color change does not appear uniform, gently tap the plate to ensure thorough mixing.

g. Read the Optical Density (O.D.) at 450 nm using a microtiter plate reader within 15 min.

h. Calculate the O.D. value for each standard and sample. All O.D. Values are subtracted by the value of the blank well before result

**The paper explained**

**Problem**

In soft tissues, the process of disease progression is often accompanied by changes in biomechanical properties, which can therefore be applied to the early diagnosis of the disease, monitoring of its course, and prognosis. However, the changes and role of biomechanics in the subclinical stage, where clinical symptoms have not yet appeared, are unclear.

**Results**

Our study focused on the biomechanical cues during corneal disease triggered by PM2.5, a major toxic component of air pollution, and found that changes in corneal biomechanical properties preceded the appearance of clinical symptoms in cellular, animal, and human models. As an important regulatory molecule, PAI-2 can regulate the mechanical response of cells to PM2.5 and drive the damage progression intracellularly, whereas extracellularly, it can combine with the biomechanical properties to reflect the degree of corneal damage caused by PM2.5. The ocular delivery system by targeting intracellular PAI-2 attenuates PM2.5-induced ocular damage.

**Impact**

Our study suggests that biomechanical cues together with key biomarkers may serve as important targets for diagnosis and intervention in the subclinical stage of soft tissue diseases.

interpretation. Construct the standard curve using graph paper or statistical software. To determine the amount in each sample, first locate the O.D. value on the Y-axis and extend a horizontal line to the standard curve. At the point of intersection, draw a vertical line to the $X$ axis and read the corresponding concentration. A new standard curve should be obtained for each experiment.

## Statistics

The mean values between the two groups were compared by the unpaired $t$ test. Spearman's correlation analysis was applied to examine the overall effects of PM2.5 on OSDI, ophthalmic examination, and PAI-2 content in tears. All of these analyses were conducted by GraphPad Prism 9 (GraphPad Software, San Diego, CA). $P < 0.05$ was the threshold for statistical significance. BKMR model was used to estimate PIPs for each PM2.5 constituent and calculate exposure−response curves for constituents with PIPs values above 0.9. All of these analyses were conducted via R software using the 'bkmr' package. All in vivo analysis was done blindly to avoid subjective bias.

## Graphics

Figures 1A, 2F, 3I, 5H, 6A, 7E, TFM schematic diagram in Fig. 2B, AFM schematic diagram in Fig. 2C and synopsis were created with BioRender.com.

## Data availability

The raw sequencing data and processed data included in this study have been deposited in the Gene Expression Omnibus database

with the accession number GSE290537. The mass spectrometry proteomics data have been deposited to the ProteomeXchange Consortium via the PRIDE partner repository with the dataset identifier PXD059302.

The source data of this paper are collected in the following database record: biostudies:S-SCDT-10_1038-S44321-025-00341-0.

## Peer review information

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

## Acknowledgements

The following grants provided financial support for this study: National Natural Science Foundation of China (82471054, 81670833, 81300641, 82271063, 81870641 and 82070939), Key research and development program of Zhejiang Province (2025C02157, 2020C03035, 2019C03091), Fundamental Research Funds of the Central Universities (2019QNA7026), Central Guidance for Local Scientific and Technological Development Funding Projects (2024ZY01057), Medical Science and Technology Project of Zhejiang Province (2022RC122), Zhejiang Provincial Natural Science Foundation of China (LY20H120010), Science and Technology Program of Traditional Chinese Medicine in Zhejiang Province (2023ZL353), Science and Technology Project of Zhejiang Provincial Department of Water Resources (RC2242). We thank Novogene for bulk RNA-seq experiments.

## Author contributions

**Shengjie Hao**: Conceptualization; Formal analysis; Validation; Investigation; Methodology; Writing—original draft; Project administration; Writing—review and editing. **Guangsong Xie**: Formal analysis; Validation; Investigation; Writing—original draft; Project administration; Writing—review and editing. **Dan Li**: Investigation; Writing—original draft; Project administration; Writing—review and editing. **Kexin Su**: Investigation; Writing—original draft; Project administration; Writing—review and editing. **Feiyin Sheng**: Investigation. **Lu Chen**: Formal analysis; Validation; Investigation. **Yuzhou Gu**: Investigation. **Hongying Jin**: Funding acquisition; Methodology. **Yili Xu**: Investigation. **Rongrong Chen**: Investigation. **Zhenwei Qin**: Supervision; Validation. **Dandan Xu**: Funding acquisition; Project administration. **Peiwei Xu**: Project administration. **Lei Zhou**: Software; Supervision. **Na Kong**: Conceptualization. **Hao Ding**: Software; Methodology. **Zhijian Chen**: Conceptualization; Resources; Funding acquisition; Methodology; Project administration; Writing—review and editing. **Shuai Liu**: Conceptualization; Resources; Methodology; Project administration; Writing—review and editing. **Baohua Ji**: Conceptualization; Data curation; Supervision; Validation; Methodology; Writing—review and editing. **Ke Yao**: Conceptualization; Supervision; Funding acquisition; Methodology; Writing—review and editing. **Qiuli Fu**: Conceptualization; Resources; Data curation; Supervision; Funding acquisition; Methodology; Writing—review and editing.

Source data underlying figure panels in this paper may have individual authorship assigned. Where available, figure panel/source data authorship is listed in the following database record: biostudies:S-SCDT-10_1038-S44321-025-00341-0.

## Disclosure and competing interests statement

The authors declare no competing interests.

# Expanded View Figures

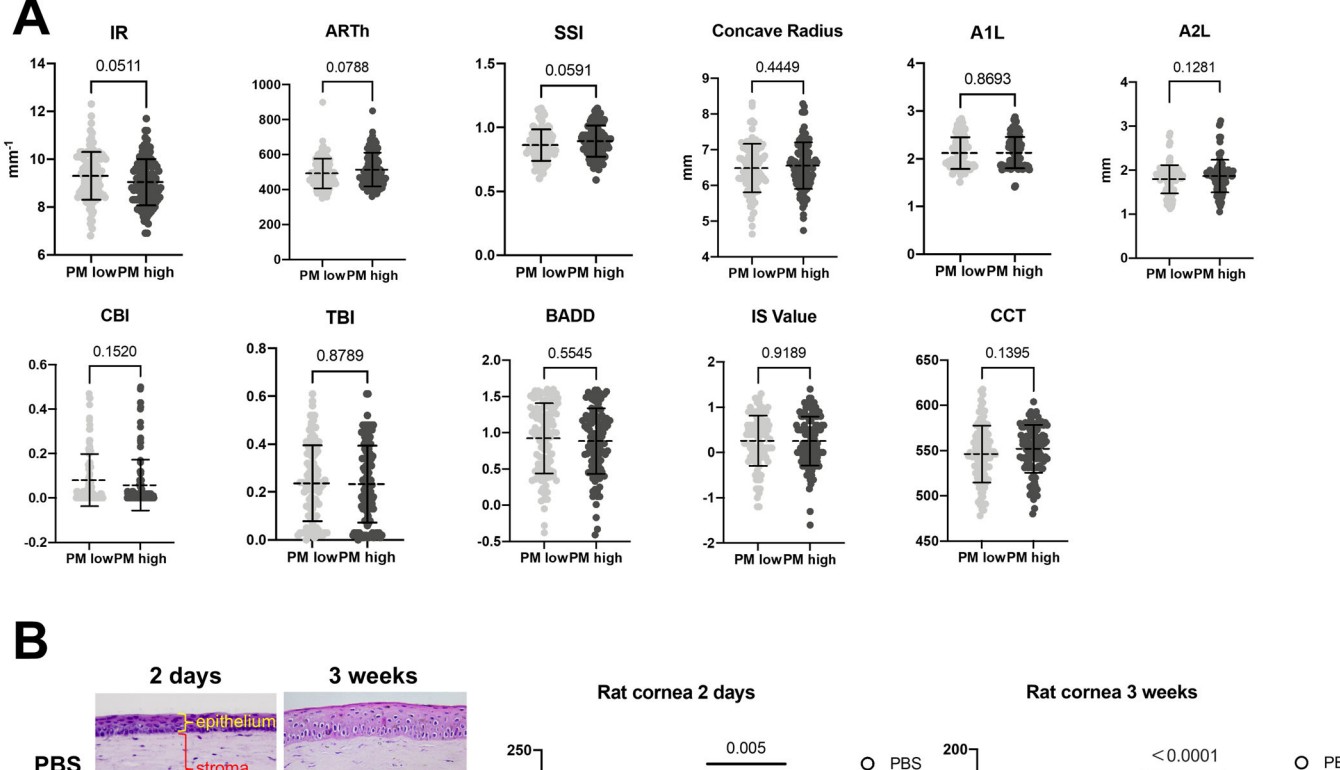

**Figure EV1.  Short-term PM2.5 exposure changes corneal biomechanical cues in humans and rat models.**

(**A**) Corneal biomechanical parameters from Corvis ST in PM2.5 high-exposure group (PM2.5 ≥ 75 μg/cm³, $n = 113$) and PM2.5 low-exposure group (PM2.5 < 35 μg/cm³, $n = 105$). (**B**) H&E staining of rat cornea in two groups and corneal thickness measurement (scale bar, 50 μm). $n = 5$. Data in (**A**, **B**) are graphed as mean ± standard deviation with individual values shown as dots or circles. Statistical analysis was conducted using the unpaired *t* test in (**A**, **B**). The *P* values are labeled in the figure. IR integrated radius, ARTh the Ambrosio relational thickness horizontal profile, SSI stress strain index, A1L first applanation length, A2L second applanation length, CR concave radius, CBI Corvis biomechanical index, TBI tomographic biomechanical index, BADD Belin/Ambrósio Deviation, IS value inferior minus superior value, CCT central corneal thickness.

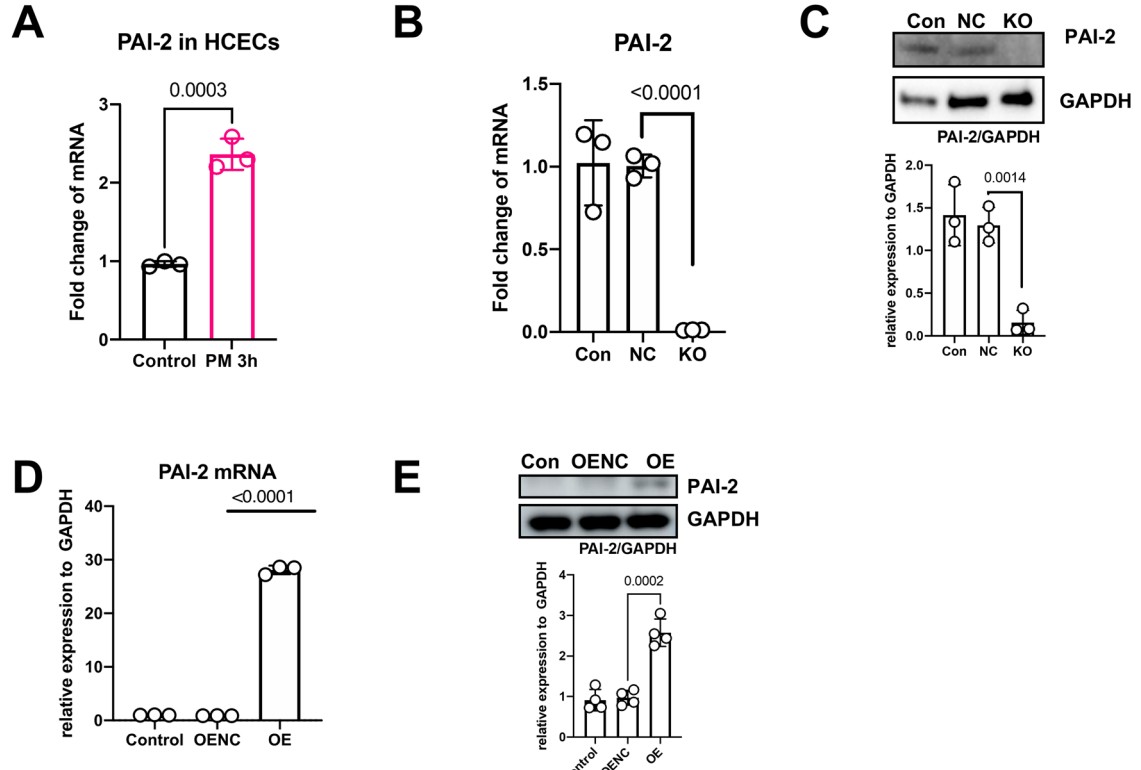

**Figure EV2. The efficiency of PAI-2 knockdown and overexpression in HCECs.**

(**A**) The mRNA level of PAI-2 in HCECs after short-term (3-h) PM2.5 exposure (*n* = 3 biological replicates). (**B**) The PAI-2 mRNA expression level of NC and KO (*n* = 3 biological replicates). (**C**) The PAI-2 protein expression level of NC and KO (*n* = 3 biological replicates). (**D**) The PAI-2 mRNA expression level of OENC and OE (*n* = 3 biological replicates). (**E**) The PAI-2 protein expression level of OENC and OE (*n* = 3 biological replicates). Data in (**A–E**) are graphed as mean ± standard deviation with individual values shown as circles. Statistical analysis was conducted using the unpaired *t* test in (**A–E**). The *P* values are labeled in the figure.

# A

PM2.5 exposure time (hr)

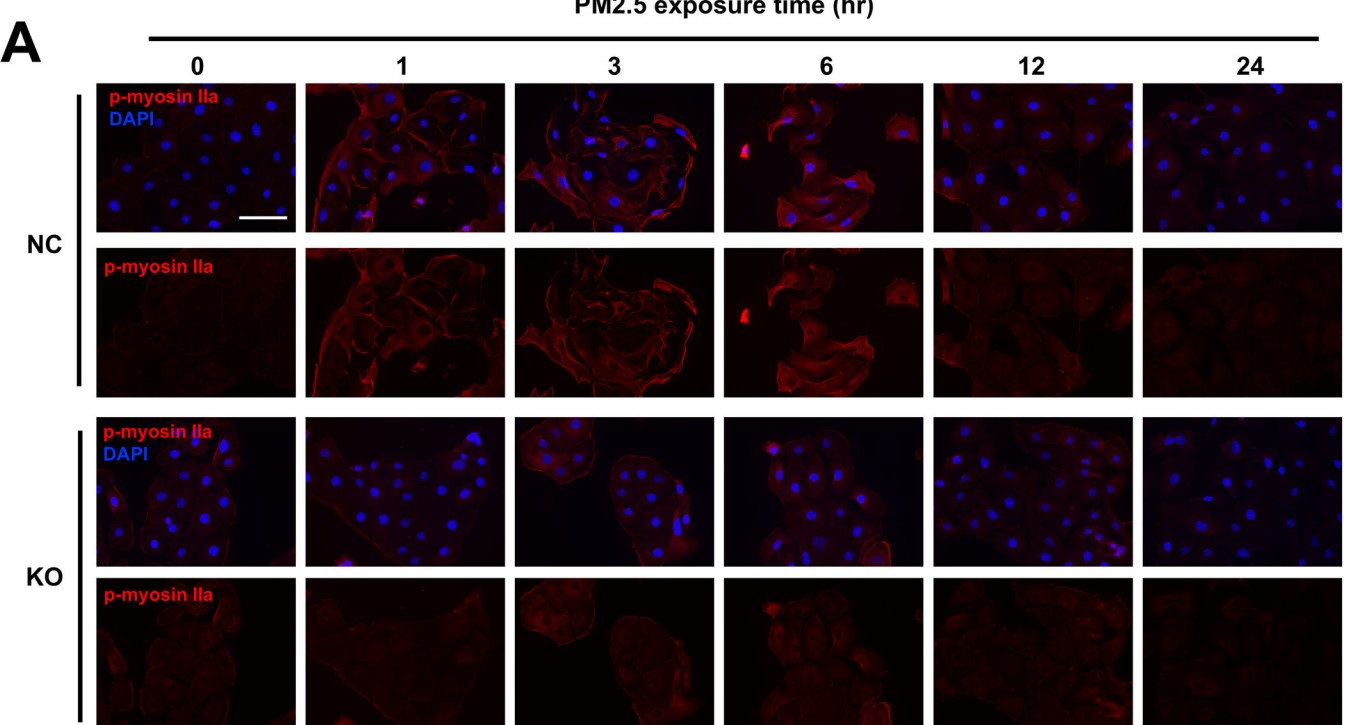

# B

PM2.5 exposure time (hr)

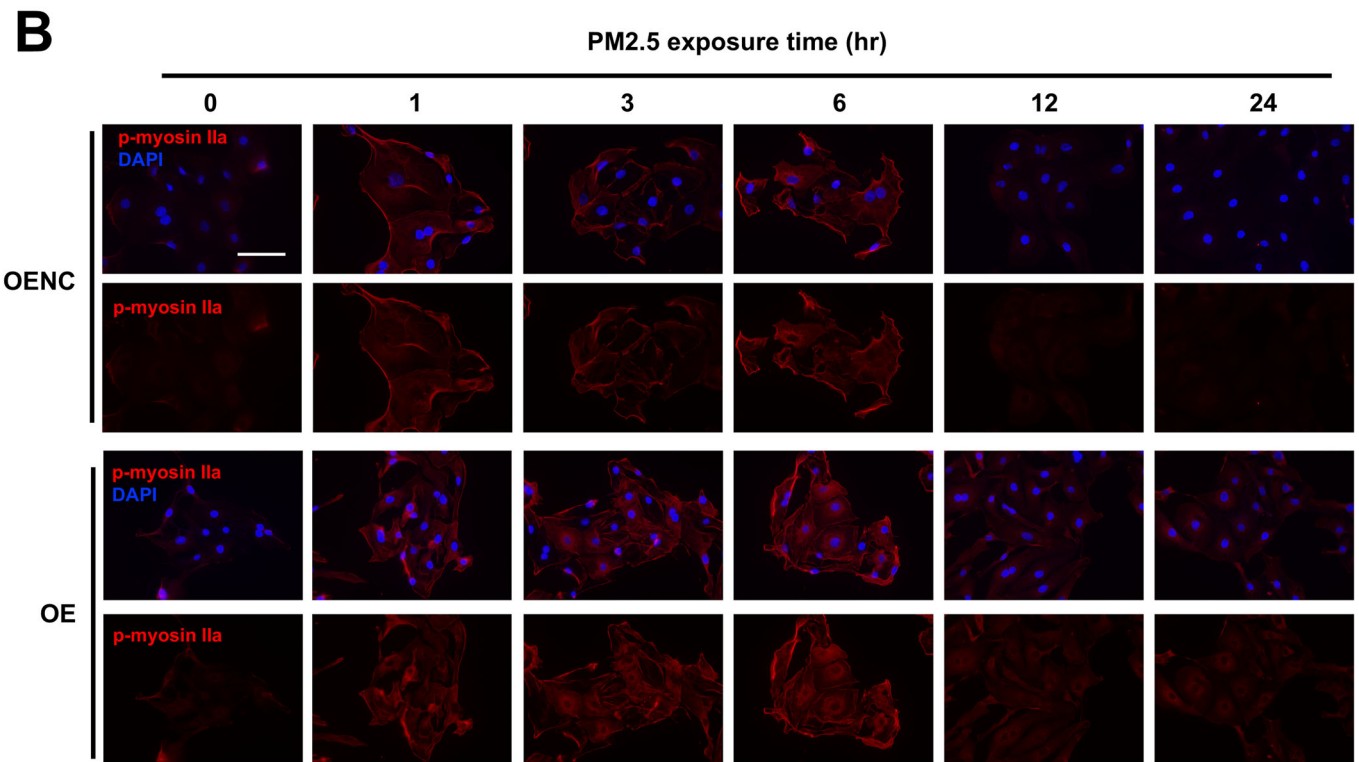

**Figure EV3. Phosphorylated non-muscle myosin IIa staining in NC, KO, OENC and OE with PM2.5 exposure.**

(A) Phosphorylated myosin IIa staining in NC and KO with PM2.5 exposure (scale bar, 75 μm) ($n = 3$ biological replicates). (B) Phosphorylated myosin IIa staining in OENC and OE with PM2.5 exposure (scale bar, 75 μm) ($n = 3$ biological replicates).

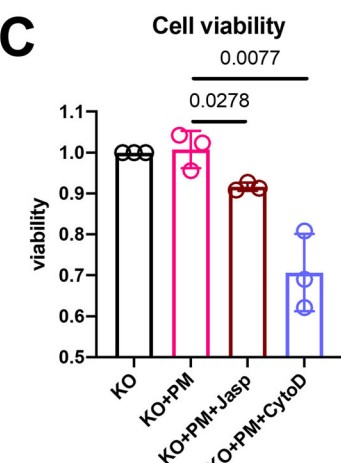

◄ **Figure EV4. The staining of F-actin and YAP in OENC and OE with PM2.5 exposure.**

(**A**) Staining of F-actin in OENC and OE exposed to PM2.5 for 3 h, and F-actin fiber density and PARs were measured (scale bar, 25 μm) ($n = 3$ biological replicates). The white arrow pointed to PAR. (**B**) YAP staining and YAP n/c ratio of OENC and OE with 24-h PM2.5 exposure (scale bar, 75 μm) ($n = 3$ biological replicates). The white arrow indicates cytoplasmic YAP, the yellow arrow indicates nuclear YAP. (**C**) Cellular viability after 24 h of PM2.5 treatment accompanied by PAR being interfered by jasp or cytoD ($n = 3$ biological replicates). Data in (**A–C**) are graphed as mean ± standard deviation with individual values shown as circles. Statistical analysis was conducted using the unpaired $t$ test in (**A–C**). The $P$ values are labeled in the figure.

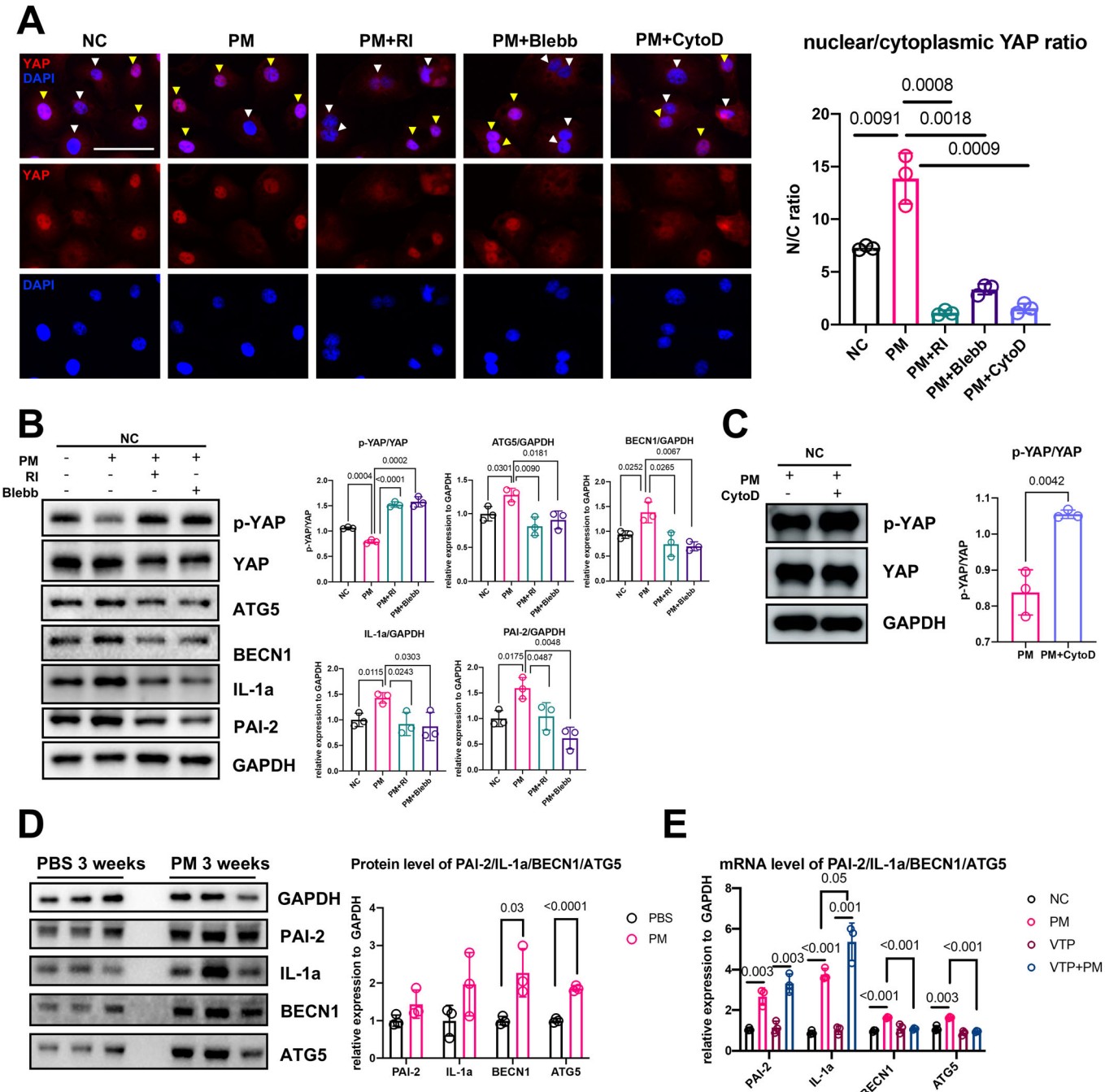

**Figure EV5. YAP nuclear translocation mediates the expression of PAI-2, IL-1a, BECN1 and ATG5.**

(A) Staining of YAP in NC after 24-h PM2.5 treatment with blebb, RI or cytoD and the nuclear/cytoplasmic YAP ratio (scale bar, 75 μm) (*n* = 3 biological replicates). White arrows refer to cytoplasmic YAP, and yellow arrows refer to nuclear YAP. (B) Protein level of p-YAP/YAP, ATG5, BECN1, IL-1a and PAI-2 in NC after 24-h PM2.5 exposure accompanied with blebb or RI (*n* = 3 biological replicates). (C) Protein expression ratio of p-YAP and YAP in NC after 24-h PM2.5 exposure accompanied with cytoD (*n* = 3 biological replicates). (D) The protein expression level of PAI-2, IL-1a, BECN1 and ATG5 in rat cornea after 3-week PM2.5 exposure (*n* = 3 biological replicates). (E) The mRNA expression level of PAI-2, IL-1a, BECN1 and ATG5 in NC after 24-h PM2.5 treatment with VTP (*n* = 3 biological replicates). Data in (A–E) are graphed as mean ± standard deviation with individual values shown as circles. Statistical analysis was conducted using the unpaired *t* test in (A–E). The *P* values are labeled in the figure.

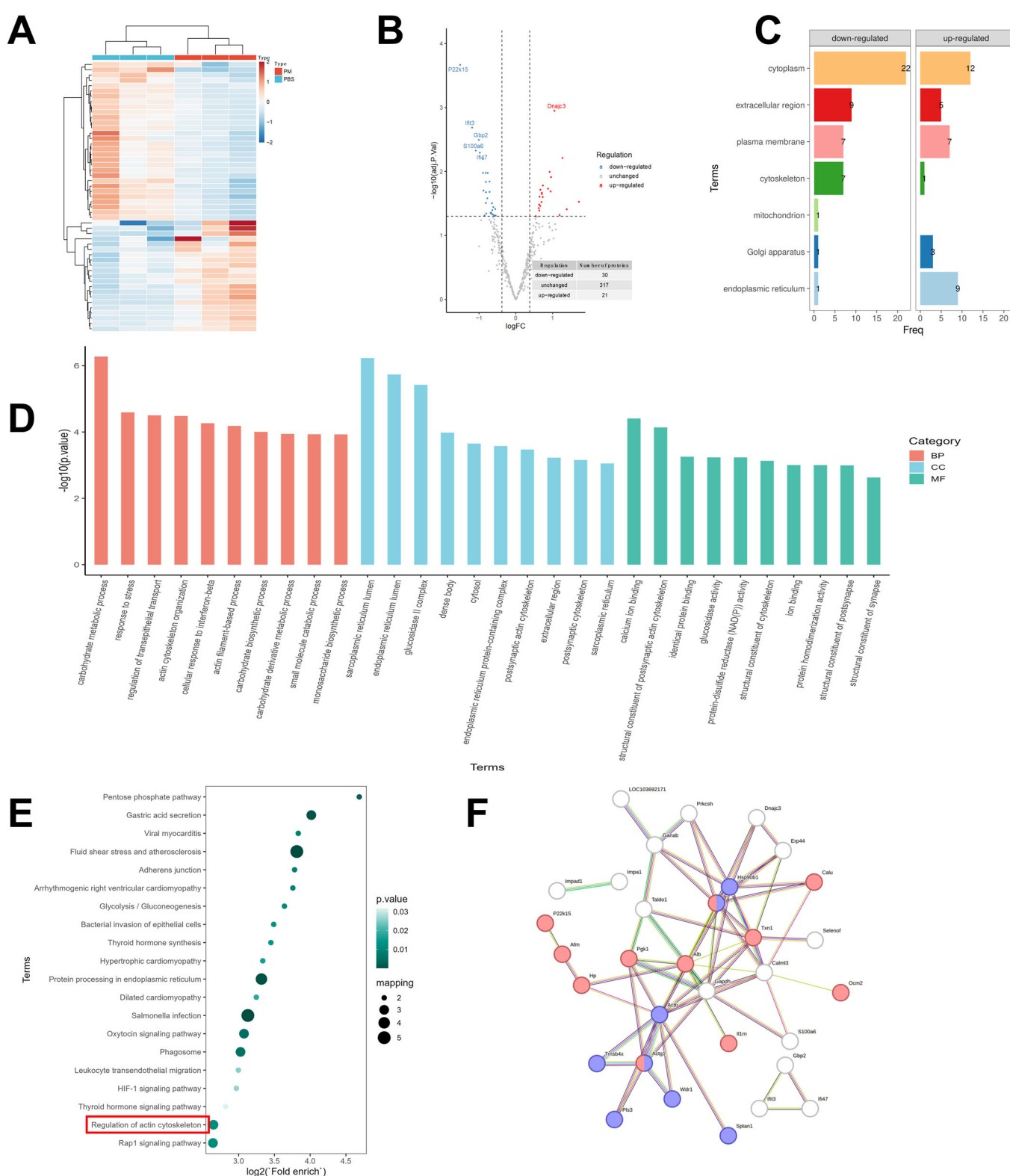

**Figure EV6. Proteomic analysis of tear fluid from rats exposed to PM2.5 for 2 days.**

(A) The heatmap showed the different expression profiling in PBS and PM2.5 groups. $n = 3$. (B) The volcano plot indicated the DEPs between two groups. (C) Subcellular localization classification of DEPs. (D) GO analysis of DEPs including biological process, cellular component, and molecular function. (E) KEGG enrichment analysis of DEPs. (F) PPI analysis of DEPs. The red spots refer to DEPs located in the extracellular region, the purple spots refer to DEPs related to actin organization.

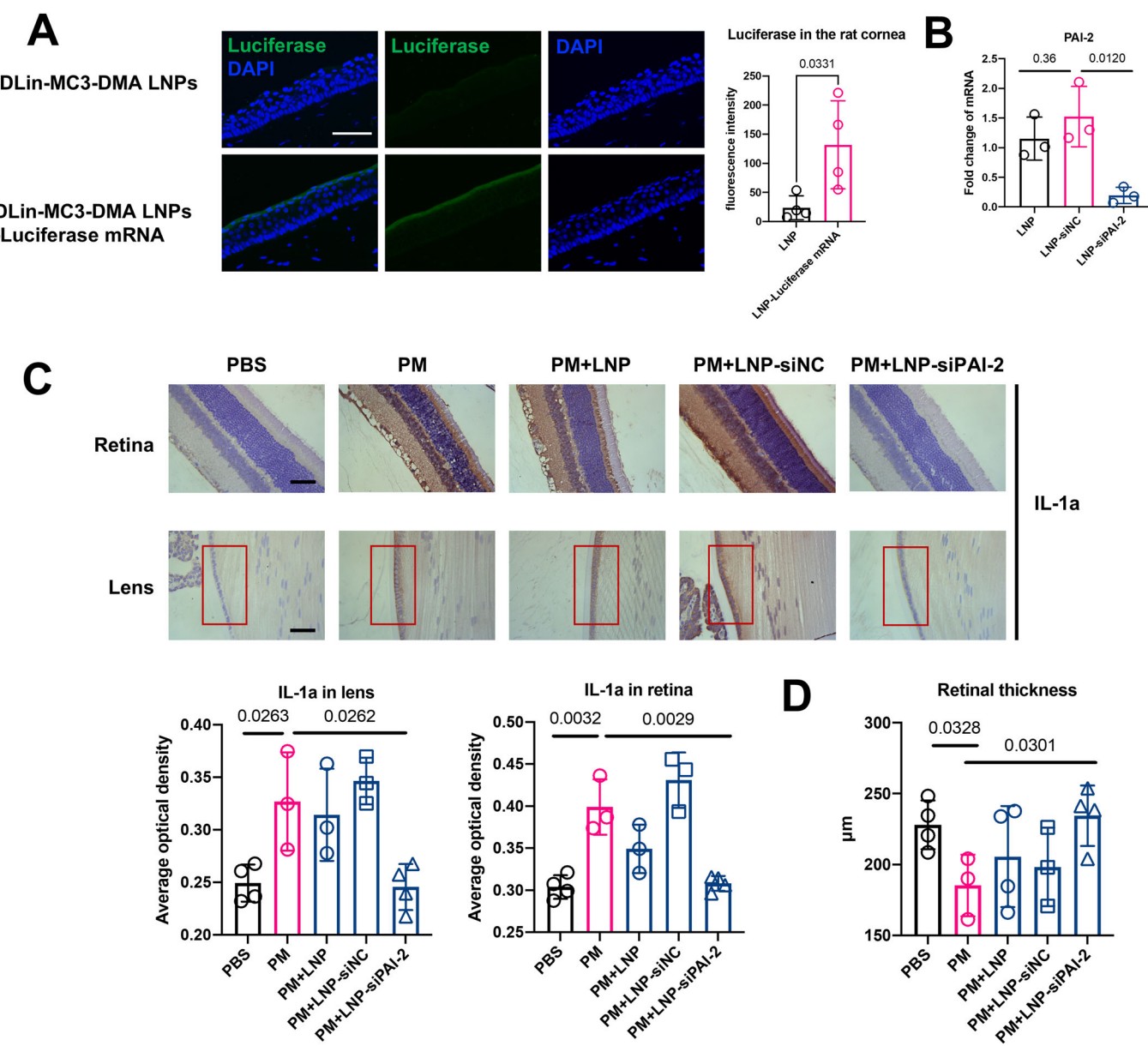

**Figure EV7. LNP (siPAI-2) treatment system relieves PM2.5-induced ocular damage.**

(**A**) DLin-MC3-DMA LNPs can deliver the load (luciferase mRNA) into the corneal epithelium of rats (scale bar, 75 μm). $n = 4$. (**B**) Detection of PAI-2 knockdown efficiency by LNP (siPAI-2) ($n = 3$ biological replicates). (**C**) Immunohistochemistry showed the expression level of IL-1a in rat lens and retinas of different groups (scale bar, 50 μm). $n = 4$ (PBS), 3 (PM), 3 (PM + LNP), 3 (PM + LNP-siNC), 4 (PM + LNP-siPAI-2). The red boxed refer to the rat lens epithelial cells. The average optical density of IL-1a in rat lens and retinas of different groups. (**D**) The retinal thickness based on H&E staining in Fig. 7B. $n = 4$ (PBS), 3 (PM), 4 (PM + LNP), 3 (PM + LNP-siNC), 4 (PM + LNP-siPAI-2). Data in (**A–D**) are graphed as mean ± standard deviation with individual values shown as circles, squares, or triangles. Statistical analysis was conducted using the unpaired $t$ test in (**A–D**). The $P$ values are labeled in the figure.

