## [Peer Review File · EMBO Molecular Medicine]

Corneal biomechanical cues mediated by PAI-2 : the origin of PM2.5-induced corneal disease

Shengjie Hao, Guangsong Xie, Dan Li, Kexin Su, Feiyin Sheng, Lu Chen, Yuzhou Gu, Hongying Jin, Yili Xu, Rongrong Chen, Zhenwei Qin, Dandan Xu, Peiwei Xu, Lei Zhou, Na Kong, Hao Ding, Zhijian Chen, Shuai Liu, Baohua Ji, Ke Yao, and Qiuli Fu

Corresponding authors: Ke Yao (xlren@zju.edu.cn) , Zhijian Chen (zhjchen@cdc.zj.cn), Qiuli Fu (2313009@zju.edu.cn), Baohua Ji (bhji@zju.edu.cn), Shuai Liu (shuailiu@zju.edu.cn)

Review Timeline:

Submission Date:	1st Mar 25
Editorial Decision:	24th Apr 25
Revision Received:	28th Sep 25
Editorial Decision:	21st Oct 25
Revision Received:	23rd Oct 25
Accepted:	31st Oct 25

Editor: Lise Roth

Transaction Report:

24th Apr 2025

Dear Prof. Fu,

Thank you for submitting your manuscript to EMBO Molecular Medicine and please accept my apologies for the delay in getting back to you. As explained in an earlier correspondence, we had initially only managed to secure 2 reviewers, 1 of whom did not provide a report for several weeks. I have therefore contacted 2 more referees. We have now received their reports as well as the 2 original reports.

As you will see below, the reviewers recognize the potential interest of the findings but also raise several important concerns including - but not limited to - lack of direct evidence of a causal relationship between PAI-2 and transient contractility, adequacy of tear model vs. atmospheric exposure, lack of quantitative measurements, limited human study (n, duration, covariates).

If you feel that you have satisfactorily addressed the points raised by the reviewers, you may submit a revised version of your manuscript. Please include a covering letter detailing how you have addressed each of the points raised by the reviewers. A revised manuscript will be reviewed again, and we cannot guarantee at this stage that the outcome will be favorable. Revising the manuscript according to the reviewers' recommendations appears to require a lot of additional work and experimentation, and given the potential interest of your findings, we are prepared to extend the deadline to 6 months.

If you would like to discuss further the points raised by the referees or your revision plan, I am available to do so via email or video. Let me know if you are interested in this option.

EMBO Molecular Medicine encourages a single round of revision only and therefore, acceptance or rejection of the manuscript will depend on the completeness of your responses included in the next, final version of the manuscript. For this reason, and to save you from any frustrations in the end, I would strongly advise against returning an incomplete revision. Should you find that the requested revisions are not feasible within the constraints outlined here and prefer, therefore, to submit your paper elsewhere, we would welcome a message to this effect.

4) A .docx formatted letter INCLUDING the reviewers' reports and your detailed point-by-point responses to their comments. As part of the EMBO Press transparent editorial process, the point-by-point response is part of the Review Process File (RPF), which will be published alongside your paper.

5) A complete author checklist, which you can download from our author guidelines (<https://www.embopress.org/page/journal/17574684/authorguide#submissionofrevisions>). Please insert information in the checklist that is also reflected in the manuscript. The completed author checklist will also be part of the RPF.

6) All Materials and Methods need to be described in the main text using our 'Structured Methods' format. According to this format, the Methods section includes a Reagents and Tools Table (listing key reagents, experimental models, software and relevant equipment and including their sources and relevant identifiers) followed by a Methods and Protocols section describing the methods, ideally using a step-by-step protocol format. The aim is to facilitate adoption of the methodologies across labs. Please download and fill our Reagents and Tools Table template (.docx), which you can find in our author guidelines: <https://www.embopress.org/page/journal/14693178/authorguide#structuredmethods>.

7) Please note that all corresponding authors are required to supply an ORCID ID for their name upon submission of a revised manuscript.

8) It is mandatory to include a 'Data Availability' section after the Materials and Methods. Before submitting your revision, primary datasets produced in this study need to be deposited in an appropriate public database, and the accession numbers and database listed under 'Data Availability'. Please remember to provide a reviewer password if the datasets are not yet public (see <https://www.embopress.org/page/journal/17574684/authorguide#dataavailability>).

9) For data quantification: please specify the name of the statistical test used to generate error bars and P values, the number (n) of independent experiments (specify technical or biological replicates) underlying each data point and the test used to calculate p-values in each figure legend. The figure legends should contain a basic description of n, P and the test applied. Graphs must include a description of the bars and the error bars (s.d., s.e.m.). Please provide exact p values.

10) Our journal encourages inclusion of *data citations in the reference list* to directly cite datasets that were re-used and obtained from public databases. Data citations in the article text are distinct from normal bibliographical citations and should directly link to the database records from which the data can be accessed. In the main text, data citations are formatted as follows: "Data ref: Smith et al, 2001" or "Data ref: NCBI Sequence Read Archive PRJNA342805, 2017". In the Reference list, data citations must be labeled with "[DATASET]". A data reference must provide the database name, accession number/identifiers and a resolvable link to the landing page from which the data can be accessed at the end of the reference. Further instructions are available at .

11) We replaced Supplementary Information with Expanded View (EV) Figures and Tables that are collapsible/expandable online. EV Figures should be cited as 'Figure EV1, Figure EV2' etc... in the text and their respective legends should be included in the main text after the legends of regular figures.

12) The paper explained: EMBO Molecular Medicine articles are accompanied by a summary of the articles to emphasize the major findings in the paper and their medical implications for the non-specialist reader. Please provide a draft summary of your article highlighting

13) Author contributions: CRedit has replaced the traditional author contributions section because it offers a systematic machine readable author contributions format that allows for more effective research assessment. Please remove the Authors Contributions from the manuscript and use the free text boxes beneath each contributing author's name in our system to add specific details on the author's contribution. More information is available in our guide to authors.

Please also suggest a visual abstract to illustrate your article as a PNG file 550 px wide x 300-600 px high. A cropped portion of this image will serve as thumbnail for the table of content on our webpage.

16) As part of the EMBO Publications transparent editorial process initiative (see our Editorial at <http://embomolmed.embopress.org/content/2/9/329>), EMBO Molecular Medicine will publish online a Review Process File (RPF) to accompany accepted manuscripts.

In the event of acceptance, this file will be published in conjunction with your paper and will include the anonymous referee reports, your point-by-point response and all pertinent correspondence relating to the manuscript. Let us know whether you agree with the publication of the RPF and as here, if you want to remove or not any figures from it prior to publication. Please note that the Authors checklist will be published at the end of the RPF.

EMBO Molecular Medicine has a "scooping protection" policy, whereby similar findings that are published by others during review or revision are not a criterion for rejection. Should you decide to submit a revised version, I do ask that you get in touch after six months if you have not completed it, to update us on the status.

I look forward to receiving your revised manuscript.

Yours sincerely,

Lise Roth

**** Reviewer's comments ****

Referee #1 (Comments on Novelty/Model System for Author):

Robust methodology and medical impact

Referee #1 (Remarks for Author):

The manuscript entitled "Corneal biomechanical cues mediated by PAI 2 the origin of PM2.5 induced corneal disease" by Hao and colleagues was submitted to EMBO Molecular Medicine for evaluation.

This manuscript investigates how fine particulate matter (PM2.5) exposure leads to corneal disease by altering corneal biomechanics before obvious clinical symptoms arise. Using a multi-tiered approach-human clinical data, rat models, and cultured human corneal epithelial cells (HCECs)-the authors find that short-term PM2.5 exposure increases corneal hysteresis, cell stiffness, and tensile stress, highlighting a "subclinical" phase where biomechanical changes are detectable prior to overt pathology. They identify PAI-2 (plasminogen activator inhibitor-2) as a key mediator of this process, showing that its elevated intracellular expression drives a positive feedback loop involving myosin II, F-actin polymerization, and YAP activation, thereby exacerbating autophagy and inflammation. Concurrently, extracellular PAI-2 secreted into tears decreases in correlation with PM2.5 dose, suggesting that tear PAI-2 could serve as a convenient biomarker for early intervention. Finally, the authors demonstrate that delivering siRNA against PAI-2 (via lipid nanoparticles) to the ocular surface alleviates damage in PM2.5-exposed animals, indicating a promising direction for early therapeutic strategies.

The authors focused on Subclinical vs. Clinical Stages: A central concept is to delineate a "subclinical" stage (with primarily biomechanical changes) from the full "clinical" stage (with evident corneal damage). Strategically separating these stages is valuable for a mechanistic understanding of early disease progression.

The Results section is largely well-structured, moving logically from observational findings (short-term PM2.5 exposure in humans) to mechanistic cell and animal studies.

Figures are arranged to show progression from short-term changes (biomechanical) to more extensive cellular and tissue damage. This helps the reader follow the argument that biomechanical changes precede overt pathology.

The manuscript is well built and robust, here are some strengths:

- Innovative Link to Biomechanics: The paper offers a fresh perspective that biomechanical cues precede overt tissue damage, providing an "early alert" concept with potential impact on how ophthalmic pathologies are diagnosed and monitored.
- Robust, Multiscale Approach: The authors' careful alignment of human observational data with rat models and in vitro experiments lends credibility to their mechanistic conclusions.

- Therapeutic Translation: Proposing (and testing) siRNA-laden lipid nanoparticles for topical corneal application showcases direct translational potential.

However, I have noticed some weaknesses:

- Exposure Modeling: More detailed real-world exposure protocols for animals or prospective longer-term human panel studies with repeated measures would strengthen external validity.
- In-Depth Mechanistic Details: While the signaling path PAI-2 → Myosin II → F-actin → YAP is supported, clarifying whether additional upstream signals (e.g., integrin activation) or other co-transcription factors are involved might be enlightening.
- Tear Biomarker Validation: Larger cohorts should be studied to validate tear PAI-2 or other possible biomarkers across diverse demographics and pollution levels.

I would recommend to address these weaknesses, and amend the manuscript according to the following questions:

1. Are the statistical analysis appropriate? The Bayesian Kernel Machine Regression (BKMR) analysis of PM2.5 constituents is advanced and well-motivated. This highlights which components (e.g., metals, organic carbon) most strongly correlate with tear PAI-2 levels. The approach is solid but is carried out on a relatively small sample, which may require caution interpreting the exposure-response curves.
2. Could you provide more detailed criteria used to recruit participants for the short-term exposure study (e.g., exclusion of confounding comorbidities, differences in daily environments, prior eye conditions)?
3. Have you considered or planned any longitudinal measurements (more than a single day of exposure) in human participants to better capture real-world exposure fluctuations?
4. For the animal model, have you contemplated alternative PM2.5 exposure methods (e.g., inhalation chambers) that might replicate real-world exposures more closely?
5. Do you have data showing whether repeated exposure or longer-term (beyond 24 hours) PM2.5 treatment in vitro affects HCECs differently, compared to your short- and medium-term exposures?
6. Could you provide more quantitative metrics (e.g., scoring scales, defect area measurements) for corneal epithelial damage in animal models, in addition to representative images?
7. In the AFM and traction force microscopy (TFM) experiments, could you elaborate on the number of cells measured, indentation protocols, and statistical power to ensure that these biomechanical assays are robust?
8. What role might integrins or focal adhesions (FAK-paxillin) play upstream in your proposed PAI-2/myosin II/F-actin/YAP pathway? Do you see consistent alterations in these adhesion-associated proteins?
9. Do you suspect that PM2.5-induced oxidative stress leads to PAI-2 polymerization or altered secretion? Could you examine redox states or measure polymeric forms of PAI-2?
10. Beyond autophagy and IL-1 α expression, have you explored whether YAP might regulate other downstream genes (e.g., those involved in epithelial barrier function) that could contribute to corneal pathology?
11. Could you clarify the precise time course of PAI-2, myosin II, and YAP activation versus cellular damage? For instance, are there distinct "waves" of activation (1-3 hours, 6-12 hours, etc.)?
12. How do you operationally define "subclinical stage" in the animal and cell experiments? Are there objective markers you use to confirm the absence of overt damage while capturing biomechanical changes?
13. Could you offer details on the reproducibility and sensitivity of your tear PAI-2 detection method (e.g., sample processing, assay validation, potential interfering substances in tears)?
14. Were any other tear proteins or inflammatory cytokines similarly evaluated but not included in the main text? If so, might they corroborate or contrast with PAI-2 findings?
15. Have you explored a dose-response or gradient exposure in the animal or cell models to link PM2.5 concentration with corneal biomechanical and biochemical changes?
16. How long is siRNA effectively delivered to corneal cells via DLin-MC3-DMA LNPs following topical application (e.g., hours, days)? Could you address the duration of knockdown effects?
17. Have you assessed potential off-target or inflammatory effects of LNP-siPAI-2 therapy, especially with repeated instillations over multiple weeks?
18. You note that PM2.5 exposure also affected the lens and retina; do you have mechanistic data (inflammatory or oxidative) clarifying how corneal epithelial damage propagates or correlates with deeper ocular structures?
19. Are there additional autophagy markers (LC3 flux, p62/SQSTM1) or imaging techniques used to confirm autophagic activity, rather than just elevated expression of individual proteins?
20. Did you measure oxidative stress (e.g., ROS production, antioxidant defenses) in corneal cells or tissues under PM2.5 exposure, and how might that tie in with your mechanotransduction findings?

Referee #2 (Comments on Novelty/Model System for Author):

In this new study, the authors report that exposure to fine particulate matter (PM2.5) with a diameter of 2.5 μm alters the mechanical properties of the cornea and corneal cells. However, given their previous research identifying plasminogen activator inhibitor-2 (PAI-2) as a mediator of PM2.5-induced cellular changes, the novelty of the current study is limited, primarily focusing on the observation of a transient contractility response to PM2.5. Furthermore, the mechanistic links between the presented

observations, specifically the causal relationships between PAI-2 and transient contractility, and between contractility and autophagy, are not adequately established, limiting the study's contribution to a descriptive account. The conclusions drawn by the authors are not fully supported by the presented data. While they suggest altered myosin II activity ("indicating PAI-2 regulated myosin II activity"), their analysis is limited to reporting a transient increase in myosin II expression (Figures 2b and 4b). They do not provide direct measurements of myosin activity, such as phosphorylation, which is typically required to substantiate such claims. Similarly, they report changes in mRNA PAI-2 expression following short-term PM2.5 exposure (at 3h), but corresponding protein level data (measured by immunofluorescence at 48 hours for example), are absent. This omission raises concerns regarding PAI-2's role in the observed short-term (3-hour) response. Figure 5b and 5f reveal differences in total YAP levels, which make the interpretation of the experiment very difficult as these differences could lead to a decrease in p-YAP without necessarily reflecting changes in YAP activity. To accurately assess the impact on YAP activity, the authors should analyze YAP nuclear localization (by IF, preferably with higher quality imaging to distinguish between nuclear and cytoplasmic localization). Additionally, the fluorescence imaging lacks adequate resolution and quantification to support robust conclusions regarding subcellular actin localization (figure 4). Overall, a more rigorous methodology, incorporating gold-standard quantitative techniques, is essential to validate these findings and demonstrate a significant impact.

Referee #2 (Remarks for Author):

The authors report that PM2.5 exposure (2.5 μm) induces alterations in corneal mechanical properties and cellular changes prior to symptom onset, evidenced by changes in mechanical properties, myosin expression, and PAI-2 expression. However, the study lacks critical logical connections, particularly regarding the mechanisms by which PAI-2 induces transient changes in myosin expression and the link between contractility and autophagy. Furthermore, inconsistencies exist between the authors' claims and the supporting data. Overall, a more rigorous methodology, incorporating gold-standard quantitative techniques, is essential to validate these findings and demonstrate a significant impact.

The authors claim that PM2.5 exposure affects corneal hysteresis, but corneal hysteresis itself is not assessed. Although several methods are used to evaluate the mechanical properties of cornea (young modulus etc...), corneal hysteresis-reflecting the eye's ability to absorb shock and overall biomechanical characteristics-is not directly tested in this study.

The authors suggest altered myosin II activity ("indicating PAI-2 regulated myosin II activity"), their analysis is limited to reporting a transient increase in myosin II expression (Figures 2b and 4b). They do not provide direct measurements of myosin activity, such as phosphorylation, which is typically required to substantiate such claims.

The authors report changes in mRNA PAI-2 expression following short-term PM2.5 exposure (at 3h), but corresponding protein level data (measured by immunofluorescence at 48 hours for example), are absent. This omission raises concerns regarding PAI-2's role in the observed short-term (3-hour) response.

Figure 5b and 5f reveal differences in total YAP levels, which make the interpretation of the experiment very difficult as these differences could lead to a decrease in p-YAP without necessarily reflecting changes in YAP activity. To accurately assess the impact on YAP activity, the authors should analyze YAP nuclear localization (by IF, preferably with higher quality imaging to distinguish between nuclear and cytoplasmic localization).

Additionally, the fluorescence imaging lacks adequate resolution and quantification to support robust conclusions regarding subcellular actin localization (figure 4).

Minor points:

Line 67: The authors' assertion that this model is 'the best' for studying the relationship between biomechanical cues and disease lacks sufficient substantiation. This claim is subjective and not supported by comparative data or a clear rationale.

Line 255 the results showed an increase in Young's modulus upon 3-h exposure, indicating an improvement in cell cortical stiffness (Fig.2c-2e). What do the authors mean by improvement?

Referee #3 (Comments on Novelty/Model System for Author):

The study combines cell, animal, and human models, enabling in-depth exploration of the mechanisms of PM2.5-induced corneal diseases from different levels.

Referee #3 (Remarks for Author):

Overall Evaluation:

This manuscript by Hao et al. explores the role of corneal biomechanical cues in PM2.5-induced corneal diseases, with a focus on the mediator PAI-2. The study is of great significance as it provides new insights into the early diagnosis and intervention of corneal diseases. However, it also has several areas that need improvement.

Major revisions:

1. After adding CytoD, did the structure of PAR disappear? The authors are recommended to explain the reason more clearly.
2. Overexpress PAI-2 to further investigate whether the role of PAI-2 in the PAI-2/myosin II/F-actin/YAP pathway is the same as that in the knockout experiment.
3. If elements such as Se, Hg, Cd, Al, and OC are removed from PM2.5, will it no longer cause corneal damage?

4. When reading the article, PAI-2 seems more like a target for treating PM2.5-induced corneal damage. However, to prove that it is a specific biomarker for early real-time diagnosis, the authors should conduct more detailed quantitative measurements of PAI-2 in tears at finer time points after PM2.5 exposure.
5. Is PAI-2 related to corneal damage caused by other inappropriate mechanical stimuli, such as wearing contact lenses?

Minor points:

1. The full names of abbreviations such as LNP, siRNA, PM2.5, DSPC, ROCK, etc. should be provided.
2. The coordinates in Figure 2d, 2g, 4c, and 4f are unclear.
3. What are the sequences of the siPAI-2 and si-NC?
4. The following articles are relevant to your research and could be cited:
(1) Xiaoyu He, Yidian Fu, Liang Ma, Yizheng Yao, Shengfang Ge, Zhi Yang, Xianqun Fan. AAV for Gene Therapy in Ocular Diseases: Progress and Prospects. *Research*. 2023;6:0291
(2) B. Kong, C. Qi, H. Wang, T. Kong, Z. Liu, Tissue adhesives for wound closure. *Smart Med*. 2023, 2(1), e20220033.

Referee #4 (Remarks for Author):

In the manuscript titled "Corneal biomechanical cues mediated by PAI-2: the origin of PM2.5-induced corneal disease", Hao et al. investigate the role of biomechanical changes and the serpin protein PAI-2 in the early pathogenesis of PM2.5-induced corneal injury. This study introduced that extracellular PAI-2 in tears may serve as a biomarker for early diagnosis, and proposed an LNP-based siRNA delivery system targeting intracellular PAI-2 as a preventive treatment strategy. However, there are some concerns about this manuscript, as outlined below:

Major concerns:

1. While the rationale was briefly mentioned, the authors should better elucidate the rationale for the PM2.5 eye drop model was chosen over atmospheric exposure. Eye drop administration may induce more direct and acute corneal injury, which could not fully reflect real-world chronic exposure dynamics. Clarification on this choice is essential, as environmental chamber-based models may offer greater physiological relevance. More data and a comprehensive discussion are warranted to support the translational applicability of the model and to better explore the toxic effects of PM2.5 in corneal disease.
2. PM2.5 increases the risk of diseases such as keratitis and dry eye, yet these conditions differ in onset and progression. The authors should clarify the rationale for classifying observed changes as subclinical. Specifically, the different PM2.5 exposure durations in human, animal, and cell models raise questions about whether the findings truly reflect the subclinical biomechanical stage rather than the early clinical stage. For example, increased corneal thickness in rats after two days of PM2.5 eye drops may already represent disease onset (Figure 1e). Greater clarity is needed to distinguish subclinical from clinical stage responses across models.
3. The authors should consider the potential impact of participant age range (21-68 years) on the study outcomes. Age may be a factor influencing corneal biomechanical properties and tear film composition, including secretion volume and protein content. These changes alter tissue responsiveness to PM2.5 exposure and affect baseline or induced levels of tear PAI-2. It is recommended that the authors include age as a covariate in their statistical models or conduct age-stratified analyses to determine whether age modulates the ocular response to PM2.5. Addressing this would enhance the robustness and generalizability of the findings.
4. The authors proposed a positive feedback loop wherein PAI-2 regulates YAP activation via myosin II and F-actin remodeling. While the data demonstrated that myosin II inhibitors (RI or blebb) altered YAP phosphorylation and reduced nuclear localization (Figure 5a), it would be beneficial to acknowledge that this evidence primarily supports an indirect regulatory relationship between F-actin and YAP activity. The experimental approach did not establish direct physical or biochemical interactions between these components. It suggests clarifying YAP modulation by F-actin likely occurs through alterations in cytoskeletal tension and mechanical properties rather than through direct molecular association (Figure 5h).

Minor concerns:

1. In the Results section, please ensure that each result title is a conclusive statement based on the study findings. Avoid generic titles such as "PAI-2/myosin II/F-actin/YAP positive feedback loop regulates autophagy" or "PAI-2 exocytosis is affected by PM2.5." Additionally, each result should have only one title. For example, under the heading "Corneal biomechanical changes precede clinical symptoms under PM2.5 exposure," there are three subtitles. Please revise your section titles accordingly.
2. Some figures should be reorganized, combining those with a similar experimental purpose. For example, Figure 1e-f measured rat corneal thickness and can be merged into a single figure. In addition, Figure 2a-b show immunofluorescence staining for nonmuscle myosin II. Figures 2j-k are a staining image and its corresponding quantification, not two separate experiments. Please reorganize similar experimental results into a single figure.
3. In Figure 3b, the mRNA level of PAI-2 in rat corneas after short-term PM2.5 exposure. Please specify the exact exposure duration used in the short-term PM2.5 exposure. Additionally, please verify whether the figure legend is accurate and complete for Figure 3, and revise or remove any descriptions that are missing or not applicable.
4. Some results that are less connected to the main findings could be moved to the supplementary results section. For example, the knockout efficiency of PAI-2 shown in Figure 3d could be combined with Figures S3a and S3b.
5. Please check the use of blebb or RI in Figure 5a, whether they were used in combination or applied individually in the

experiment.

6. The graphical abstract should clearly highlight findings derived from actual experimental results. Please explain the relationship between biomechanical response and PM2.5 exposure as shown in Figure 2i, particularly how it reflects the disease progression curve and the statistical methods used. If no such analysis was conducted, please clearly indicate that this is a conceptual illustration.

Dear reviewers,

Thank you very much for your comments and professional advice. These opinions help to improve academic rigor of our article. Based on your suggestion and request, we have made corrected modifications on the revised manuscript. We would like to show the details as follows:

Reviewer 1#

1. Are the statistical analysis appropriate? The Bayesian Kernel Machine Regression (BKMR) analysis of PM_{2.5} constituents is advanced and well-motivated. This highlights which components (e.g., metals, organic carbon) most strongly correlate with tear PAI-2 levels. The approach is solid but is carried out on a relatively small sample, which may require caution interpreting the exposure-response curves.

Response: We appreciate your thoughtful assessment of our statistical approach. We acknowledge the limitation of our modest sample size, which may affect the precision of exposure-response estimates. Our current study did not delve deeply into the effects of various components in PM_{2.5} on tear PAI-2, but only preliminarily screened out potential components that may be related to tear PAI-2. We agree that larger cohorts are needed to refine these associations and have highlighted this as a priority for future work. Therefore, we have revised the wording in the corresponding paragraph to make it more rigorous: ...Notably, PAI-2 levels were inversely associated with Hg, Cd, Al, and organic carbon (OC), suggesting that these constituents may be key contributors to reduced PAI-2 in tears. However, we interpret these univariate findings with caution given the limited sample size, further validation in larger cohorts is warranted (**line642-645**).

2. Could you provide more detailed criteria used to recruit participants for the short-term exposure study (e.g., exclusion of confounding comorbidities, differences in daily environments, prior eye conditions)?

Response: Thanks for your question. A total of 28 participants aged 21-68 years were recruited from a community in Jinhua City, Zhejiang Province, China, in March 2023, the main inclusion and exclusion criteria are as follows:

Inclusion criteria:

- (1) Age \geq 18 years;
- (2) Residence in the study area for more than 6 months with no plan to move out;
- (3) Able to cooperate in completing questionnaires and examinations;
- (4) Signed informed consent.

Exclusion criteria:

- (1) History of ophthalmic disease or surgery;
- (2) The subject or his/her immediate family members within three generations have hereditary ophthalmic diseases;
- (3) History of other systemic diseases related to ophthalmic diseases, including hypertension, diabetes mellitus, hyperthyroidism and so on.
- (4) History of smoking or long-term exposure to smoke and dust.

We have supplemented the above content in the Methods section (**line244-251**).

3. Have you considered or planned any longitudinal measurements (more than a single day of exposure) in human participants to better capture real-world exposure fluctuations?

Response: Thanks for your advice. The goal of the current study is to initially screen for PM2.5-related disease biomarkers in tear fluid, and we are in the process of establishing a larger and more prolonged exposure cohort, which will be further validated for sensitivity and specificity and translational applications in the future.

4. For the animal model, have you contemplated alternative PM2.5 exposure methods (e.g., inhalation chambers) that might replicate real-world exposures more closely?

Response: We agree with your suggestion that inhalation chambers are closer to real-world exposures, and that our current experimental methodology for eye drops better reflects the effects of direct contact exposure to airborne PM2.5 on the cornea. The rationale for the exposure treatment rat models is explained below. According to the data provided by the National Ministry of Environmental Protection of China and Environmental Protection Agency of US, 100~250 $\mu\text{g}/\text{m}^3$ was selected as the PM concentration of mild PM2.5 pollution. Taking an adult engaged in indoor work as an example, the average radius of an adult's cornea is approximately 6 mm. The estimate duration of daily outdoor activity is set as 2 hours (7200 seconds) and the movement velocity of an adult with light activity is estimated as 1 m/s. Then the mass of daily PM2.5 exposure in is calculated as follows:

$$\begin{aligned} \text{mass of daily PM2.5 exposure} &= V \cdot c = S \cdot v \cdot t \cdot c \\ &= \pi \times (0.006 \text{ m})^2 \times 1 \text{ m/s} \times 7200 \text{ s} \times (100 \sim 250) \mu\text{g}/\text{m}^3 \\ &\approx (80 \sim 200) \mu\text{g} \end{aligned}$$

V - volume of air contact with one cornea per hour

c - concentration of airborne PM2.5

S - surface area of the cornea

v - movement velocity of a person in light activity

t - PM2.5 exposure time

m - meter

s - second

m^3 - cubic meter

Based on this estimation, the equal exposure mass in a rat should be 20~50 μg , considering that the average radius of a rat's cornea is approximately 3 mm. In this study, the volume of each eye drop was 5 μL , the daily PM2.5 exposure mass in a rat is calculated as follows: 1 $\text{mg}/\text{mL} \times 5 \mu\text{L} \times 4 \text{ times}/\text{day} = 20 \mu\text{g}$.

5. Do you have data showing whether repeated exposure or longer-term (beyond 24 hours) PM2.5 treatment in vitro affects HCECs differently, compared to your short- and medium-term exposures?

Response: Thanks for your questions. Our previous study (Fu *et al*, 2017) suggested by CCK8 and EDU labeling that exposure to PM2.5 in vitro for more than 24 hours caused more severe cellular damage in HCECs compared to short- and medium-term exposures.

6. Could you provide more quantitative metrics (e.g., scoring scales, defect area measurements) for corneal epithelial damage in animal models, in addition to representative images?

Response: We are grateful for your suggestions. We rated corneal damage based on fluorescein sodium staining of rat corneas, using the following criteria: no staining = 0; 1–30 punctate stains = 1; >30 punctate stains but no fusion = 2; fusion of punctate stains = 3 (We have supplemented this content in the Methods section (**line285-288**)). Statistics showed that damage ratings were significantly higher in the PM2.5-exposed group than in the control group after 3 weeks of exposure (**Fig.1D**).

7. In the AFM and traction force microscopy (TFM) experiments, could you elaborate on the number of cells measured, indentation protocols, and statistical power to ensure that these biomechanical assays are robust?

Response: Thank you very much for your comment. Regarding the **number of cells measured** and **indentation protocols**, we have supplemented the relevant parameters and experimental procedures in the Methods section. Concerning **statistical power**, we have added the sample size (n), relevant statistical test methods, and the significance level to the corresponding figure legends. We have now made the following modifications to the main text.

Our modifications

1) Regarding number of cells measured in TFM, in the Methods section, **line239-241**, we added: 'All experiments described above were performed in at least three independent replicates. In each replicate, measurements and subsequent calculations were carried out on datasets obtained from 8-15 distinct cells per experimental group.'

Regarding number of cells measured in AFM, in the Methods section, **line211-213**, we added: 'To characterize the stiffness of individual cells, three force-indentation curves were acquired per cell. For each experimental group, measurements were performed on 8-20 distinct cells.'

2) Regarding indentation protocols in AFM, in the Methods section, **line191-203**, we added: 'AFM measurements were conducted using an Asylum MFP-3D-BIO atomic force microscope equipped with pyramidal tipped OMCL-TR400PSA cantilevers (spring constant: 0.08 N/m). Prior to stiffness measurements, all cantilever sensitivity calibrations were performed in situ under liquid conditions using the instrument's integrated calibration routines; calibration was repeated independently before each measurement group. A consistent experimental protocol was employed across all measurements: the probe loading speed was fixed at 500 nm/s, and the sampling rate was set to 1.7 kHz. Due to the varying mechanical properties of cells, the indentation depth did not correspond directly to the applied displacement. Consequently, the loading displacement was set to 1.2 μm , but for subsequent stiffness calculation, only the segment corresponding to an indentation depth of 600 nm was used for fitting.'

3) Regarding statistical power, we revise the legends of following figures:

‘Figure 2. ...**B.** Cellular tensile stress of HCECs exposed to PM2.5 (with or without blebb) for 3 h was measured by traction force microscopy (n=40 for control, n= 32 for PM, n=25 for PM+blebb). **C.** Cellular stiffness of HCECs exposed to PM2.5 for 3 h measured by atomic force microscopy (n=8)...Data in (A-E) are graphed as mean ± standard deviation with individual values shown as circles. Statistical analysis was conducted using the unpaired t-test in (A-E). The p-values are labeled in the figure.

Figure 4. ...**C.** Cellular tensile stress of NC and KO exposed to PM2.5 for 3 h measured by TFM (n= 32 for NC, n= 38 for NC+PM, n = 25 for KO, n=34 for KO+PM). **D.** Cellular stiffness of NC and KO exposed to PM2.5 for 3 h measured by AFM (n= 12 for NC, n= 16 for NC+PM, n = 17 for KO, n=12 for KO+PM)...Data in (A-E, G, H) are graphed as mean ± standard deviation with individual values shown as circles. Statistical analysis was conducted using the unpaired t-test in (A-E, G, H). The p-values are labeled in the figure.’

8. What role might integrins or focal adhesions (FAK-paxillin) play upstream in your proposed PAI-2/myosin II/F-actin/YAP pathway? Do you see consistent alterations in these adhesion-associated proteins?

Response: Thanks for your enlightening questions. Some evidences implied that PAI-2 may relate to integrins or focal adhesions. PAI-2 has been found to co-localize with actin in focal adhesions (Schroder *et al*, 2019). Furthermore, Kindlin-2, a member of an actin cytoskeleton organizing and integrin activator proteins, has been shown to regulate PAI-2 in breast cancer (Sossey-Alaoui *et al*, 2019). We detected the mRNA expression of some integrins in our cell models (with PM2.5 exposure for 3h), we found that the expression of ITGA6 and ITGB8 mRNA were obviously decreased with PAI-2 knockout. Since integrins have been clarified to regulate myosin II/F-actin/YAP pathway in several studies (Panciera *et al*, 2017), we proposed that PAI-2 may regulate myosin II/F-actin/YAP pathway through integrins.

9. Do you suspect that PM2.5-induced oxidative stress leads to PAI-2 polymerization or altered secretion? Could you examine redox states or measure polymeric forms of PAI-2?

Response: We are also very interested in this issue and are continuing to explore it further. At present, we are not certain that PAI-2 polymerization is necessarily caused by oxidative stress, as previous studies have indicated that PAI-2 spontaneously polymerizes as its expression level increases. We have established several cell lines expressing different levels of PAI-2 and found a significant negative correlation between PAI-2 expression levels and excretion levels (data not shown). Therefore, we are more inclined to believe that reduced PAI-2 excretion is related to expression levels. In the future, we will further assess PAI-2 polymerization levels and explore whether PAI-2 can function as a membrane-less organelle through phase separation to perform additional functions.

10. Beyond autophagy and IL-1 α expression, have you explored whether YAP might regulate other downstream genes (e.g., those involved in epithelial barrier function) that could contribute to corneal pathology?

Response: Thank you for your insightful question. In our preliminary research, we found that long-term exposure to PM_{2.5} in rat models caused a reduction in ZO-1 (Hao *et al.*, 2023), a protein related to corneal epithelial barrier function. Therefore, we detected ZO-1 mRNA expression in HCECs and found that ZO-1 gene expression levels recovered after YAP function was inhibited by Verteporfin (VTP), suggesting that YAP may regulate epithelial barrier function through ZO-1.

11. Could you clarify the precise time course of PAI-2, myosin II, and YAP activation versus cellular damage? For instance, are there distinct "waves" of activation (1-3 hours, 6-12 hours, etc.)?

Response: Thank you for your question. Our current research has observed that PAI-2 is activated in the early stages of PM_{2.5} exposure (1-6 hours), accompanied by the activation of stress fibers, i.e., myosin II activation and F-actin polymerization, which are the primary factors driving cellular biomechanical changes. In the late stages of PM_{2.5} exposure, YAP nuclear translocation occurs, with significantly enhanced autophagy and inflammatory processes, leading to significant cellular damage. Therefore, as you mentioned, there is an "wave-like" initiation in the early stages.

12. How do you operationally define "subclinical stage" in the animal and cell experiments? Are there objective markers you use to confirm the absence of overt damage while capturing biomechanical changes?

Response: We define the subclinical phase as a period in which damage or disease has not yet manifested but molecular biological changes have already occurred, indicating that damage or disease is imminent. We used fluorescein sodium staining to determine that there was no significant corneal epithelial damage in animal models exposed to PM_{2.5} in the early stages, but there was an increase in PAI-2 expression levels. In cell models, we used the CCK8 assay to determine that there was no obvious cellular damage after short-term exposure to PM_{2.5}, but PAI-2 expression levels were elevated at this time.

13. Could you offer details on the reproducibility and sensitivity of your tear PAI-2 detection method (e.g., sample processing, assay validation, potential interfering substances in tears)?

Response: Thanks for your questions. We have supplemented details on the reproducibility and sensitivity of tear PAI-2 detection method in the Methods section (line397-421). Here are as follows:

After rat or human tears were collected, they were immediately sent to the laboratory for testing or stored at -80 °C. Before testing, centrifuge the tear sample for 20 minutes at 1000 g and take the supernatant for testing. The test strictly followed the steps outlined in the ELISA kit to ensure **reproducibility**. According to the manual, for **sensitivity**, the minimum detectable dose of rat PAI-2 is typically less than 1.0 pg/mL, the minimum detectable dose of human PAI-2 is typically less than 0.1 ng/mL. **This assay recognizes recombinant and natural rat/human PAI-2. No significant cross-reactivity or interference was observed.** The detailed assay procedures are as follows:

- a. Prepare all reagents before starting assay procedure. It is recommended that all Standards and Samples be added in duplicate to the Microtiter plate.
- b. Add 50µl of Standard or Sample to the appropriate wells. Blank well doesn't add anything.
- c. Add 100µl of Enzymeconjugate to standard wells and sample wells except the blank well, cover with an adhesive strip and incubate for 60 minutes at 37°C.
- d. Wash the Microtiter Plate 4 times.
- e. Add Substrate A 50µl and Substrate B 50µl to each well. Gently mix and incubate for 15 minutes at 37°C. Protect from light.
- f. Add 50µl Stop Solution to each well. The color in the wells should change from blue to yellow. If the color in the wells is green or the color change does not appear uniform, gently tap the plate to ensure thorough mixing.
- g. Read the Optical Density (O.D.) at 450 nm using a microtiter plate reader within 15 minutes.
- h. Calculate the O.D. value for each standard and sample. All O.D. Values are subtracted by the value of the blank well before result interpretation. Construct the standard curve using graph paper or statistical software. To determine the amount in each sample, first locate the O.D. value on the Y-axis and extend a horizontal line to the standard curve. At the point of intersection, draw a vertical line to the X-axis and read the corresponding concentration. A new standard curve should be obtained for each experiment.

14. Were any other tear proteins or inflammatory cytokines similarly evaluated but not included in the main text? If so, might they corroborate or contrast with PAI-2 findings?

Response: Since our research focused on PAI-2 as a key target and we hoped that it could serve as an early diagnostic indicator, we did not conduct further tests on other proteins. We collected rat tear fluid 2 days after PM2.5 exposure for mass spectrometry (**Fig.EV6**), and we found that some actin organization-related proteins were significantly changed. Perhaps some of these proteins also have the potential to become new observational indicators.

15. Have you explored a dose-response or gradient exposure in the animal or cell models to link PM2.5 concentration with corneal biomechanical and biochemical changes?

Response: Thank you for your question. Prior to the experiment, we conducted a preliminary experiment on the effect of PM2.5 concentration gradients on HCECs cellular tensile stress. HCECs were treated with PM2.5 concentrations of 0, 50, and 100 µg/ml for 3 hours, and TFM detection revealed that cellular tensile stress increased with increasing PM2.5 concentration.

16. How long is siRNA effectively delivered to corneal cells via DLin-MC3-DMA LNPs following topical application (e.g., hours, days)? Could you address the duration of knockdown effects?

Response: We delivered luciferase mRNA via DLin-MC3-DMA LNPs, and we found that luciferase could be expressed in rat corneal epithelial cells 6 hours after delivery via eye drops. In the model, we administered eye drops four times a day, so the knockdown effect was sustained.

17. Have you assessed potential off-target or inflammatory effects of LNP-siPAI-2 therapy, especially with repeated instillations over multiple weeks?

Response: We detected the expression levels of PAI-2 and IL-1a in rat corneal epithelium through immunofluorescence staining and immunohistochemistry (**Fig.7C-D**). We found that LNP-siPAI-2 could effectively knock down PAI-2 without obvious off-target effects or inflammatory reactions.

18. You note that PM2.5 exposure also affected the lens and retina; do you have mechanistic data (inflammatory or oxidative) clarifying how corneal epithelial damage propagates or correlates with deeper ocular structures?

Response: Thank you for your forward-thinking question. Our preliminary research indicates that PM2.5 particles cannot directly penetrate the interior of the eyeball, but they can induce damage to the lens (Sheng *et al*, 2024) and retina (Gu *et al*, 2022) by triggering an inflammatory response within the eye. However, there is currently no evidence indicating the specific source of this inflammation. Our current research initially aimed to investigate whether inhibiting PAI-2 expression could mitigate PM2.5-induced corneal damage. Unexpectedly, we also observed reduced inflammation in the lens and retina, suggesting that PM2.5-induced inflammation may originate from the cornea. However, further mechanistic studies are required to confirm this hypothesis.

19. Are there additional autophagy markers (LC3 flux, p62/SQSTM1) or imaging techniques used to confirm autophagic activity, rather than just elevated expression of individual proteins?

Response: Our previous research has clearly demonstrated that exposure to PM2.5 enhances the autophagic response in HCECs, including increased expression levels of BECN1, ATG5, and LC3BII, as well as an increased LC3BII/LCBI ratio, and has established that autophagy plays a key role in PM2.5-induced cellular damage (Fu *et al*, 2017). This study focuses more on explaining how biomechanics induces corneal damage, with autophagy or inflammation potentially serving as damage mechanisms, so a more comprehensive assessment of the autophagy signaling pathway has not yet been conducted.

20. Did you measure oxidative stress (e.g., ROS production, antioxidant defenses) in corneal cells or tissues under PM2.5 exposure, and how might that tie in with your mechanotransduction findings?

Response: Thanks for your question. In our previous study, we measured ROS levels in rat tears at different PM2.5 exposure time points and found that ROS levels in tears increased significantly after short-term exposure (2 days), but decreased after long-term exposure (3 weeks) and were even lower than those in the control group. Therefore, we speculated that oxidative stress may not be the key factor in PM2.5-induced corneal damage, and we did not pay much attention on it.

Reviewer 2#

In this new study, the authors report that exposure to fine particulate matter (PM2.5) with a diameter of 2.5 μm alters the mechanical properties of the cornea and corneal cells. However, given their previous research identifying plasminogen activator inhibitor-2 (PAI-2) as a mediator of PM2.5-induced cellular changes, the novelty of the current study is limited, primarily focusing on the observation of a transient contractility response to PM2.5. Furthermore, the mechanistic links between the presented observations, specifically the causal relationships between PAI-2 and transient contractility, and between contractility and autophagy, are not adequately established, limiting the study's contribution to a descriptive account.

The conclusions drawn by the authors are not fully supported by the presented data. While they suggest altered myosin II activity ("indicating PAI-2 regulated myosin II activity"), their analysis is limited to reporting a transient increase in myosin II expression (Figures 2b and 4b). They do not provide direct measurements of myosin activity, such as phosphorylation, which is typically required to substantiate such claims. Similarly, they report changes in mRNA PAI-2 expression following short-term PM2.5 exposure (at 3h), but corresponding protein level data (measured by immunofluorescence at 48 hours for example), are absent. This omission raises concerns regarding PAI-2's role in the observed short-term (3-hour) response. Figure 5b and 5f reveal differences in total YAP levels, which make the interpretation of the experiment very difficult as these differences could lead to a decrease in p-YAP without necessarily reflecting changes in YAP activity. To accurately assess the impact on YAP activity, the authors should analyze YAP nuclear localization (by IF, preferably with higher quality imaging to distinguish between nuclear and cytoplasmic localization). Additionally, the fluorescence imaging lacks adequate resolution and quantification to support robust conclusions regarding subcellular actin localization (figure 4). Overall, a more rigorous methodology, incorporating gold-standard quantitative techniques, is essential to validate these findings and demonstrate a significant impact.

Referee #2 (Remarks for Author):

The authors report that PM2.5 exposure (2.5 μm) induces alterations in corneal mechanical properties and cellular changes prior to symptom onset, evidenced by changes in mechanical properties, myosin expression, and PAI-2 expression. However, the study lacks critical logical connections, particularly regarding the mechanisms by which PAI-2 induces transient changes in myosin expression and the link between contractility and autophagy. Furthermore, inconsistencies exist between the authors' claims and the supporting data. Overall, a more rigorous methodology, incorporating gold-standard quantitative techniques, is essential to validate these findings and demonstrate a significant impact.

Response: Thank you for raising your concerns about this study. Our current research has demonstrated that PAI-2 can regulate changes in early cellular mechanical properties following PM2.5 exposure, and that PAI-2 influences the autophagy process by regulating the activation of key stress fiber myosin II and F-actin. We acknowledge that there are still some shortcomings in our research. In response to your suggestions, we have refined the relevant experiments, as detailed in the following response.

The authors claim that PM2.5 exposure affects corneal hysteresis, but corneal hysteresis itself is not assessed. Although several methods are used to evaluate the mechanical properties of cornea (young modulus etc...), corneal hysteresis-reflecting the eye's ability to absorb shock and overall biomechanical characteristics-is not directly tested in this study.

Response: Thank you for your question. We apologize for the inaccuracy in the description. We assessed corneal biomechanical parameters in humans exposed to short-term high/low concentrations of PM2.5 using the Corvis ST corneal biomechanical analyzer. The principle involves recording the deformation process of the cornea after compression using high-speed dynamic imaging technology, combined with air pressure pulse and corneal response data, to calculate corneal biomechanical parameters (such as corneal stiffness and deformation amplitude). While several of these indicators can also reflect the eye's ability to absorb shock and overall biomechanical characteristics, they do not directly represent corneal hysteresis. Therefore, we have removed 'corneal hysteresis' to avoid misunderstanding.

The authors suggest altered myosin II activity ("indicating PAI-2 regulated myosin II activity"), their analysis is limited to reporting a transient increase in myosin II expression (Figures 2b and 4b). They do not provide direct measurements of myosin activity, such as phosphorylation, which is typically required to substantiate such claims.

Response: Thank you very much for your suggestion. Phosphorylated myosin II does indeed better reflect the activation state of myosin II, and adding data related to phosphorylated myosin II will make our conclusions more rigorous. Therefore, we have added detection of phosphorylated myosin II. Through immunofluorescence staining, we found that phosphorylated myosin II significantly increased within 1–6 hours of PM2.5 exposure (Fig.2A), and phosphorylated myosin II was lower than the negative control after PAI-2 knockout (Fig.4A, EV3A). We also established PAI-2-overexpressing HCECs (OE) and the negative control (OENC) (Fig.EV2D-E), where the level of myosin II phosphorylation in OE was higher than that in OENC (Fig.4B, EV3B), suggesting that PAI-2 regulates myosin II activation in HCECs under short-term PM2.5 exposure.

The authors report changes in mRNA PAI-2 expression following short-term PM2.5 exposure (at 3h), but corresponding protein level data (measured by immunofluorescence at 48 hours for example), are absent. This omission raises concerns regarding PAI-2's role in the observed short-term (3-hour) response.

Response: Thank you for your suggestion. We supplemented the study with western blot analysis to examine changes in PAI-2 protein levels 3 hours after PM2.5 exposure. Please refer to **Figure 3C** for details.

Figure 5b and 5f reveal differences in total YAP levels, which make the interpretation of the experiment very difficult as these differences could lead to a decrease in p-YAP without necessarily reflecting changes in YAP activity. To accurately assess the impact on YAP activity, the authors should analyze YAP nuclear localization (by IF, preferably with higher quality imaging to distinguish between nuclear and cytoplasmic localization).

Response: Thank you for your suggestion. We supplemented the study with immunofluorescence staining of YAP to clarify its nuclear localization. The results show that the proportion of YAP nuclear translocation increased after PM2.5 exposure, while it decreased significantly after PAI-2 knockout (**Fig.5B**), consistent with the conclusions reflected in the previous phosphorylation levels.

Additionally, the fluorescence imaging lacks adequate resolution and quantification to support robust conclusions regarding subcellular actin localization (figure 4).

Response: We apologize for not providing images with sufficient resolution. We have re-uploaded figures with higher resolution to show the F-actin localization more clearly. We also uploaded the source data of images this time. In addition, we have added quantitative results for the F-actin fiber density (**Fig.4G**).

Minor points:

Line 67: The authors' assertion that this model is 'the best' for studying the relationship between biomechanical cues and disease lacks sufficient substantiation. This claim is subjective and not supported by comparative data or a clear rationale.

Response: Thank you for pointing out this issue. We have changed 'the best model' to 'a suitable model' (**line83**).

Line 255 the results showed an increase in Young's modulus upon 3-h exposure, indicating an improvement in cell cortical stiffness (Fig.2c-2e). What do the authors mean by improvement?

Response: Thank you for pointing that out. Here, it refers to an increase in cellular cortical stiffness. The use of 'improvement' may have been inaccurate and caused misunderstanding. We have revised it to 'enhanced cell cortical stiffness' (**line477**).

Referee #3 (Comments on Novelty/Model System for Author):

The study combines cell, animal, and human models, enabling in-depth exploration of the mechanisms of PM2.5-induced corneal diseases from different levels.

Reviewer 3#

Overall Evaluation:

This manuscript by Hao et al. explores the role of corneal biomechanical cues in PM2.5-induced corneal diseases, with a focus on the mediator PAI-2. The study is of great significance as it provides new insights into the early diagnosis and intervention of corneal diseases. However, it also has several areas that need improvement.

Major revisions:

1. After adding CytoD, did the structure of PAR disappear? The authors are recommended to explain the reason more clearly.

Response: Thank you for your question. We apologize for not explaining this issue clearly. PAR is a structure composed of actin, and CytoD can inhibit actin polymerization, thereby destroying the structure of PAR and causing it to disappear. We have restated this in the Results section (**line549-551**).

2. Overexpress PAI-2 to further investigate whether the role of PAI-2 in the PAI-2/myosin II/F-actin/YAP pathway is the same as that in the knockout experiment.

Response: Thank you for your suggestion. Overexpression of PAI-2 can further validate the PAI-2/myosin II/F-actin/YAP pathway. Therefore, we constructed HCECs overexpressing PAI-2 (**Fig.EV2D-E**) and found that, after 3 hours of PM2.5 exposure, the phosphorylation level of myosin II was significantly higher than that of the control group (**Fig.4B, EV3B**), F-actin polymerization was significantly increased (**Fig.EV4A**), and the proportion of PAR structure was significantly lower than that of the control group (**Fig.EV4A**). This suggests that PAI-2 overexpression promotes myosin II activation and F-actin polymerization. After 24 hours of PM2.5 exposure, the nuclear localization of YAP in the PAI-2 overexpression group was significantly higher than that in the control group (**Fig.EV4B**), further confirming that PAI-2 regulates YAP nuclear localization. Therefore, the conclusion obtained after PAI-2 overexpression indicated that PAI-2 activates myosin II/F-actin/YAP pathway, which is consistent with conclusion from the gene knockout experiment.

3. If elements such as Se, Hg, Cd, Al, and OC are removed from PM2.5, will it no longer cause corneal damage?

Response: Thank you for your question. We would love to discuss this interesting hypothesis with you. Firstly, while our BKMR analysis identifies these components as key contributors within the mixture, their removal may not fully eliminate PM2.5-induced corneal effects. Other unmeasured constituents (e.g., microplastics, ultrafine particles) or synergistic effects between the rest components could still play a role. Secondly, our findings suggest that these components may be associated with altered tear PAI-2 levels, a potential biomarker of corneal injury. Although PAI-2 has been implicated in multiple organ damage caused by PM2.5, it is not the only molecule that causes corneal damage. Finally, we agree that targeted component reduction (e.g., regulating metal-rich emissions) could mitigate risks, but comprehensive strategies addressing PM2.5 holistically are warranted. Future mechanistic studies (e.g., in vitro/ex vivo corneal models with controlled constituent exclusion) could help clarify this.

4. When reading the article, PAI-2 seems more like a target for treating PM2.5-induced corneal damage. However, to prove that it is a specific biomarker for early real-time diagnosis, the authors should conduct more detailed quantitative measurements of PAI-2 in tears at finer time points after PM2.5 exposure.

Response: Thank you for this critical insight. We agree that establishing PAI-2 as a real-time diagnostic biomarker requires rigorous temporal profiling. In our rat model, we observed that PAI-2 levels in tears exhibit a dynamic response, underscoring its potential as an early indicator. While our preliminary human study supports this trend, we acknowledge that the current data, limited by sample size and time points, are insufficient to fully define its diagnostic specificity. In direct response to this comment, we will develop a dedicated longitudinal study plan to quantitatively track PAI-2 level changes post-exposure with high temporal resolution. This research will lay the foundation for early identification and monitoring of PM2.5-induced corneal diseases in the future.

5. Is PAI-2 related to corneal damage caused by other inappropriate mechanical stimuli, such as wearing contact lenses?

Response: Your question is very forward-thinking. Currently, there is very little research on PAI-2 in the cornea. In addition to our previous studies (Lyu *et al*, 2020a), another study has shown that PAI-2 expression increases after corneal bacterial infection (Berk *et al*, 2001). Additionally, another study found that PAI-2 levels in tears temporarily increase after corneal laser surgery, suggesting that external mechanical stimulation may trigger an increase in PAI-2 levels in tears (Csutak *et al*, 2008). However, whether PAI-2 is associated with corneal damage caused by other mechanical stimuli remains unclear.

Minor points:

1. The full names of abbreviations such as LNP, siRNA, PM2.5, DSPC, ROCK, etc. should be provided.

Response: Thanks for pointing out that, we have supplemented the full names of abbreviations throughout the text.

2. The coordinates in Figure 2d, 2g, 4c, and 4f are unclear.

Response: Thanks for pointing out that, we have modified the image to make the coordinates clearer (**Fig.2B-C, 4C-D**).

3. What are the sequences of the siPAI-2 and si-NC?

Response: Thank you for the reminder. We have added their sequence to the Methods section (**line272-276**).

The sequence of siPAI-2: sense (5'-3') GAUACAUAAGGACCUGAA(dT)(dT), antisense (5'-3') UUCAGGUCCUUAUGUAUC(dT)(dT);

The sequence of siNC: sense (5'-3') UUCUCCGACAGUGUCACGU(dT)(dT), antisense (5'-3') ACGUGACACUGUCGGAGAA(dT)(dT).

4. The following articles are relevant to your research and could be cited:

(1) Xiaoyu He, Yidian Fu, Liang Ma, Yizheng Yao, Shengfang Ge, Zhi Yang, Xianqun Fan. AAV for Gene Therapy in Ocular Diseases: Progress and Prospects. Research. 2023;6:0291

(2) B. Kong, C. Qi, H. Wang, T. Kong, Z. Liu, Tissue adhesives for wound closure. Smart Med. 2023, 2(1), e20220033.

Response: Thank you for your reminder. We have cited the aforementioned literature in the relevant section (**line82, line778**).

Reviewer 4#

In the manuscript titled "Corneal biomechanical cues mediated by PAI-2: the origin of PM2.5-induced corneal disease", Hao et al. Investigate the role of biomechanical changes and the serpin protein PAI-2 in the early pathogenesis of PM2.5-induced corneal injury. This study introduced that extracellular PAI-2 in tears may serve as a biomarker for early diagnosis, and proposed an LNP-based siRNA delivery system targeting intracellular PAI-2 as a preventive treatment strategy. However, there are some concerns about this manuscript, as outlined below:

Major concerns:

1. While the rationale was briefly mentioned, the authors should better elucidate the rationale for the PM2.5 eye drop model was chosen over atmospheric exposure. Eye drop administration may induce more direct and acute corneal injury, which could not fully reflect real-world chronic exposure dynamics. Clarification on this choice is essential, as environmental chamber-based models may offer greater physiological relevance. More data and a comprehensive discussion are warranted to support the translational applicability of the model and to better explore the toxic effects of PM2.5 in corneal disease.

Response: Thanks for your question. We agree that environmental chamber-based models may be closer to real-world exposures, and that our current experimental methodology for eye drops better reflects the effects of direct contact exposure to airborne PM2.5 on the cornea. In order to make the exposure levels more closely resemble atmospheric exposure, we converted the average exposure levels in the atmosphere to eye drop doses. The rationale for the exposure treatment rat models is explained below. According to the data provided by the National Ministry of Environmental Protection of China and Environmental Protection Agency of US, 100~250 $\mu\text{g}/\text{m}^3$ was selected as the PM concentration of mild PM2.5 pollution. Taking an adult engaged in indoor work as an example, the average radius of an adult's cornea is approximately 6 mm. The estimate duration of daily outdoor activity is set as 2 hours (7200 seconds) and the movement velocity of an adult with light activity is estimated as 1 m/s. Then the mass of daily PM2.5 exposure in is calculated as follows:

$$\begin{aligned}\text{mass of daily PM2.5 exposure} &= V \cdot c = S \cdot v \cdot t \cdot c \\ &= \pi \times (0.006 \text{ m})^2 \times 1 \text{ m/s} \times 7200 \text{ s} \times (100\sim 250) \mu\text{g}/\text{m}^3 \\ &\approx (80\sim 200) \mu\text{g}\end{aligned}$$

V - volume of air contact with one cornea per hour

c - concentration of airborne PM2.5

S - surface area of the cornea

v - movement velocity of a person in light activity

t - PM2.5 exposure time

m - meter
s - second
m³ - cubic meter

Based on this estimation, the equal exposure mass in a rat should be 20~50 µg, considering that the average radius of a rat's cornea is approximately 3 mm. In this study, the volume of each eye drop was 5 µL, the daily PM2.5 exposure mass in a rat is calculated as follows: 1 mg/mL × 5 µL × 4 times/day = 20 µg.

2. PM2.5 increases the risk of diseases such as keratitis and dry eye, yet these conditions differ in onset and progression. The authors should clarify the rationale for classifying observed changes as subclinical. Specifically, the different PM2.5 exposure durations in human, animal, and cell models raise questions about whether the findings truly reflect the subclinical biomechanical stage rather than the early clinical stage. For example, increased corneal thickness in rats after two days of PM2.5 eye drops may already represent disease onset (Figure 1e). Greater clarity is needed to distinguish subclinical from clinical stage responses across models.

Response: Thank you for your question. The subclinical phase is defined as a period during which molecular biological changes have occurred but no clear damage is yet evident. The clinical early phase is defined as a period during which clear damage is present but is still in its early stages. Our previous studies have shown that in cellular models, PM2.5 exposure for 12 hours can result in noticeable damage, and when exposure continues for 24 hours or longer, the damage gradually worsens (Fu *et al*, 2017). Therefore, 12 hours better reflects the clinical early phase. However, there is almost no damage observed before 6 hours, so we selected 3 hours as the observation point for the subclinical phase. In animal models, our previous studies found that PM2.5 exposure for one week can already show symptoms of corneal diseases such as corneal epithelial defects and reduced tear secretion (Lyu *et al*, 2020b). When exposure is extended to two weeks or three weeks, the damage gradually worsens (Lyu *et al*, 2020b), Therefore, one week of exposure can be classified as the clinical early stage. However, at two days of exposure, although rat corneas thicken due to cellular reactive proliferation, corneal damage cannot yet be observed through clinical examination, so it remains in the subclinical stage. Similarly, in human models, the observation time points correspond to periods when the subjects have no clear corneal diseases, so they can be considered the subclinical stage.

3. The authors should consider the potential impact of participant age range (21-68 years) on the study outcomes. Age may be a factor influencing corneal biomechanical properties and tear film composition, including secretion volume and protein content. These changes alter tissue responsiveness to PM2.5 exposure and affect baseline or induced levels of tear PAI-2. It is recommended that the authors include age as a covariate in their statistical models or conduct age-stratified analyses to determine whether age modulates the ocular response to PM2.5. Addressing this would enhance the robustness and generalizability of the findings.

Response: We sincerely appreciate the reviewer's thoughtful suggestion regarding the potential influence of age on corneal responses. We agree that age-related changes in ocular surface physiology could modify both baseline tear composition and PM2.5 susceptibility. In our original analysis, we had already included age and gender as covariates in accordance with the protocol of

the 'bkmr' package in R software.

4. The authors proposed a positive feedback loop wherein PAI-2 regulates YAP activation via myosin II and F-actin remodeling. While the data demonstrated that myosin II inhibitors (RI or blebb) altered YAP phosphorylation and reduced nuclear localization (Figure 5a), it would be beneficial to acknowledge that this evidence primarily supports an indirect regulatory relationship between F-actin and YAP activity. The experimental approach did not establish direct physical or biochemical interactions between these components. It suggests clarifying YAP modulation by F-actin likely occurs through alterations in cytoskeletal tension and mechanical properties rather than through direct molecular association (Figure 5h).

Response: Thank you for your question. Based on your suggestion, we further investigated changes in YAP by inhibiting F-actin polymerization under PM exposure using the F-actin polymerization inhibitor cytochalasin D (cytoD), to elucidate the regulatory role of F-actin on YAP. After adding cytoD, we found that YAP nuclear translocation was significantly reduced (**Fig.EV5A**), and the level of YAP phosphorylation was increased (**Fig.EV5C**), thereby confirming that F-actin polymerization can regulate YAP nuclear translocation.

Minor concerns:

1. In the Results section, please ensure that each result title is a conclusive statement based on the study findings. Avoid generic titles such as "PAI-2/myosin II/F-actin/YAP positive feedback loop regulates autophagy" or "PAI-2 exocytosis is affected by PM2.5." Additionally, each result should have only one title. For example, under the heading "Corneal biomechanical changes precede clinical symptoms under PM2.5 exposure," there are three subtitles. Please revise your section titles accordingly.

Response: Thank you for your suggestion. We have revised the title accordingly (**line433, 464, 517**).

2. Some figures should be reorganized, combining those with a similar experimental purpose. For example, Figure 1e-f measured rat corneal thickness and can be merged into a single figure. In addition, Figure 2a-b show immunofluorescence staining for nonmuscle myosin II. Figures 2j-k are a staining image and its corresponding quantification, not two separate experiments. Please reorganize similar experimental results into a single figure.

Response: Thank you for your suggestion. We have rearranged the layout to place similar experimental results within the same panel whenever possible. However, due to layout constraints, some results cannot be placed within the same panel, such as Figure 5A and 5C, and Figure 5B and 5D. We added note in legends to show they are from the same experiments. We appreciate your understanding.

3. In Figure 3b, the mRNA level of PAI-2 in rat corneas after short-term PM2.5 exposure. Please specify the exact exposure duration used in the short-term PM2.5 exposure. Additionally, please verify whether the figure legend is accurate and complete for Figure 3, and revise or remove any descriptions that are missing or not applicable.

Response: Thank you for your suggestion. The exact exposure duration used in the short-term PM2.5 exposure is 2 days for rat model, we have specified this in the figure legend (**line1047**).

Additionally, we inadvertently included the legend for other figures within the legend for Figure 3. Thank you for pointing this out; it has now been corrected.

4. Some results that are less connected to the main findings could be moved to the supplementary results section. For example, the knockout efficiency of PAI-2 shown in Figure 3d could be combined with Figures S3a and S3b.

Response: Thank you for your suggestions. We have rearranged the layout of panels in figures and moved some of the results that are less connected to the main findings to the Expanded View Figures section.

5. Please check the use of blebb or RI in Figure 5a, whether they were used in combination or applied individually in the experiment.

Response: Thank you for pointing this out. Blebb and RI are used separately and independently. We had incorrectly labeled them in the diagram and have now made the correction (**Fig.EV5B**).

6. The graphical abstract should clearly highlight findings derived from actual experimental results. Please explain the relationship between biomechanical response and PM2.5 exposure as shown in Figure 2i, particularly how it reflects the disease progression curve and the statistical methods used. If no such analysis was conducted, please clearly indicate that this is a conceptual illustration.

Response: Thank you for pointing this out. This curve provides a conceptual illustration of the correlation between mechanical response and disease progression caused by PM2.5 exposure; it does not reflect precise numerical changes. We have added this note in the legend (**line1039-1041**).

Reference:

- Berk RS, Katar M, Dong Z & Day DE (2001) Plasminogen Activators and Inhibitors in the Corneas of Mice Infected with *Pseudomonas aeruginosa*. *Invest Ophthalmol Vis Sci* 42: 1561–1567
- Csutak A, Silver DM, Tozsér J, Steiber Z, Bagossi P, Hassan Z & Berta A (2008) Plasminogen activator inhibitor in human tears after laser refractive surgery. *J Cataract Refract Surg* 34: 897–901
- Fu Q, Lyu D, Zhang L, Qin Z, Tang Q, Yin H, Lou X, Chen Z & Yao K (2017) Airborne particulate matter (PM2.5) triggers autophagy in human corneal epithelial cell line. *Environ Pollut* 227: 314–322
- Gu Y, Hao S, Liu K, Gao M, Lu B, Sheng F, Zhang L, Xu Y, Wu D, Han Y, *et al* (2022) Airborne fine particulate matter (PM(2.5)) damages the inner blood-retinal barrier by inducing inflammation and ferroptosis in retinal vascular endothelial cells. *Sci Total Environ* 838: 156563
- Hao S, Chen Z, Gu Y, Chen L, Sheng F, Xu Y, Wu D, Han Y, Lu B, Chen S, *et al* (2023) Long-term PM2.5 exposure disrupts corneal epithelial homeostasis by impairing limbal stem/progenitor cells in humans and rat models. *Part Fibre Toxicol* 20: 1–17
- Lyu D, Almansoob S, Chen H, Ye Y, Song F, Zhang L, Qin Z, Tang Q, Yin H, Xu W, *et al* (2020a) Transcriptomic profiling of human corneal epithelial cells exposed to airborne fine

- particulate matter (PM_{2.5}). *Ocul Surf* 18: 554–564
- Lyu D, Chen Z, Almansoob S, Chen H, Ye Y, Song F, Zhang L, Qin Z, Tang Q, Yin H, *et al* (2020b) Transcriptomic profiling of human corneal epithelial cells exposed to airborne fine particulate matter (PM_{2.5}). *Ocul Surf* 18: 554–564
- Pancieria T, Azzolin L, Cordenonsi M & Piccolo S (2017) Mechanobiology of YAP and TAZ in physiology and disease. *Nat Rev Mol Cell Biol* 18: 758–770
- Schroder WA, Hirata TD, Le TT, Gardner J, Boyle GM, Ellis J, Nakayama E, Pathirana D, Nakaya HI & Suhrbier A (2019) SerpinB2 inhibits migration and promotes a resolution phase signature in large peritoneal macrophages. *Sci Rep* 9: 12421
- Sheng F, Gu Y, Hao S, Liu Y, Chen S, Lu B, Chen L, Zhao W, Wu D, Xu Y, *et al* (2024) Ferroptosis is involved in the damage of ocular lens under long-term PM_{2.5} exposure in rat models and humans. *Ecotoxicol Environ Saf* 288: 117397
- Sossey-Alaoui K, Pluskota E, Szpak D & Plow EF (2019) The Kindlin2-p53-SerpinB2 signaling axis is required for cellular senescence in breast cancer. *Cell Death Dis* 10: 539

21st Oct 2025

Dear Prof. Fu,

Thank you for submitting your revised study. We have now received the reports from the referees. As you will see below, they are satisfied with the revisions, and I will therefore be able to accept your manuscript once the following editorial concerns are addressed:

1/ Referees' comments: please address the comments from referee #1.

2/ Manuscript text:

- Please remove the red font and only keep in track changes mode any new modification in the text.
- We note that you currently have together with you, a total of 5 co-corresponding authors. Do you confirm equal contribution of these 5 people, able to take full responsibility for the paper and its content? While there is no limit per se to the number of co-corresponding authors, 4 is rare, 5 even more so, and may not reflect as intended to the community. Similarly, 4 co-first authors are listed. Please confirm whether this is correct.
- All corresponding authors are required to supply an ORCID ID for their name upon submission of a revised manuscript. ORCID identifiers are missing for Yao, Chen, and Liu. We regret that we cannot do this linking on your behalf for security reasons. We also cannot add your ORCID number manually to our system because there is no way for us to authenticate this number with ORCID.
- Please correct the order and headings of the manuscript sections to: Abstract, Introduction, Results, Discussion, Methods, Acknowledgements, Disclosure and competing interests statement, References, Figure legends, Tables and their legends, Expanded View Figure legends.
- In the methods, please indicate the origin of the cells, and whether they were authenticated and tested for mycoplasma contamination. Please also provide antibody dilutions.
- Data availability section: Please note that this section is restricted to new primary data that are part of this study (remove "Any data and materials used in this analysis can be made available on request to the corresponding author except as restricted by material transfer agreements (MTAs)."). Please provide URLs to access your deposited datasets, and note that they have to be publicly available before acceptance of the manuscript.
- Please rename 'Declaration of Interests' to 'Disclosure statement and competing interests'.
- Please remove BioRender from the Acknowledgements, and add as a dedicated section of the Methods:

Graphics:

(some of the... OR Figure #... OR synopsis) Graphics were created with BioRender.com.

3/ Figures:

- Please remove the table legends from the manuscript text and ensure that they are included at the top of the page in each corresponding table. Please also ensure that the nomenclature is consistently either Table EV1 - 5, or Table 1 - 5.
- Please make sure that all figures/figure panels/tables are cited in the manuscript text, and in chronological order (currently, a callout is missing for Table EV5; Fig EV3A is cited before Fig EV2D-E).
- Please check the composition of your Figure 4A and Figure EV3A, as we noted possible similarities /re-use. Figure re-use is allowed, but must be indicated in the figure legend.

4/ Checklist:

- please fill in the section cell materials/authentication and mycoplasma contamination.
- please check whether you need to fill the section Ethics/specimen and field samples, as I don't think it applies to your study.

5/ Synopsis:

I introduced minor changes in your synopsis text, please let me know if you agree or amend as you see fit:

"The role of biomechanical cues in the progression of PM2.5-induced corneal disease was investigated, and PAI-2 was identified as a key molecule regulating biomechanical responses, with potential as a target for early diagnosis and intervention.

- PM2.5 exposure induces corneal biomechanical cues alterations, which precede corneal injury during the subclinical stage.
- PAI-2 regulates cellular mechanical response and drives PM2.5-related disease progression through a PAI-2/myosin II/F-actin/YAP positive feedback loop.
- PAI-2 in tears holds potential for mirroring PM2.5-induced disease progression and may be utilized for early diagnosis.
- LNP-siPAI-2 system can relieve PM2.5-induced ocular damage for early intervention."

Thank you for providing a nice visual abstract. I have cropped a small portion of this image to serve as thumbnail for the table of content on our webpage (attached); please let me know if you agree or provide an alternative image at the same dimensions (115px x 70px).

6/ As part of the EMBO Publications transparent editorial process initiative (see our Editorial at <http://embomolmed.embopress.org/content/2/9/329>), EMBO Molecular Medicine will publish online a Review Process File (RPF) to accompany accepted manuscripts.

This file will be published in conjunction with your paper and will include the anonymous referee reports, your point-by-point response and all pertinent correspondence relating to the manuscript. Let us know whether you agree with the publication of the RPF and as here, if you want to remove or not any figures from it prior to publication.

I look forward to receiving your revised manuscript.

Yours sincerely,

Lise Roth

***** Reviewer's comments *****

Referee #1 (Comments on Novelty/Model System for Author):

The manuscript is well built and robust, here are some strengths:

- Innovative Link to Biomechanics: The paper offers a fresh perspective that biomechanical cues precede overt tissue damage, providing an "early alert" concept with potential impact on how ophthalmic pathologies are diagnosed and monitored.
- Robust, Multiscale Approach: The authors' careful alignment of human observational data with rat models and in vitro experiments lends credibility to their mechanistic conclusions.
- Therapeutic Translation: Proposing (and testing) siRNA-laden lipid nanoparticles for topical corneal application showcases direct translational potential.

Referee #1 (Remarks for Author):

The authors responded adequately to all my comments.

I think that there are some residual limitations (small human cohort, largely cross-sectional human evidence; limited upstream adhesion data in the main figures; multiplicity control in figure-level stats) do not undermine the central claims, which are now backed by added mechanistic and quantitative evidence.

To mitigate these limitations, I am suggesting 3 small text modifications:

- Adding a brief note in the Discussion explicitly acknowledging the limited human sample size/timeframe
- Softening the mechanistic sentence regarding ITGA6/ITGB8
- Adding a one-line caveat on the translational gap between topical PM2.5 dosing and ambient exposure.

With these modifications, the manuscript will be suitable for publication.

Referee #2 (Remarks for Author):

The authors addressed the majority of my comments, and their revisions significantly strengthened the manuscript.

Referee #3 (Remarks for Author):

Is suitable for publication

Referee #4 (Remarks for Author):

This is a revised manuscript. Most of the comments have been addressed adequately in the current version.

The authors addressed the remaining formatting issues.

31st Oct 2025

Dear Prof. Ke,

Thank you for submitting your revised files. I am pleased to inform you that your manuscript is accepted for publication and is now being sent to our publisher to be included in the next available issue of EMBO Molecular Medicine.

Yours sincerely,
